# Deformation of feldspar at greenschist facies conditions – the record of mylonitic pegmatites from the Pfunderer Mountains, Eastern Alps

Felix Hentschel[1], Claudia A. Trepmann[1], Emilie Janots[2]

[1]Department for Earth and Environmental Sciences, Ludwig-Maximilians-Universität München, Germany
[2]Univ. Grenoble 1, ISTerre, 38041 Grenoble, France

*Correspondence to*: Felix Hentschel (felix.hentschel@lmu.de)

**Abstract.** Deformation microstructures of albitic plagioclase and K-feldspar were investigated in mylonitic pegmatites from the Austroalpine basement south of the western Tauern Window by polarized light microscopy, electron microscopy and electron backscatter diffraction to evaluate feldspar deformation mechanisms at greenschist facies conditions. The main
mylonitic characteristics are alternating almost monophase quartz and albite layers, surrounding porphyroclasts of deformed feldspar and tourmaline. The dominant deformation microstructures of K-feldspar porphyroclasts are intragranular fractures at high angle to the stretching lineation. The fractures are healed or sealed by polyphase aggregates of albite, K-feldspar, quartz and mica, which also occur along intragranular fractures of tourmaline and strain shadows around other porphyroclasts. These polyphase aggregates indicate dissolution-precipitation creep. K-feldspar porphyroclasts are partly
replaced by albite characterized by a cuspate interface. This replacement is interpreted to take place by interface-coupled dissolution-precipitation driven by a solubility difference between K-feldspar and albite. Albite porphyroclasts are replaced at boundaries parallel to the foliation by fine-grained monophase albite aggregates of small strain-free new grains mixed with deformed fragments. Dislocation glide is indicated by bent and twinned albite porphyroclasts with internal misorientation. An indication of effective dislocation climb with dynamic recovery, for example by the presence of
subgrains, is systematically missing. We interpret the grain size reduction of albite to be the result of coupled dislocation glide and fracturing (low-temperature plasticity). Subsequent growth is by a combination of strain-induced grain boundary migration and formation of growth rims resulting in an aspect ratio of albite with the long axis within the foliation. This strain-induced replacement by nucleation (associated dislocation glide and microfracturing) and subsequent growth is suggested to result in the observed monophase albite layers, probably together with granular flow. The associated quartz
layers, show characteristics of dislocation creep by the presence of subgrains, undulatory extinction and sutured grain boundaries. We identified two endmember matrix microstructures: (i) alternating layers of a few hundreds of µm width, with isometric, fine-grained feldspar (in average 15 µm in diameter) and coarse-grained quartz (a few hundreds of µm in diameter), representing lower strain compared to (ii) alternating thin layers of some tens of µm width composed of fine-grained quartz (< 20 µm in diameter) and coarse elongate albite grains (long axis a few tens of µm) defining the foliation,
respectively. Our observations indicate that grain size reduction by strain-induced replacement of albite (associated dislocation glide and microfracturing) followed by growth and granular flow simultaneous with dislocation creep of quartz are playing the dominating role in formation of the mylonitic microstructure.

# 1 Introduction

Assessment of the rheological behaviour of the continental crust requires an understanding of grain-scale deformation mechanisms of the main rock-forming minerals at not directly accessible depths. In deep parts of seismically active shear zones (10-20 km) the rheological behaviour is controlled by the deformation of granitoid rocks, mainly composed of feldspar and quartz, at greenschist facies conditions. A vast number of experimental studies exist to analyse the deformation mechanisms and to derive flow laws for high-temperature creep of feldspar (e.g. Gleason and Tullis, 1993; Kruse and Stünitz, 1999; McLaren and Pryer, 2001; Stünitz et al., 2003; Rybacki and Dresen, 2004) and quartz (e.g., Jaoul et al., 1984; Patterson and Luan, 1990; Gleason and Tullis, 1993; Hirth et al., 2001). However, the extrapolation of experimentally deduced flow laws for monomineralic material to the flow behaviour of polymineralic rocks at geologically reasonable conditions is problematic (e.g., Pfiffner and Ramsay, 1982; Tullis and Tullis, 1986; Paterson, 1987; Jordan, 1988). Also, the application of flow laws to model the rheological properties of the continental lithosphere (e.g. Brace and Kohlstedt, 1980; Kohlstedt et al., 1995) is a matter of debate (e.g., Rutter and Brodie, 1991; Burov, 2007; Bürgmann and Dresen, 2008; Burov, 2011). Uncertainties in models for the rheological properties of the continental lithosphere is partly due to a poor knowledge of the deformation mechanisms actually proceeding at depth as well as the interplay between multiple factors influencing rock strength such as stress variations, fluid content, and metamorphic reactions. The comparison of experimental results with microstructural and mineralogical observations of exhumed metamorphic granitoid rocks, which record the grain-scale mechanical and chemical transformations at depths, is therefore indispensable.

The extrapolation of experimental flow laws for dislocation creep of quartz to natural conditions is found to agree well to natural observations (e.g., Stöckhert et al., 1999, Hirth et al., 2001, Stipp et al., 2002). However, there are large discrepancies in experimental and natural observations on the most abundant mineral of the continental crust, feldspar. Deformation experiments suggest that dislocation creep of feldspar in high strain shear zones is dominant only at high temperatures above about 900°C (e.g., Rybacki and Dresen, 2004). In contrast, ductile deformation with grain-size reduction and formation of new feldspar grains, commonly assumed to imply dislocation creep, is observed already at greenschist facies conditions (Voll, 1976; Tullis, 1983; Gapais, 1989; Stünitz, 1993; Prior and Wheeler, 1999; Ishii et al., 2007). This discrepancy is partly due to the unclear and strongly varying contribution of brittle, dissolution-precipitation and crystal-plastic processes (e.g., Tullis and Yund, 1987; Fitz Gerald and Stünitz, 1993; Stünitz and Fitz Gerald, 1993; Tullis et al., 1996; Prior and Wheeler, 1999; Kruse and Stünitz, 2001; Ree et al., 2005; Menegon et al., 2006, 2008; Stünitz et al., 2003; Mehl and Hirth, 2008; Sinha, et al., 2010; Kilian et al., 2011; Brander et al., 2012; Mukai et al., 2014; Eberlei et al., 2014). Such a creep behaviour governed by the interaction of different deformation mechanisms and chemical reactions in the presence of fluids is especially difficult to assess in experimental approaches. This is partly because experiments have to be performed at high temperatures to realize feasible strain rates, which, however, affects phase assemblages and material properties, for example by partial melting. Therefore, activated mechanisms may strongly differ from those at natural strain rates and greenschist facies conditions. Tullis and Yund (1987) found in their deformation experiments at strain rates of $10^{-4}$ s-1 to $10^{-6}$ s-1

effective dislocation climb with subgrain formation and subgrain rotation only effective at temperatures >900°C. They concluded that optically visible subgrains in feldspar from low-grade rocks should not directly be assumed to arise from crystal plasticity but may arise from cataclasis and subsequent healing. Dislocation climb necessary for dynamic recovery and recrystallization requires intracrystalline diffusion. At temperatures <550°C, the NaSi ↔ CaAl interdiffusion rates for

plagioclase are very low (Yund, 1986; Korolyuk and Lepezin, 2009). In the presence of water, the diffusion coefficient has been interpreted to be higher (Yund, 1986), which was suggested to account for the weakening observed in experiments where fluid is present (e.g. Rybacki and Dresen, 2004). TEM studies also show that sufficient dislocation climb to produce subgrains is effective only at temperatures from the middle amphibolite upward (e.g. White, 1975; Stünitz et al., 2003). The experiments by Tullis and Yund (1985) show that grain boundaries may migrate into areas of higher dislocation density

introduced by microfracturing driven by the reduction in strain energy (Tullis and Yund, 1987) at conditions, at which recovery is not active. However, whether extrapolation to natural conditions is valid can only be evaluated by a comparison to natural microstructures.

To analyse the deformation behaviour of feldspar at greenschist facies conditions, we use in this study the record of mylonitic pegmatites of the Austroalpine basement south of the western Tauern Window and north of the Periadriatic line

(Fig. 1). They show a wide range of feldspar deformation microstructures and are compositionally and mineralogically relatively simple, as they are characterized by a Ca-poor bulk-rock composition (Stöckhert, 1987). The goal of this study is to correlate characteristic microstructures to specific processes responsible for their formation and to discuss the rheological behaviour of the rocks based on our findings.

## 2 Geological setting and sampling

The pegmatites occur within high-grade polymetamorphic upper Austroalpine basement rocks located between the western Tauern window in the north and the Deferegger-Antholz-Valser (DAV) shear zone in the south (Fig. 1; e.g. Hoffmann et al., 1983; Stöckhert, 1987; Stöckhert et al., 1999; Mancktelow et al., 2001; Schmid et al., 2004; Müller et al., 2000). Pegmatite crystallization age is generally assumed to be Permian (262 ± 7 Ma, Borsi et al. 1980) consistent with other pegmatite occurrences in the Austroalpine basement (e.g. Habler et al., 2009; Thöni and Miller, 2009). The pegmatites are

characterized by a Ca-poor composition originated from water-rich anatectic melts (Stöckhert, 1987, Schuster and Stüwe, 2008).

The intrusion of the Rensen and Rieserferner tonalites and related magmatic dikes into the Austroalpine basement rocks took place at ca. 30 Ma (Borsi et al., 1978; 1979; Steenken et al., 2000). The amount of uplift and erosion since the intrusion of these magmatic bodies is increasing from about 10 km in an eastern area of the Rieserferner to about 15 to 25 km in the

Rensen area in the west (Trepmann et al., 2004). The main activity of the DAV shear zone was at about the same time as the magmatic intrusions and is characterized by an oblique strike slip movement with some tens of kilometres of horizontal component (sinistral sense of shear) and a few kilometres of vertical component, where the northern part of the Austroalpine

basement is uplifted relative to the southern part (Borsi et al., 1978; Kleinschrodt, 1987; Schulz, 1989; Ratschbacher et al., 1991; Stöckhert et al., 1999; Mancktelow et al., 2001). It is accompanied by smaller sinistral shear zones to the north, some of which were already active in the Eocene (Mancktelow et al., 2001). The DAV shear zone is viewed as part of the Southern limit of Alpine Metamorphism (SAM) as defined by Hoinkes et al. (1999). To the north of the DAV shear zone,

tertiary ages for several mineral systems are widespread (Mancktelow et al., 2001; Schulz et al., 2008) and the metamorphic conditions are constrained to 450±50 °C and pressures of about 0.7 GPa by phase relations in the metapelitic Austroalpine basement rocks (Stöckhert 1982; 1987; Schulz et al., 2008; unpubl. data). South to the DAV shear zone, the Austroalpine basement rocks have been solely affected by minor metamorphism associated with brittle deformation, in accord with no resetting of the Rb/Sr biotite Permian ages of 288-299 Ma (Borsi et al., 1978; Stöckhert, 1982; Kleinschrodt, 1987; Schulz,

10  1994).

The association of the deformation microstructures recorded by the pegmatites with the Alpine history is discussed controversially (e.g. Mancktelow et al., 2001; Schulz et al., 2008). Stöckhert (1982, 1984, 1987) proposed that the annealed quartz and feldspar microstructures in the mylonitic pegmatites correlate to an early Alpine deformation stage at metamorphic temperatures of 450±50 °C, pressures of about 0.7 GPa and at about 100 Ma (white mica K-Ar data, Stöckhert,

1984). This age corresponds to the Eoalpine (Cretaceous) tectonometamorphic event recorded from other units of the Eastern Austroalpine basement (e.g. Thöni and Miller, 1996; Habler et al., 2009). A later (Oligocene) deformation stage at 300 to 350°C was proposed by heterogeneous high-stress quartz microstructures in quartz-rich lithologies of Austroalpine basement rocks related to the movement along the DAV shear zone (Stöckhert, 1982, 1984; Kleinschrodt, 1987; Stöckhert et al., 1999). Mancktelow et al.(2001), however, argued, based on microstructural and kinematic studies, that the deformation

microstructure of feldspar could also be Paleogene in age and they find no clear distinction between separate low- and high-T events.

We sampled about 100 pegmatites in three field campaigns in 2015 and 2016. Their appearance varies mostly between m- to cm-sized veins or lenses and occasionally km-sized bodies occur (Stöckhert, 1987, Hofmann et al., 1983). In a few cases a mineral zoning is present. Deformed pegmatites, in veins or layers, have a foliation, which is parallel to that of the host

gneisses. The largest pegmatite bodies appear macroscopically undeformed. We selected pegmatites with a pronounced foliation and stretching lineation (Fig. 2a; Appendix 1, Table 1). There is no apparent systematic variation in strain with distance to the DAV shear zone, yet the distribution of different matrix microstructures, as described and discussed in sections 4.4. and 5.5., is different from west to east (Fig. 1).

**3 Methods**

Samples were cut perpendicular to the foliation and parallel to the stretching lineation. Thin sections of ~30 µm thickness were first polished mechanically and then chemo-mechanically in a colloidal silica-solution. For scanning electron microscopy (SEM) thin sections were coated with a thin layer (~5 nm) of carbon. Electron microscopic investigations were

performed on a Hitachi SU5000 with a field emission gun. Semi-quantitative chemical measurements by energy dispersive spectroscopy (EDS, AzTec, Oxford instruments) were acquired using an accelerating voltage of 20 kV and a working distance of 10 mm. Cathodoluminescence (CL) imaging using a Gatan MiniCL detector was performed at 5 kV and 10 mm working distance.

Crystallographic orientations were analysed using a HKL NordlysNano high-sensitivity Electron Backscatter Diffraction (EBSD) detector(Oxford Instruments). The EBSD signals were acquired using the AzTec analysis software (Oxford instruments). We used a sample holder pre-tilted at 70° with respect to the electron beam, an accelerating voltage of 20 kV and a working distance of 20 - 25 mm. The step size for automatic mapping was in the range of 1-2 µm, dependent of the required resolution, grain size and size of the area measured.

EBSD data were analysed with the MTEX toolbox for Matlab, developed by Ralf Hielscher (https://mtex-toolbox.github.io/; e.g. Bachmann et al., 2010). Small non-indexed pixels were filled during data smoothing by a half-quadratic filter (Bergmann et al., 2015). For grain reconstruction a thresholding value of 10° was used. For grain reconstruction, Dauphiné twin boundaries in quartz are neglected by merging grains along boundaries characterized by a misorientation angle of 60° and a [0001] rotation axis. Evaluating albite grain boundaries in full misorientation space (Krakow et al., 2017) revealed that

almost all twins correspond to the albite law and some to the pericline law. For grain reconstruction of albite, grains were merged along the corresponding twin boundaries. Evaluating mean grain orientations neglecting the twin orientations requires the use of a higher symmetry, which contains the symmetry element responsible for twinning, which is the point group 121 for albite and 622 for quartz. The mean orientation of the "higher symmetry" grain is the modal orientation of the "lower symmetry" grain. Using the higher symmetry yields the same grain reconstruction result as merging along twin

boundaries. Grain size analysis was by area normalization excluding border grains. The mean and median of the area distribution are given in histograms. The aspect ratio and the trend of the grain long axis were calculated from an area-equivalent best-fit ellipse. Pole figures were calculated either from the de-noised EBSD-data (scatter plots) or from orientation distribution functions (ODFs). ODFs were calculated from the grain mean orientation or from every pixel. For the calculation a "de la Vallée-Poussin" kernel was used (https://mtex-toolbox.github.io/; e.g. Bachmann et al., 2010). Kernel

width was estimated with the Kullback–Leibler crossvalidation function of MTEX (https://mtex-toolbox.github.io/; e.g. Bachmann et al., 2010). Pole figure densities (pfJ; L2-norm of the pole figures) and texture index (TI, L2-norm of the ODF) are used to characterize texture strength (Mainprice et al., 2015).

For qualitative comparison, we choose to display poles to the (100), (010) and (001) planes in upper hemisphere pole figures for albite. The internal misorientation within grains is dependent on the density of geometrically necessary dislocations and

thus is commonly used as a measure of crystal-plastic strain (e.g. Nicolas and Poirier, 1976; Poirier, 1985; Wheeler et al., 2009). To compare the intragranular misorientation between grains, we used the grain kernel average misorientation (gKAM) (Kilian, 2017), which can be computed in MTEX. The kernel average misorientation is the misorientation angle averaged over a certain kernel width for every measured point. We used a kernel size of 24 pixels (3rd order neighbours) and

ignored misorientation angles above 8°. The sum of these misorientation angles divided by the number of measurements in a grain is gKAM.

Compositions of feldspars and other major minerals were measured by a Cameca SX100 electron microprobe, using 15 kV voltage, 10 nA beam current and 1 µm spot size. The ZAF correction scheme provided by Cameca was used.

## 4 Results

The primary magmatic assemblage of the pegmatite comprises quartz, albite-rich plagioclase, K-feldspar and muscovite with accessory tourmaline, garnet, zircon, apatite and monazite. The foliation and stretching lineation are characterized by large fragmented magmatic tourmaline and feldspar, here referred to as porphyroclasts and alternating quartz-, albite- and mica-rich layers (Fig. 2a-d). The plane normal to the foliation and the stretching lineation are taken as the principal axes of the finite strain ellipsoid z and x, respectively, which are indicated in micrographs. The K-feldspar is Na-poor (<10%) and rarely shows perthitic exsolution. Plagioclase porphyroclasts have a narrow compositional range of Ab96-100. In few samples magmatic plagioclase with Ab95-86 is present and in these grains zoisite inclusions are common. Magmatic garnet is Mn-rich and low in Ca (on average Alm70Gro4Sp26). During metamorphism, garnet with a higher Ca- and lower Fe-component (Alm35Gro45Sp20) partly replaced magmatic grains (Fig. 3a). Epidote and Fe-bearing phengitic white-mica (2 wt% FeO) grew in the foliation plane of mylonitic pegmatites (Fig. 3b).The Fe-bearing phengite sometimes directly replaces magmatic white mica. The modal percentage of albite and K-feldspar varies in the different samples: albite comprises about 60-40 % and K-feldspar about 5-30%. The matrix layers comprise about 95% albite, independent on the ratio of K-feldspar to albite porphyroclasts, which varies from 8:1 to 1:9. This observation is consistent with whole-rock compositions with a marked variation of Na2O and K2O reported by Stöckhert (1987). Samples show homogenously low CaO (<1 wt.%) and FeO, MgO, MnO (< 1 wt.%) contents (Stöckhert,1987). The variations in the whole-rock composition were interpreted to be due to different compositions of the anatectic melt or mineral zoning in the pegmatite body and affected by external fluids (Stöckhert, 1987). In the following, we describe the specific feldspar porphyroclast and matrix microstructures.

### 4.1. Strain shadows

In all samples, polyphase aggregates of K-feldspar, albite, quartz and mica occur in prismatic strain shadows between tourmaline and feldspar fragments and surrounding feldspar porphyroclasts (Fig. 2b, Fig. 4a, b). Asymmetric strain shadows are characterized by different microstructures displayed in Figure 4a-c: polyphase aggregates of albite, K-feldspar and quartz characterize the upper-left and lower-right quadrants, whereas the lower-left and upper right quadrants contain almost monophase albite aggregates that alternate with quartz layers. This asymmetric strain shadow is indicating a sinistral sense of shear, with the polyphase aggregate representing extensional quadrants and the monophase aggregate compressional quadrants. The shape of the albite grains in the monophase layers is rather elliptical with a long axis parallel to the layer, whereas the shape of the albite and K-feldspar grains in the polyphase aggregates are irregular and rather isometric. Grain

sizes vary with long axes between 2 and 150 µm, the average of grain diameters is at 25µm (1σ = 19 µm) (Fig. 4h, i). The plagioclase composition uniformly ranges between Ab97-100. EBSD measurements of albite in strain shadows were analysed comparing single grain orientations with that of the host, comparing pole figures of scattered measurements as well as density plots recalculated from ODF (Fig. 4d-g). The EBSD data reveal no obvious orientation relationship between new grains within aggregates or a specific relationship between new grains and porphyroclasts, although some new grain orientations might correlate with that of the clast (compare to Fig. 4 f). Generally, the internal misorientation angles of the grains in aggregates with a typical diameter of 25-30µm is low with a maximum internal misorientation generally lower than 5°. The relative misorientation within the albite porphyroclast is lower than 10° (Fig. 4 c).

### 4.2. K-feldspar porphyroclasts

Single fractures in K-feldspar porphyroclasts are sealed by aggregates of K-feldspar, albite and quartz, representing prismatic strain shadows (Figs. 2b, 5c, d). Dispersed fluid inclusion trails at high angle to the stretching lineation are interpreted as healed microcracks (Figs. 5a-c). Areas comprising a high amount of healed microcracks are associated to undulous extinction, consistent with a bending of the crystal (Fig. 5a). This bending can be quantified by a change in misorientation angle of about 20° over a distance of 700 µm. Yet, in K-feldspar porphyroclasts that do not show dispersed healed microcracks, the internal misorientation angle within one grain is generally below 5° (Fig. 5d) over a grain size of several mm. The K-feldspar porphyroclast interface with new albite grains is cuspate due to protrusions of small albite grains into K-Feldspar over a length of a few tens of µm. Albite protrusions often have lobate grain boundariesat contact with K-feldspar porphyroclast (Fig. 6). The occurrence of this cuspate boundary is independent on position with respect to the stretching lineation or foliation (Fig. 5, 6). EBSD analysis reveals that there is no crystallographic relationship between K-feldspar porphyroclast and new albite grains and the misorientation angle to the porphyroclast is generally high (Fig. 5d, e). The new albite grains often contain numerous pores and inclusions of tiny (< 5 µm) apatite needles, at the vicinity of the K-feldspar porphyroclast interface (Fig. 6b, d). The apatite inclusions are in some places also present in the K-feldspar (Fig. 6b, arrows). Healed microcracks terminate at new albite grains, which therefore are interpreted to have formed after fracturing (arrows in Fig. 5c, d).

### 4.3. Albite porphyroclasts

Albite porphyroclasts are commonly twinned, bent and fragmented (Fig. 7, 8). The deformed fragments are surrounded by a fine-grained albite aggregate (grain diameters of 27 µm in average) with a very similar composition (Ab96-100) compared to the host albite, yet with a tendency to a somewhat higher albite component (< 1% higher Ab component). An irregular, patchy An-rich seam (up to An20) is commonly observed around new albite grains (Fig. 7f). New albite grains occur along fractures of deformed porphyroclasts that are oriented subparallel to the foliation, i.e., along sites of high strain and along boundaries parallel to the foliation (Fig. 7a-c, e). The new albite grains are typically not twinned, in contrast to fragments of the host (Fig. 7b, d). Larger host fragments have a relative high internal misorientation with angles typically of about 10°

along a profile length of 100 µm, ignoring twin domains (Fig. 8g, h). Grain kernel average misorientation (gKAM) values (0.4-0.7 °) for new grains are lower than for the porphyroclast or its fragments (0.7-1) (Fig. 8b). Low-angle boundaries are typically observed oriented at high angle to the stretching lineation, indicating that they represent healed cracks associated with a slight misorientation rather than indicating subgrains (Fig. 8a). Curved low-angle boundaries bounding subgrains were not observed. The orientation of the new grains scatters around the orientation of the host crystal (Fig. 8c, d). The misorientation angle distribution shows an excess of low and deficit of high misorientation angles for new grains compared to a random distribution (Fig. 8e particularly for correlated (neighbouring) measurements).

### 4.4. Monophase albite matrix alternating with quartz layers

In the fine-grained matrix, layers of almost purely albitic plagioclase Ab97-100 (i.e., similar or slightly more Ab-rich compared to plagioclase porphyroclasts) alternate with quartz-rich layers (Fig. 9; Appendix 1, Table 1). This mylonitic matrix is often deflected by albite porphyroclast and can also be deflected by albite aggregates replacing former porphyroclasts (sect. 4.3). The microstructure of the layers differs characteristically in their grain size and shape. Based on these two properties, we distinguish two endmembers of quartz-albite matrix microstructure:

Type A) The albite grains in the a few hundred µm wide layers are isometric (aspect ratio: 1 − 1.3) with grain diameters varying between 10 − 70 µm, in average of about 15 µm (Fig. 9, 10). The grains usually show no twinning and have a low internal misorientation of generally lower than 5°. The grain boundaries are irregular to smoothly curved (Fig. 11). Inclusions of apatite and domains with high porosity are common (Fig. 11). Grains show compositional zoning (arrows in Fig. 11), which is often only apparent in CL images and is therefore probably linked to changes in trace element contents. This zoning might be truncated by the growth of other grains, generally in the direction of their long axes (green arrow in Fig. 11b). A weak shape preferred orientation (SPO) parallel to the foliation can be deflected around the largest porphyroclasts (Fig. 10 a, b). The misorientation angle distribution (Fig. 10c) and polefigures (Fig. 10d) reveal a random texture. Associated quartz-layers are typically a few hundred µm wide and composed of coarse-grained aggregates (diameter of 100 − 1000 µm, Figs. 2c, d, 9a,b). Quartz in layers shows undulatory extinction, subgrains and sutured grain boundaries (Fig. 9a, b).

Type B) The albite grains in the layers typically a few tens of µm wide are elongate (aspect ratio: in average 2.3 and up to 9) and show a marked SPO. The average grain diameter is with 30 µm larger than in the type A microstructure (Fig. 9 c, d, Fig. 12). Similar to the type A microstructure, there is no apparent crystallographic preferred orientation (CPO) of albite and grains have a low internal misorientation (Fig. 12b, c). Some K-feldspar can be present as larger clasts (Fig. 12b) or as irregular flakes (Fig. 13a). The grain boundaries are mostly serrated but can vary to smoothly curved and even straight, then they are at low angle to the foliation (Fig. 13a, b). Straight segments can be parallel to the traces of (001) and (010) cleavage plane, representing energetically favoured boundaries (e.g. Tröger, 1982). The sutures are affected by intragranular cracks, indicated by trails of pores at high angle to the stretching lineation (arrows in Fig. 13a, b). Numerous tiny apatite needles occur in zones generally restricted to the centre of the grains but can be cut off by grain boundaries (Fig. 13c) or

microcracks. Rarely, grains with twins occur (Fig. 13e). Some grains show Ca-enriched zones and areas with a higher porosity (Fig. 13e, f). The porosities parallel to the short axes of grains, the elongate shape and the zoning indicate that grains grew at boundaries perpendicular to the stretching lineation. Generally, new albite grains do not show an orientation contrast observable by BSE imaging, in contrast to twinned remnants of porphyroclasts (Fig. 7).

Samples that show the type B matrix microstructure are interpreted to correlate to a higher strain because of the high aspect ratio (up to 9) and narrow spacing of the alternating quartz-albite layers (tens of µm) compared to the type A microstructure with a larger spacing of the layers of a few hundreds of µm and a lower aspect ratio (Fig. 2a, b, 9c, d, 12a, b, d). These microfabrics correlate with the observation from the field, where samples showing a type B microstructure are characterized by a more narrow spacing of the foliation planes, lower abundance and diameter of porphyroclast as well as a more

pronounced stretching lineation.

## 5. Discussion

In the following, we discuss the deformation and replacement mechanisms of feldspar leading to the mylonitic fabric and implications on the Alpine deformation.

### 5.1. Interface-coupled K-feldspar replacement by albite

The replacement of K-feldspar by albite is a widely observed reaction in deforming granitoids at low temperatures. Different types of replacements of K-feldspar by albite have been discussed, which can be divided into two groups:

1)  Neocrystallisation or heterogeneous nucleation during metamorphic reactions and/or precipitation from the pore fluid, produces distinct albite grains without any crystallographic relationship to the replaced K-feldspar. The replacements often appear in strings and patches inside the host grain and may be related to fractures (e.g. Fitz

20        Gerald and Stünitz, 1993; Stünitz, 1998; Menegon et al., 2013).

2)  Interface-coupled dissolution of K-feldspar (or plagioclase) and spatially coupled precipitation of albite leads to a strong structural coherence across the reaction interface, i.e. of the primary mineral on the orientationof secondary mineral as found in rocks (Plümper and Putnis, 2009; Putnis, 2009) and experiments (Norberg et al., 2011; Hövelmann et al., 2010). These studies reported that the new albitemight be porous and might contain secondary

25        inclusions. Norberg et al. (2011) observed associated microcracking in the K-feldspar adjacent to the reaction front. The dissolution of K-feldspar has been found to be orientation-dependent (Norberg et al., 2011). Cuspate protrusions of albite growing into the host K-feldspar have been found to be characteristic of such interface-coupled replacements (Norberg et al., 2011).

The cuspate boundaries between new grains of albite and K-feldspar porphyroclasts are interpreted to indicate interface-

coupled replacement (Fig. 6), supported by the porosity and apatite inclusions in albite replacing K-feldspar (Fig. 6b, d). The K-feldspar replacement is independent fromthe orientation of the boundary to the foliation and stretching lineation and is

therefore interpreted to be not directly related to the strain field during deformation, not excluding some influence of higher strain along the boundary compared to within the crystal. The driving force is interpreted to be the difference in solubility between albite and K-feldspar at the given greenschist-facies metamorphic conditions (Putnis, 2009). Whereas locally albite grew to replace K-feldspar, K must have been transported through the pore fluid, either to form metamorphic phengitic mica in the foliation plane or to precipitate K-feldspar in polyphase strain shadows (see chapter 5.4.).

The albite in grains along intragranular fractures within K-feldspar (Fig. 2b, 5d) might represent neocrystallization of albite replacing K-feldspar. As the intragranular fractures perpendicular to the stretching lination (x), these sealed fractures might represent prismatic strain shadows, i.e. albite may have precipitated from the pore fluid and not necessarily replacing K-feldspar.

## 5.2. Strain-induced replacement of albite

Albite porphyroclasts are mostly deformed at boundaries parallel to the foliation, commonly associated with bent mechanical twins (Fig. 7a-e). Bent twins are corresponding to an undulous extinction indicating a continuous internal misorientation, which is usually taken to result from the presence of geometrically necessary dislocations (e.g. Nicolas and Poirier, 1976; Poirier, 1985; Wheeler et al., 2009), though some microcracking might also be involved, as pointed out by Tullis and Yund (1987). For albite this continuous internal misorientation is not associated to distributed healed microfractures, as observed for K-feldspar (compare Figs. 5 and 7), which is taken to indicate a relative higher importance of dislocation glide for the deformation of albite compared to K-feldspar. Strained areas (fractures, porphyroclast boundaries) parallel to the foliation are replaced by new, strain free grains that are generally not twinned (Fig. 7b, d, e-f). The composition of the new albite grains can be the same as that of the replaced porphyroclast, though it can also show slightly higher Na-content, as already reported by Stöckhert (1987). Plagioclase with Ca-richer compositions occurs locally as thin rims at grain boundaries of new grains with no systematic occurrence (Figs. 7f). Because new strain-free albite grains replace twinned porphyroclasts with internal misorientation at boundaries parallel to the foliation (Figs. 7e, f, 8a, b) and along intragranular microcracks parallel to the foliation (Fig. 7a, b), the replacement is interpreted to be driven by the reduction in stored strain energy. Strain-induced grain boundary migration and formation of growth rims following dislocation glide and microfracturing is consistent with an orientation scatter around the orientation of the host porphyroclast (Fig. 8d, f). The similar composition of the new albite compared to the replaced porphyroclasts, with a tendency of a slightly increased Na-content, suggests a contribution of chemical driving forces although strain-induced grain boundary migration is dominating (Stöckhert, 1982; Stünitz, 1998). Whereas dislocation glide is indicated by bent and twinned porphyroclasts and fragments with internal misorientation, we did not observe subgrains in deformed fragments (Fig. 8a-c), even not in strongly bent porphyroclasts (Fig. 7 a, b, e). This is consistent with the general finding that albite shows only little evidence of dislocation climb with dynamic recovery and recrystallization at $T \leq 550°C$ (e.g. Tullis, 1983; Fitz Gerald and Stünitz, 1993; Kruse and Stünitz, 2001).

We suggest that albite porphyroclasts deform in the regime of low-temperature plasticity, where dislocation climb is ineffective and where dislocation glide leads to strain hardening and microfracturing. Additionally, dislocations can be

induced by microfracturing (Tullis and Yund, 1987). Subsequently, grains grow by strain-induced grain boundary migration, where crystalline volume with higher strain energy is dissolved and strain-free crystalline volumes precipitated, as suggested by Tullis and Yund (1987). Growth can additionally be by precipitation along areas of lower solubility leading to growth rims with a shape-preferred orientation with long axes in the foliation plane. Strain-induced grain boundary migration might

be enhanced by chemical disequilibrium (Stöckhert, 1982; Stünitz, 1998). Our interpretation of strain-induced replacement is similar to the "micro-crush zones" described by Tullis and Yund(1992) associated with undulous extinction, shear bands and grain size reduction that are usually associated to crystal plastic mechanisms (Mclaren and Pryer, 2001; Stünitz et al., 2003). This process is also similar to the "neocrystallization" in the sense of Fitz Gerald and Stünitz (1993) and Menegon et al. (2013), which may or may not cause some compositional variations, dependent on the local fluid present. We, however,

suggest the term "strain-induced replacement" for the nucleation by low-temperature plasticity (associated dislocation glide and microfracturing) and growth to stress firstly the importance of dislocation glide (as opposed to the term "micro-crush zones" that stresses brittle mechanisms) and secondly to stress the difference to precipitation with nucleation of new phases from the pore fluid in strain shadows (as opposed to the term "neocrystallization").

## 5.3. Intragranular fracturing of K-feldspar

K-feldspar porphyroclasts deformed dominantly by intragranular fracturing with no comparable strain-induced replacement associated with a grain size reduction as observed for albite. A preferred crystallographic relation of the intragranular fractures was not detected given their orientation at high angle to the stretching lineation independent on crystallographic orientation (Fig. 5), ruling out a major influence of cleavage fractures. Single fractures are sealed with albite, K-feldspar and quartz representing prismatic strain shadows, or they are healed (Fig. 5). Bending of K-feldspar porphyroclast associated to

undulous extinction is restricted to sites of distributed microcracking (Fig. 5a).The high amount of healed microcracks at high angle to the stretching lineation indicates that here, microfracturing was dominating over dislocation glide in the bending of the crystal, as opposed to albite porphyroclasts (compare Fig. 5a and Fig. 7 a, e). In contrast to plagioclase, where mechanical twinning is commonly observed, mechanical twinning of K-feldspar is hindered by the Si/Al-ordering (Tullis, 1983).

Reaction weakening of K-feldspar, commonly in association with myrmekites,is known to play a major role during grain size reduction and ductile deformation at many metamorphic conditions (e.g. Simpson and Wintsch, 1989; Tsurumi et al., 2003; Ree et al., 2005; Menegon et al., 2006, 2008, 2013; Abart et al., 2014). In the mylonitic pegmatites described here, myrmekitic replacements are very rare and apart from the cuspate replacements, K-feldspar porphyroclasts are well preserved. Thus, reaction weakening of K-feldspar is rheologically not relevant for the mylonitic deformation described here.

## 5.4. Precipitation in strain shadows

The occurrence of polyphase aggregates of K-feldspar, albite, mica and quartz in strain shadows between fragments of tourmaline and feldspar porphyroclasts, as well as surrounding porphyroclast (Figs. 2a, b, 4a-c, 5c, d), with random texture

and absent systematic crystallographic relationships indicate that these aggregates represent precipitates of a saturated pore fluid during deformation by dissolution-precipitation creep (e.g. Groshong, 1988; Passchier and Trouw, 2005; Wassmann and Stöckhert, 2013). That few grain orientations in the strain shadow correlate with that of the host crystal is interpreted to be due to the presence of fragments of the host crystal (Fig. 4d). The precipitation of K-feldspar, quartz and albite in strain shadows and albite growth rims is restricted to boundaries at high angle to the stretching lineation (x), i.e. controlled by strain, yet an additional chemical driving force is clearly not ruled out but rather probable. Additionaly, some replacement might also occur in strain shadows. The sites of dissolution are much more difficult to identify, as the material has been removed.

A polyphase matrix of K-feldspar, albite quartz and mica in mylonitic granitoids is often attributed to fine-grained reaction products (e.g. Stünitz and Fitz Gerald, 1993; Rosenberg and Stünitz, 2003; Kilian et al., 2011). Other authors, suggest polyphase matrix to develop by mechanical phase mixing in mylonites at highest strain (Fliervoet, 1995). In the mylonitic pegmatites reported here, however, no indication of active "phase mixing" is observed and we attribute the occurrence of a polyphase matrix to precipitation. Also, the characteristic mylonitic microstructure of the pegmatites is associated not with a polyphase matrix but with the monophase quartz and albite layers.

## 5.5. Formation of monophase albite layers

Based on our observations that albite in layers shows the same characteristics as albite grains replacing albite porphyroclasts (missing subgrains and internal misorientations, apatite inclusions, weak chemical zoning and porosity, and remnants of twinned porphyroclast fragments), we suggest, that the strain-induced replacement of albite is the most important process of grain size reduction to form the monophase albite layers (Fig. 14). Additionally, part of the albite in the mylonitic matrix stems from the replacement of K-feldspar (sects. 4.2 and 5.1), as suggested by K-feldspar-relicts (Figs. 12b; 13a). Because the matrix layers comprise about 95% albite, independent on the ratio of K-feldspar to albite porphyroclasts, we suggest that albite is taking up a higher amount of strain as compared to K-feldspar. Dislocation creep of albite is ruled out as main process to form the fine-grained almost monophase albite layers, given a missing systematic CPO as well as missing evidence of effective dislocation climb. Also, precipitation from the pore fluid as dominating process can be ruled out, given the monophase composition of the layers, in contrast to polyphase aggregates in strain shadows. Cataclasis would suggest a higher amount of twinned and deformed fragments. Instead, only very rarely twinned grains are observed (Fig. 13e).

We suggest that after grain size reduction, the fine-grained albite matrix was undergoing a mixture of dissolution-precipitation processes, microcracking and sliding of grains, commonly referred to as granular flow (e.g. Behrmann and Mainprice, 1987; Stünitz and Fitz Gerald, 1993; Jiang et al., 2000). Sliding might have occurred along straight boundaries weakly inclined to the foliation (Fig. 13b). Microcracking is indicated by the fractures at high angles to the foliation (Fig. 13a, b, e). The weak zoning of grains (Figs. 11 and 13) suggests the involvement of dissolution-precipitation. Granular flow would also cause weakening of a domainal CPO resulting from the replacement of albite porphyroclasts (e.g. Jiang et al., 2000; Hildyard et al., 2011).

Quartz layers of coarse recrystallized grains systematically correlate with albite layers of small isometric grains in the type A matrix microstructure (sect. 4.4; Figs. 9a, b; 10; 11). In contrast, narrow quartz layers with fine-grained quartz aggregates and marked CPO are correlated with elongate coarser albite in the type B matrix microstructure (Sect. 4.4; Figs. 9c, d; 12; 13). The elongate shape of albite and zones of high porosity at boundaries at high angle to the stretching lineation in the type B matrix microstructure indicates growth by precipitation, resulting in a shape-preferred orientation (Fig. 13e, f). The microstructure correlates with the overall strain of the mylonitic matrix (Fig. 14). In samples with higher overall strain, albite grains in the layers are coarser and more elongate and quartz aggregates are finer-grained and have a marked CPO.

## 5.6. Implications for rock rheology and deformation history

The prismatic strain shadows of polyphase material between fragmented tourmaline and feldspar as well as strain shadows surrounding porphyroclasts indicate that dissolution-precipitation creep did play a role in the rheology of the rocks. Yet, other indicators of dissolution-precipitation creep, as for example evidence of dissolved feldspar porphyroclasts along boundaries parallel to the foliation, i.e. strain caps, are remarkably low. The monophase alternating quartz and albite layers are the main characteristic of the mylonites. The dominating process of grain size reduction of feldspar is interpreted to be the strain-induced replacement of albite with associated dislocation glide and fracturing (Fig. 14). Subsequent growth by strain-induced grain boundary migration and formation of growth rims by precipitation resulted in a SPO (Fig. 14), which took place probably simultaneously together with granular flow and dislocation creep of quartz. As such, for considering the rock's rheology, these different deformation mechanisms and associated processes have to be taken into account, where the relative contributions additionally vary with time. In contrast, dissolution-precipitation creep with dissolution along boundaries parallel to the foliation and precipitation with nucleation of new phases in strain shadows is interpreted to play an only subordinate role for the formation of the mylonitic alternating quartz-albite layers, although precipitation of albite forming elongate grains with SPO in monophase aggregates is important. Furthermore, a major role of dislocation creep of feldspar (i.e., deformation by dislocation glide with simultaneous dynamic recovery) on the formation of the mylonitic microstructure as may be suggested by monophase layers of fine-grained feldspar aggregates, is not supported by any further microstructural observation (e.g. systematic missing of LAGBs) and therefore interpreted to be rheologically not relevant during deformation.

The observation of newly precipitated grains from the pore fluid between tourmaline and K-feldspar fractures at strain shadows as well as the microcracks in K-feldspar that are cut off by the wedge shaped albites replacing K-feldspar (Figs. 2a, b; 3c, d) indicate growth of grains after fracturing and during ongoing deformation. The observation of the deflected mylonitic foliation around former porphyroclasts, which are now replaced by new grains (Fig. 2c, d; 7c), indicates that the new grains grew after or during the formation of the mylonitic layers, but not before. Thus, strain-induced replacement of albite must have played an important role during an early stage of deformation and was ongoing during granular flow. Thus, a specific sequence of different deformation episodes at markedly different metamorphic stages, as had been discussed, is not apparent (Stöckhert, 1987; Mancktelow et al., 2001). The indication of the type A matrix microstructure being

dominating in the eastern area and the type B matrix microstructure representing in comparison higher strain in the western area (Fig. 1) might be correlated with higher metamorphic temperature conditions in the western area, as the amount of uplift and erosion since the intrusion of magmatic bodies at 30 Ma is increasing from about 10 km in an eastern area of the Rieserferner to about 15 to 25 km in the Rensen area in the west (Trepmann et al., 2004).

**6. Conclusions**

The mylonitic pegmatites record the deformation behaviour of feldspar at greenschist facies conditions. Based on our observations and discussions we draw the following conclusions:

1. K-feldspar porphyroclasts deformed dominantly by fracturing and only subordinate dislocation glide, without major grain size reduction. Healed or sealed intragranular fractures in large porphyroclasts at high angle to the stretching
lineation are the dominating deformation microstructures of K-feldspar (Fig. 5).

2. Interface-coupled replacement of K-feldspar by albite is mainly driven by chemical disequilibrium as indicated by the cuspate albite-K-feldspar phase boundaries independent on the orientation of the boundary to the foliation or the stretching lineation (Fig. 6).

3. Grain size reduction of albite porphyroclasts is by combined fracturing and dislocation glide, i.e. low-temperature
plasticity. Dislocation glide is indicated by bent and twinned remnant host albites with internal misorientation. Evidence of significant amount of dislocation climb allowing effective dislocation creep with dynamic recovery of feldspar is systematically missing (no subgrains, negligible internal misorientation of new grains; Figs. 10c, d, 12c, e).

4. Subsequent strain-induced grain boundary migration and formation of growth rims produced aggregates of strain-
free albite grains with SPO at porphyroclast boundaries parallel to the foliation (Figs. 7c-f, 8). The observed tendency of slightly enriched Na-content (decrease of Ca-content) of the new albite grains compared to albite porphyroclasts is in agreement with an additional, though subordinate driving force for grain boundary migration by chemical disequilibrium (Stöckhert, 1982).

5. Monophase quartz layers formed dominantly by dislocation creep (dislocation glide, dynamic recovery and
recrystallization) of quartz, as indicated by sutured grain boundaries, CPO, subgrains and undulatory extinction (Figs. 2c, 9, 12a, d). Some influence of dissolution-precipitation creep cannot be excluded; though microstructural evidence has not been observed.

6. Granular flow of the new albite grains with overgrowth of albite forming a SPO together with quartz dislocation creep is interpreted to result in the alternating monophase albite-quartz layers (Figs. 9-13).
For considering the rock's rheology, these different deformation mechanisms and associated processes have to be taken into account, where the relative contributions additionally vary with time.

## Acknowledgements

This study was funded by the Deutsche Forschungsgemeinschaft (DFG grant TR534-4-1).The reviews by Rüdiger Kilian and an anonymous reviewer are gratefully acknowleged, as is Bernhard Stöckhert  for discussions. Namvar Jahanmehr and Michael Herrmann are acknowledged for preparation of thin sections, as are Patrick Eschenbacher and Dominique Mackensen for helping with field work.

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

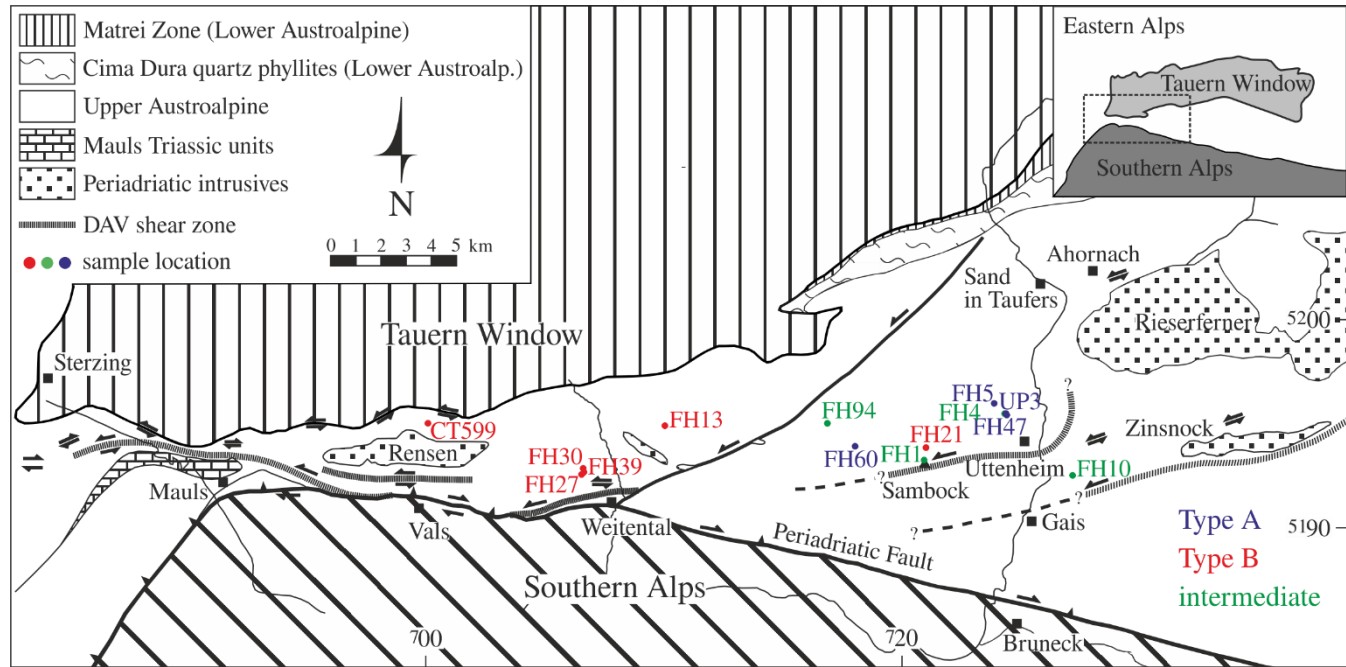

**Figure 1: Geologic map of the study area (modified after Mancktelow et al., 2001). The sample numbers are colored according to the type of albite-quartz matrix (see text and Appendix 1, Table 1)**

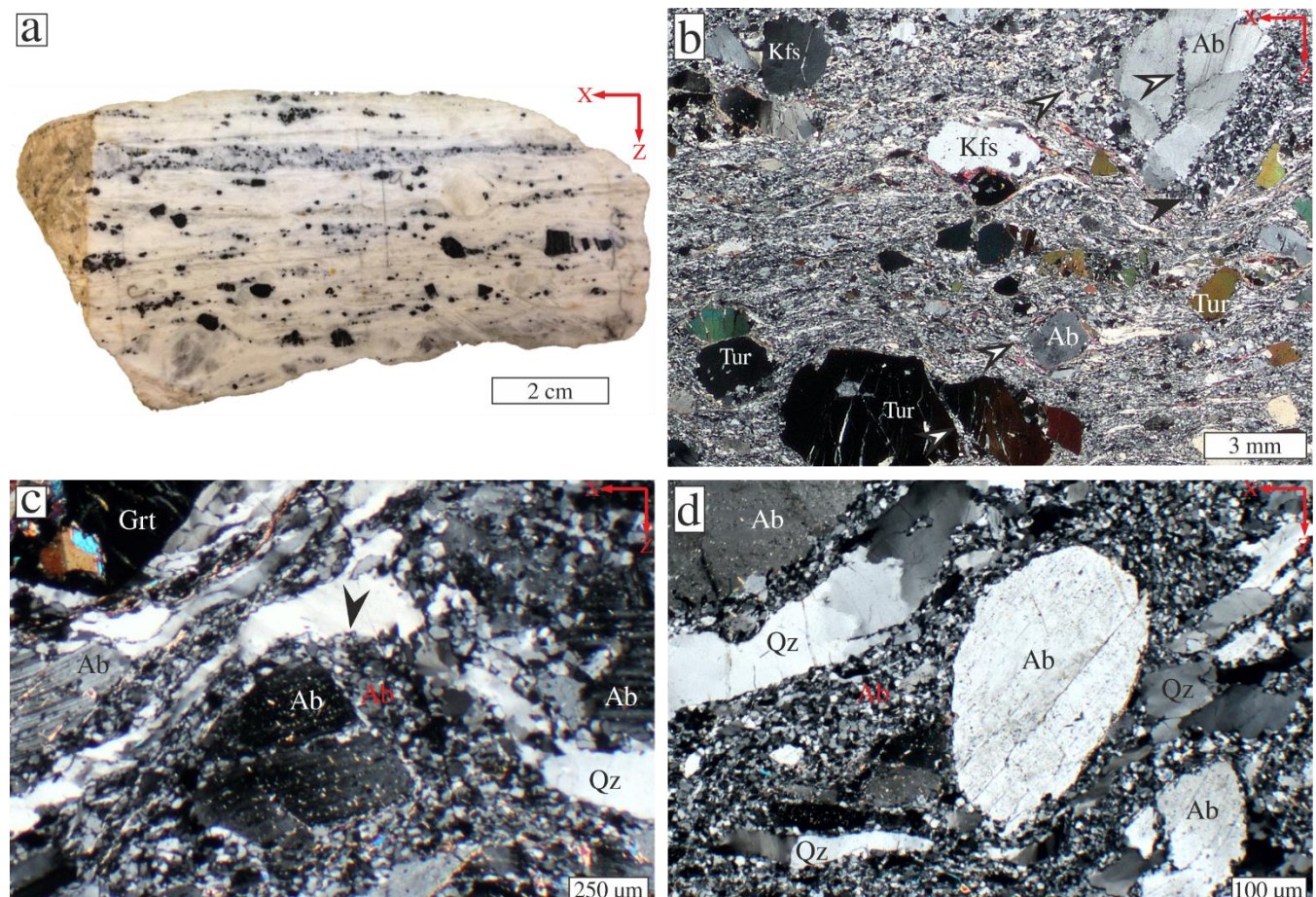

**Figure 2: Photograph of polished surface (a) and thin section micrograph taken with crossed polarizers (b) of sample CT599. K-feldspar (Kfs), albite (Pl) and tourmaline (Tur) porphyroclasts are embedded in a fine-grained matrix. Elongate fractured tourmaline crystals are oriented with their long axes parallel to the stretching lineation (x). Fractures are commonly oriented at low angle to the shortening direction (z). Arrows point to strain shadows surrounding porphyroclast and prismatic strain shadows between fragments of tourmaline and feldspar. Black arrow points to mylonitic foliation flowing around strain shadow. (c, d) Polarized light micrographs (crossed polarizers, sample FH5b) showing mylonitic foliation defined by quartz layers (Qz) flowing around garnet (Grt) and albite porphyroclasts (Pl), which are partly disintegrated into a fine-grained albite matrix.**

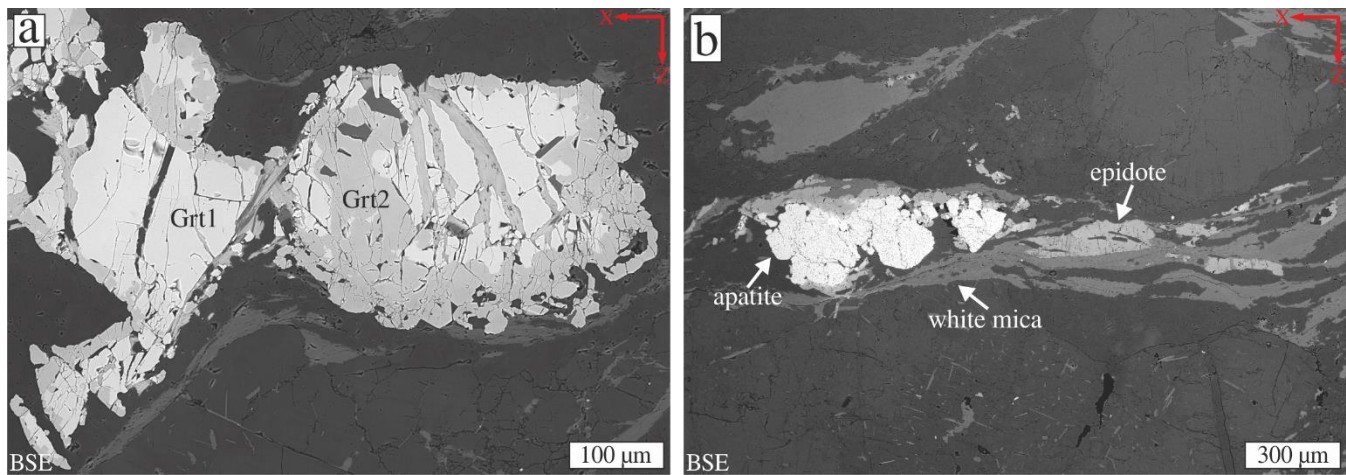

**Figure 3: BSE images from sample FH27 showing the typical accessory mineral assemblage in the deformed pegmatites: (a) Ca-rich garnet (Grt2) replacing magmatic Fe-rich garnet (Grt1). (b) Epidote and white mica aligned in the foliation with apatite porphyroclasts.**

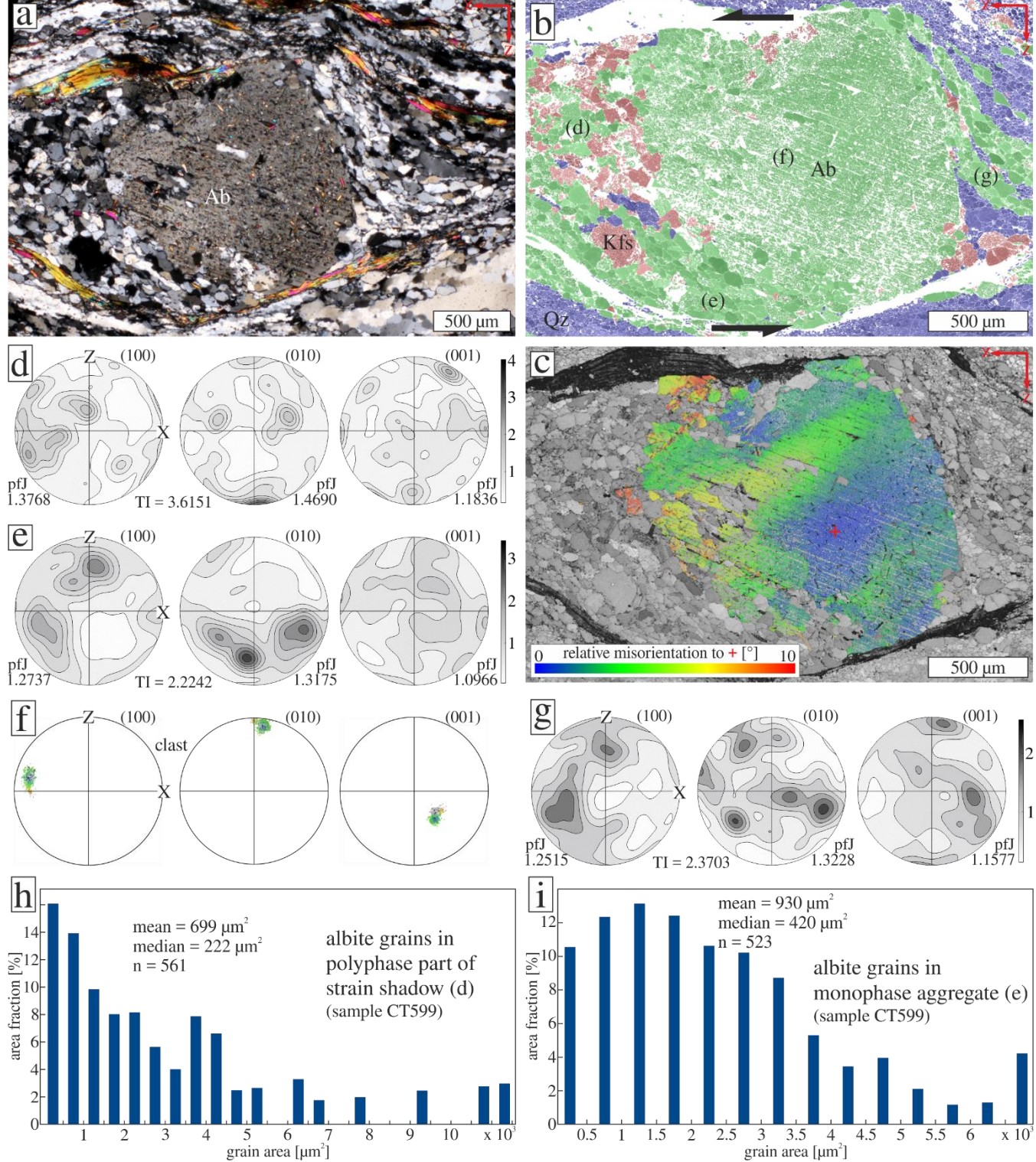

**Figure 4: (a) Asymmetric strain shadow around albite porphyroclast in sample CT599 in thin section micrograph with crossed polarizers. (b) EBSD-phase map of the same area (quartz: blue, albite: green, K-feldspar: red) and (c) EBSD-relative misorientation map (0-10°) of the albite porphyroclast. Polyphase aggregates occur mostly in the upper left and lower right of the clast. In the other quadrants monophase albite dominates. Pole figures show the orientation of albite in the strain shadow in the upper left quadrant (d), lower left quadrant (e), albite porphyroclast (f) and in the upper right quadrant (g). Grain area distribution histograms of albite in polyphase aggregates strain shadow (h) and in monophase albite aggregates (i).**

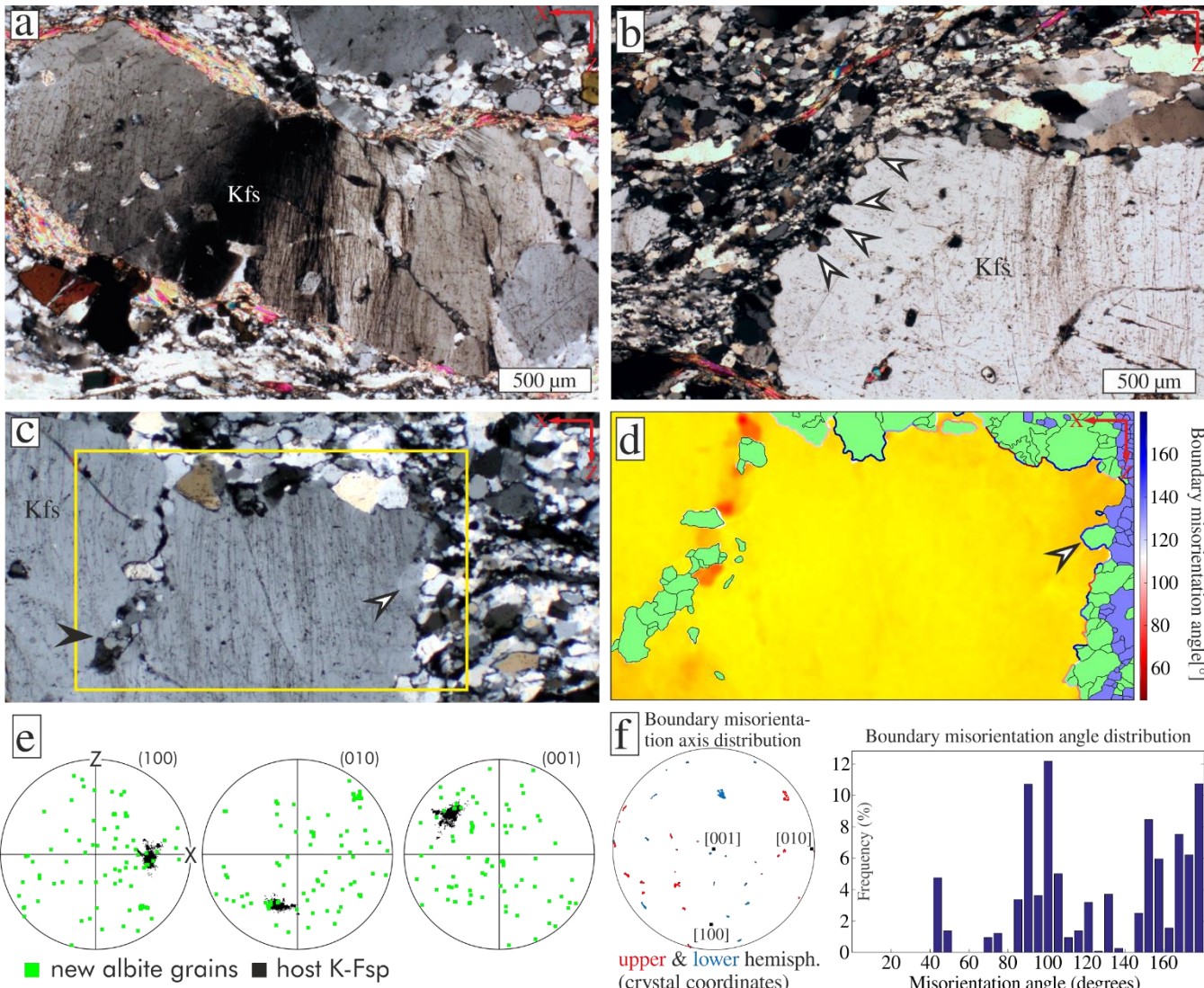

**Figure 5: K-feldspar deformation microstructures (sample CT599). (a, b) Polarized light micrographs (crossed polarizers) of bent K-feldspar porphyroclast with healed microcracks parallel to the shortening direction and cuspate grain boundaries (white arrows in (b)). (c) Polarized light micrograph (crossed polarizers) showing K-feldspar with fractures filled with albite (black arrow) and cuspate phase boundaries. Healed microcracks are cut-off by newly formed albite (white arrow). Yellow rectangle shows area of EBSD map in (d). (d) EBSD map, where boundaries between replacing albite grains (green) and the K-feldspar-host (yellow) are colored after their misorientation angle. Quartz in the matrix is blue in color. (e) Polefigures of poles to (100), (010) and (001)**

planes of new albite grains green) and K-feldspar porphyroclast (black). **(f)** Misorientation angle and axis distribution for the boundaries colored in **(d)**.

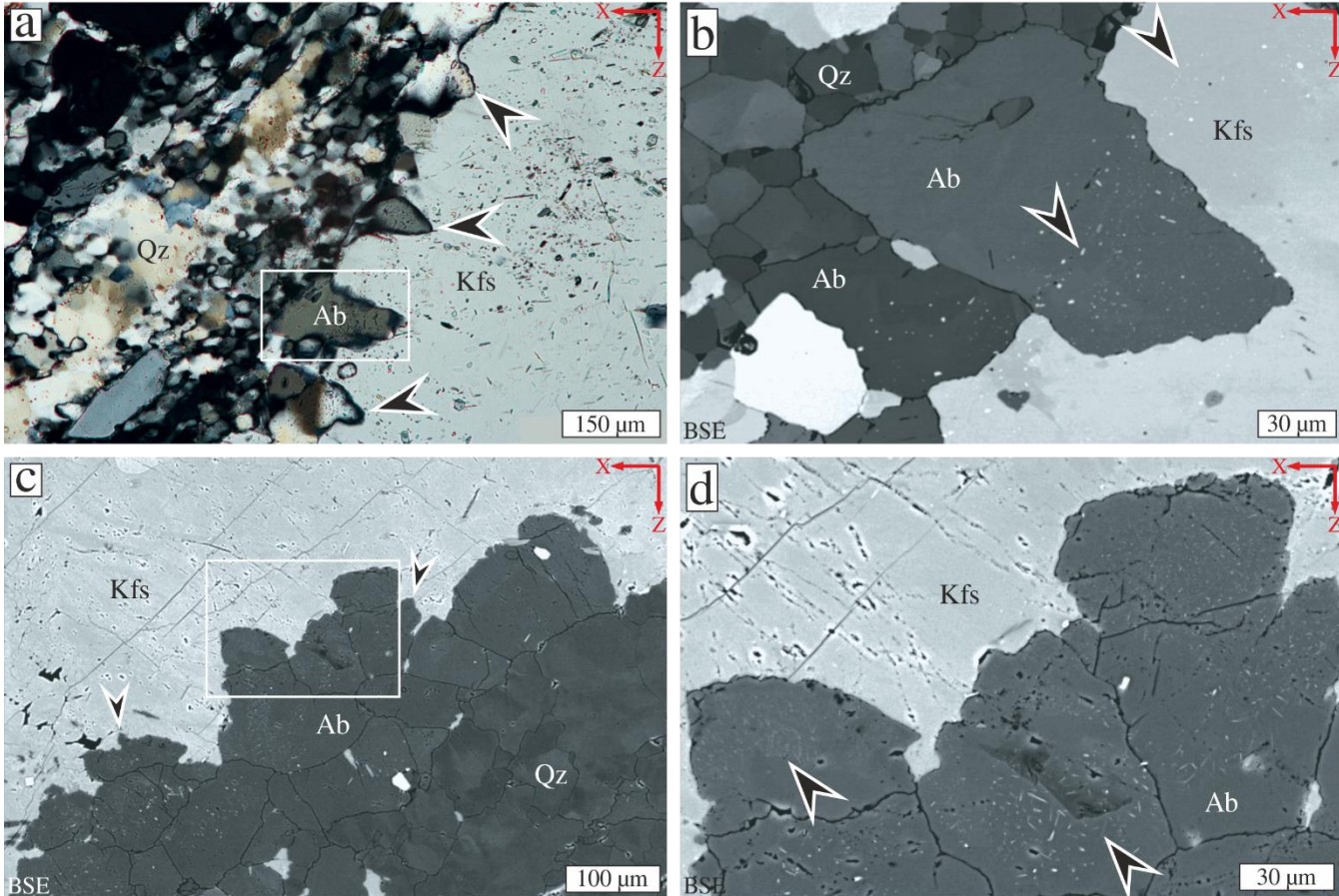

Figure 6: **(a)** Photomicrograph of cuspate interface between K-feldspar clast and new albite grains (sample CT599). **(b)** Close-up BSE-image of location indicated by white box in (a). Note the tiny apatite inclusions in the albite (determined by EDS, arrows). **(c)** BSE-image of cuspate interface between albite replacing K-feldspar in sample FH14. The arrows point to protrusions. **(d)** Close-up BSE-image of white box in (c) showing the numerous tiny (< 5 μm) apatite inclusions in the replacing albite.

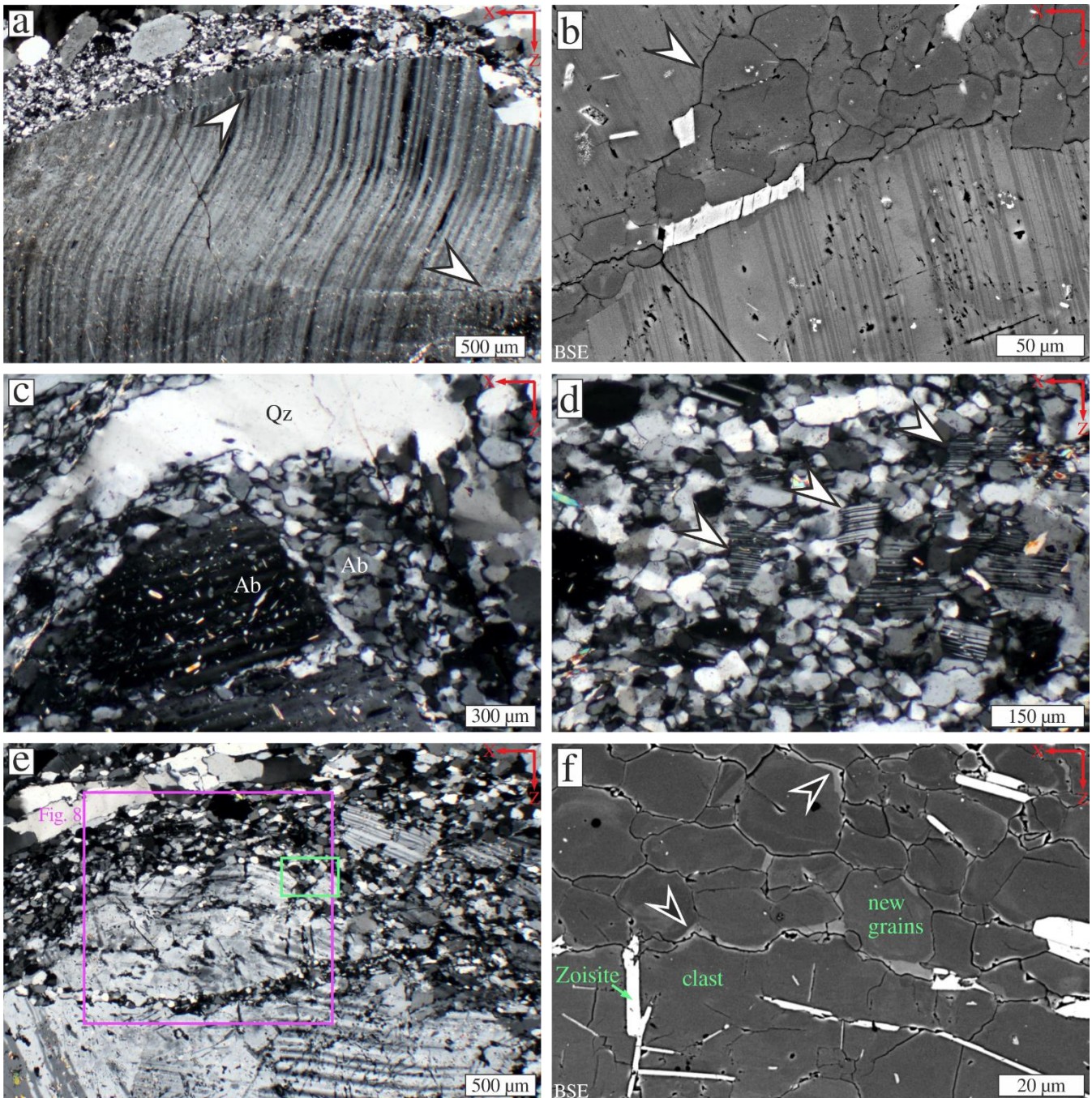

**Figure 7: (a) Polarized light micrograph (crossed polarizers) and (b) BSE image of bent and kinked twins in albite porphyroclast, sample FH5. New albite grains occur along fractures parallel to kink band boundaries (arrows). (c) Polarized light micrograph (crossed polarizers), showing fragmented albite porphyroclast partly replaced by new grains and surrounded by quartz layer, sample FH5. (d) Polarized light micrograph (crossed polarizers) showing twinned albite fragments surrounded by fine-grained albite matrix, sample FH5. (e) Polarized light micrograph (crossed polarizers) of twinned and fractured albite porphyroclast, sample UP3. Green box indicates area of BSE image in (f), violet box indicates EBSD map shown in Fig 8. (f) BSE image showing**

new albite grains adjacent to the albite porphyroclast. New grains often have an outer rim of less albitic plagioclase, up to $An_{20}$ (arrows). The bright phase are zoisite needles.

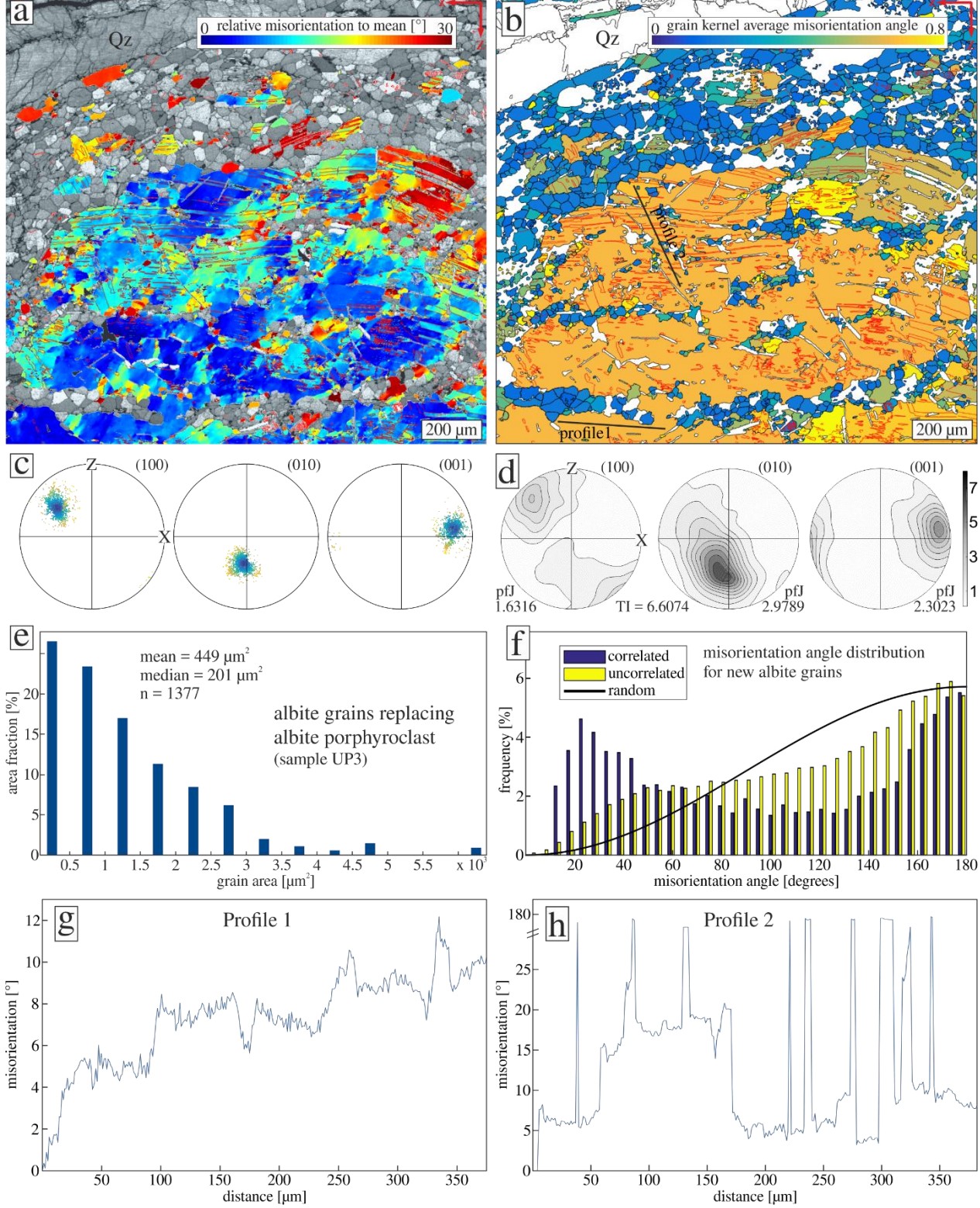

**Figure 8: (a, b) EBSD-relative misorientation map for albite relative to the mean orientation of the porphyroclast (a) and grain kernel average misorientation (gKAM)-map, showing a lower inferred dislocation density for new grains (b) of the area in the violet box in Fig. 7b. Red lines in are albite and pericline twin boundaries. (c, d) Corresponding pole figures colour coded corresponding to EBSD-relative misorientation map (c) and scatter plot (color-coded corresponding to (a)), where only grains smaller than 100 µm and free of visible twins were used. (e) Grain area distribution for new grains smaller than 100 µm. (f) Misorientation angle distribution for adjacent and random pairs of new albite grains. (g, h) Misorientation profiles (relative to origin) along lines shown in (b).**

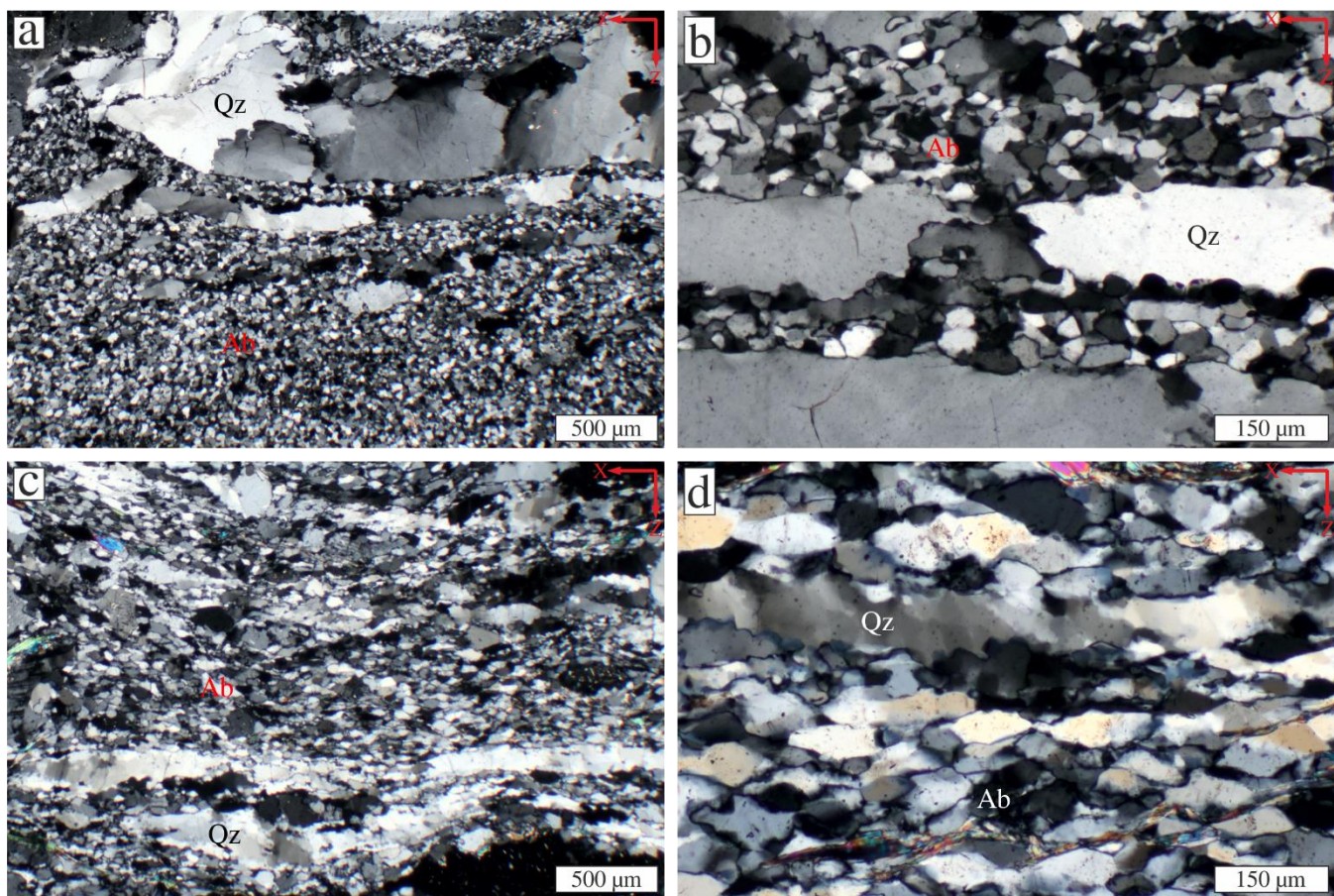

**Figure 9: Polarized light micrographs taken with crossed polarizers showing the two types of quartz-albite matrix microstructure. (a, b) Type A matrix is characterized by coarse quartz layers and albite layers with isometric small grains, sample FH5. (c, d) Type B matrix is characterized by albite layers characterized by coarser and elongate grains parallel to the foliation, and fine-grained, dynamically recrystallized quartz layers, samples CT599.**

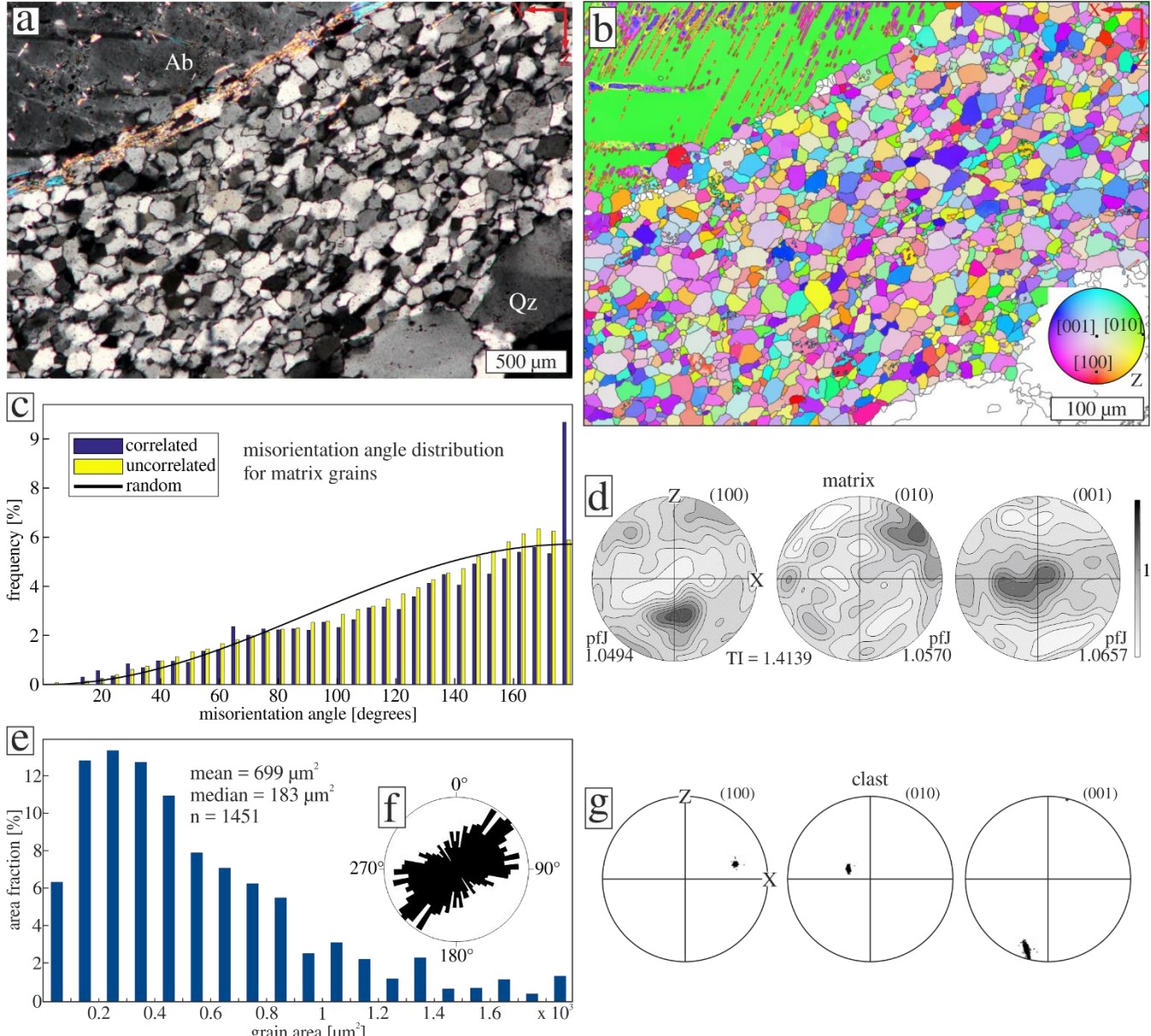

**Figure 10: EBSD-analysis of type A albite matrix in sample FH5. (a) Photomicrograph of the analysed area. (b) EBSD-map with inverse pole figure colouring (see lower inset). Twin boundaries in the porphyroclast are shown as red lines. (c) Misorientation angle distribution showing an essentially random distribution of neighbouring or random grain pairs. (d) Contoured pole figures showing orientation distribution of albite matrix grains. (e) Grain area distribution of the measured matrix grains. (f) Rose diagram showing the orientation of the long axis of grains. (g) Pole figures showing the orientation of the albite porphyroclast from (a, b).**

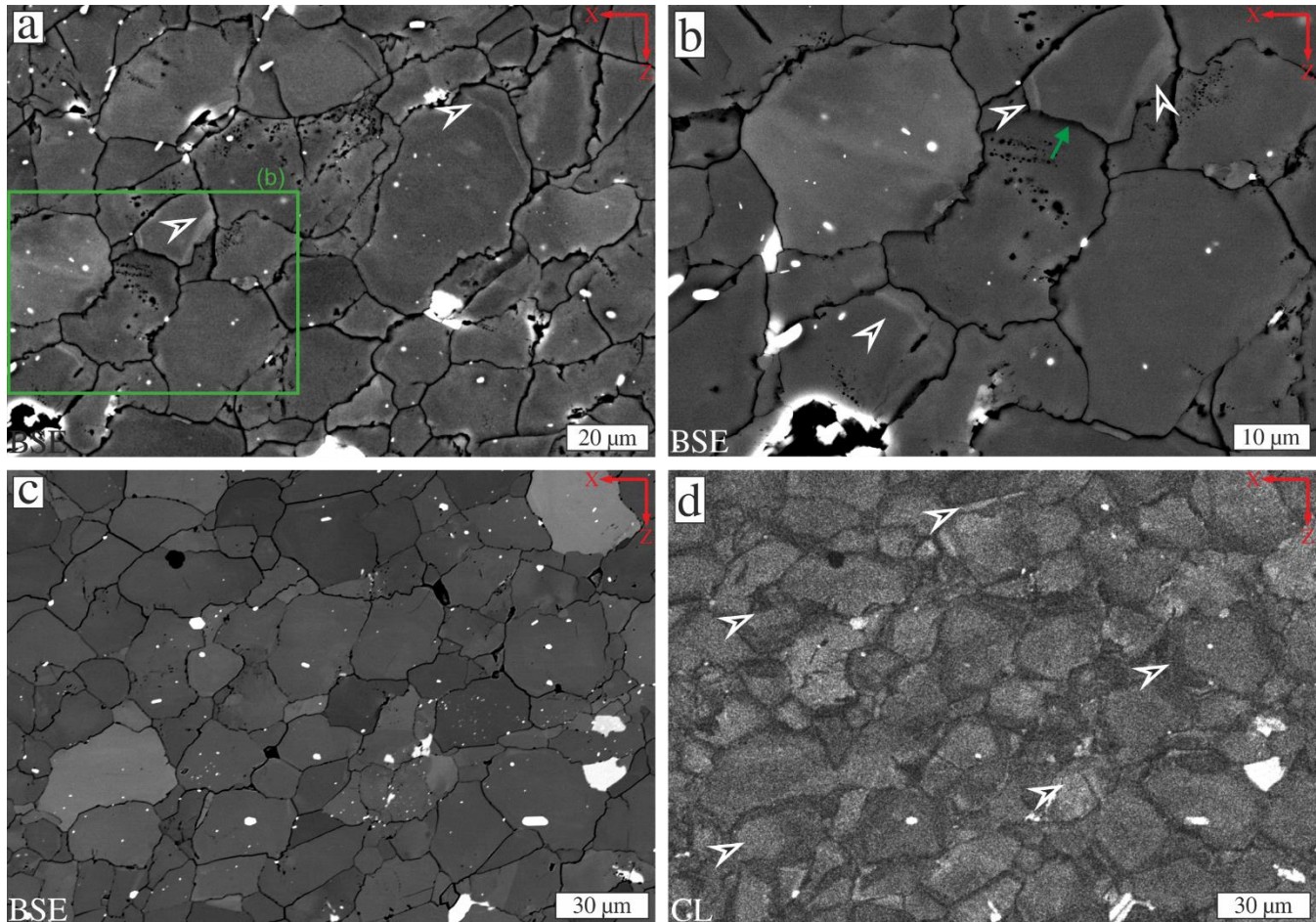

**Figure 11: Type A matrix, sample FH5. (a, b) BSE images show albite matrix with irregular grain boundaries, porosity and weak zonation (black arrows). In (b), the zonation of the grain in the upper right is truncated, possibly by growth of the grain below (green arrow). (c) BSE image with grey shades representing both, orientation and compositional contrast (bright phase is apatite) and (d) corresponding CL image showing zonations, not visible in the BSE image (arrows). Grain boundaries in the CL image are also associated to darker grey shades.**

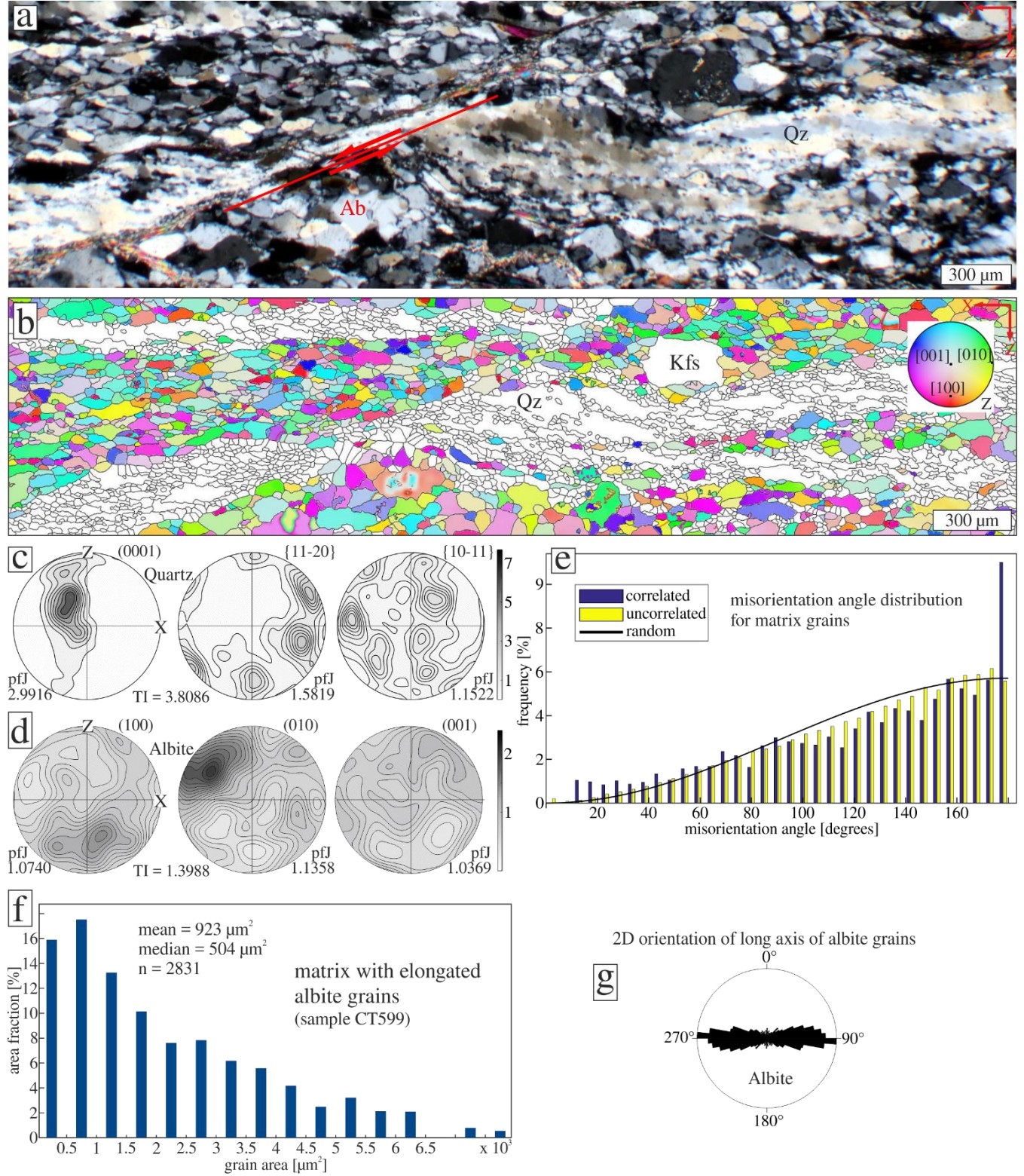

**Figure 12: (a) Photomicrograph of the type B matrix with coarse monophase albite and fine-grained quartz layers in sample CT599. Quartz and albite layers are offset at the shear band marked red. (b) EBSD-map of the area from (a) with an IPF colour code (Z-axis). Only albite is coloured after the IPF-colour-code. (c) Pole figures for albite grains from (b). (d) Pole figures for quartz grains from (b). (e) Misorientation angle distribution showing an essentially random distribution of neighbouring or random grain pairs. (f) Grain area distribution diagram. (g) Rose diagram of the long axis of albite matrix grains from the area measured by EBSD.**

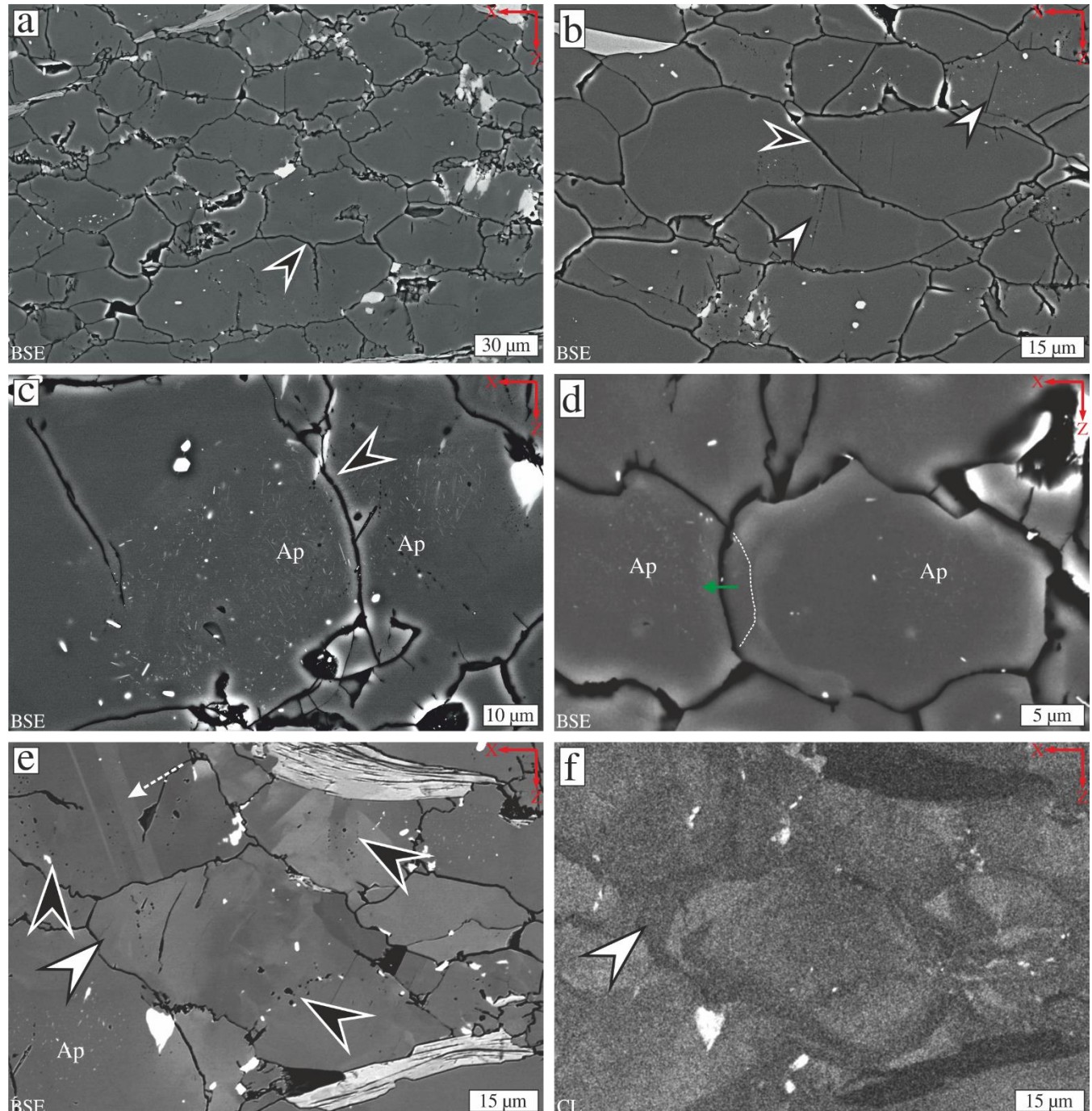

**Figure 13: BSE images of albite grains in type B matrix. (a) Overview showing elongate albite grains and some relict Kfs grains (light grey). Note grain boundary affected by a crack in the grain below (arrow). (b) Boundaries can be remarkably straight, especially at low angle to the foliation (black arrow). Healed microcracks at low angle to shortening direction are indicated by increased porosity (white arrows). (c) Apatite needles in albite grains. Apatite rich zone is crosscut by grain boundary (arrow). (d) Albite grains showing Ca-enriched zone, representing growth rim (green arrow represents growth direction) with the former grain boundary preserved by the zonation (dashed line). (e) Grain with twins (dashed arrow). Black arrows point to porosity, which is**

associated with zonation (white arrow), best seen in the CL image (f). The zonation is probably due to changing contents of trace elements, which cannot be resolved in the BSE image.

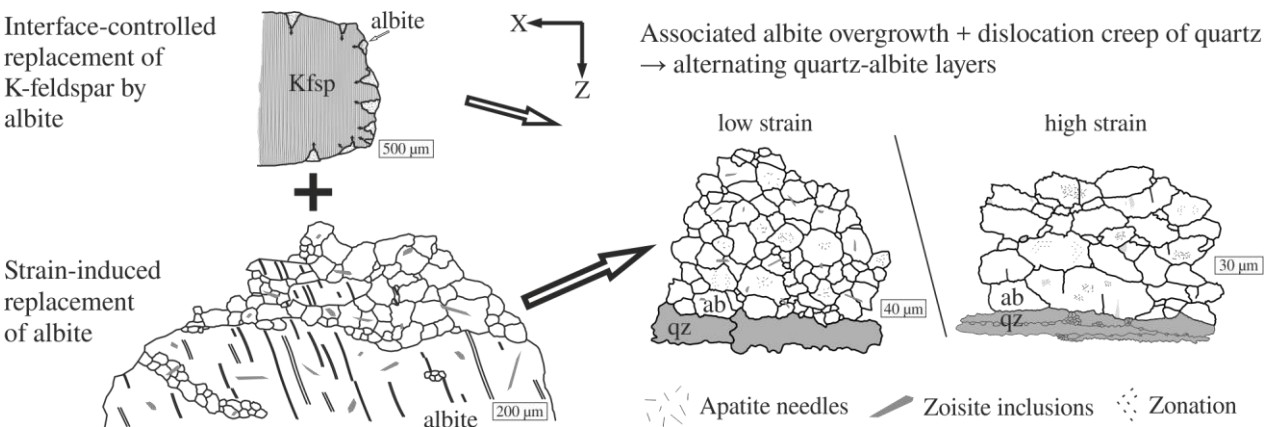

5 **Figure 14: Conceptual sketch of the formation of the mylonitic albite matrix. The contribution of replacement of albite porphyroclasts by albite is larger than that of replacement of K-feldspar (see text for discussion).**

**Appendix**

**Table 1: Overview of the investigated samples**

| Sample | Locality | Coordinates | modal analysis | mode of clasts | Matrix micro-structure | Quartz microstructure |
|--------|----------|-------------|----------------|----------------|------------------------|-----------------------|
| CT599 | Vals | 46°52'33"N 11°37'28"E | Qz: 10%, Plag: 60%, Kfsp: 20%, Trm: 10%, Ms: <1% | Plag: 20 % Kfsp: 80 % | Type B | several mm long and a few tens of µm wide layers; grain size < 20µm |
| UP3 | Uttenheim | 46°52'27.2" N 11°56'0.4" E | Qz: 10%, Plag: 80%, Kfsp: 5%, Trm: 5%, Ms: < 1% | Plag: 90% Kfsp: 10% | Type A | aggregates; grain size 100s of µm, sutured grain boundaries, long (10 µm), straight segments |
| FH1 | Sambock | 46°51'15.8" N 11°53'43.7" E | Qz: 30%, Plag: 50%, Kfsp: 5%, Ms: 15% | Plag: 90% Kfsp: 10% | intermediate | aggregates, grain size 30 µm - 300 µm, sutured grain boundaries, undulous extinction |
| FH4 | Uttenheim | 46°52'12.2" N 11°56'22.6" E | Qz: 50%, Plag: 50% | Plag: 100% Kfsp: 0% | intermediate | isolated grains (50-150 µm) |
| FH5 | Uttenheim | 46°52'26.2" N 11°56'0.8" E | Qz: 15%, Plag: 55%, Kfsp: 25%, Ms: 5% | Plag: 50% Kfsp: 50% | Type A | aggregates; grain size 100s of µm, sutured grain boundaries, long (10 µm), straight segments |

| | | | | | | |
|---|---|---|---|---|---|---|
| FH9 | Uttenheim | 46°52'12.2" N 11°56'22.6" E | Qz: 15%, Plag: 50%, Kfsp: 30%, Ms: 5% | Plag: 50% Kfsp: 50% | intermediate | aggregates, grain size 30 µm - 400 µm, sutured grain boundaries, undulous extinction |
| FH10 | Mühlbach | 46°50'48.5" N 11°58'24.7" E | Qz: 30%, Plag: 45%, Kfsp: 15%, Ms:10% | Plag: 65% Kfsp: 35% | intermediate | aggregates, grain size 30 µm - 400 µm, sutured grain boundaries, undulous extinction |
| FH13 | Eidechs-spitze | 46°52'12.7" N 11°45'33.6" E | Qz: 10%, Plag: 80%, Kfsp: 10%, Ms:<1% | Plag: 85% Kfsp: 15% | Type B | several mm long and a few tens of µm wide layers; recrystallized grains < 20µm in diameter and ~5 µm in shear bands and along fractures |
| FH21 | Sambock | 46°51'31,9" N 11°53'48.2" E | Qz: 15%, Plag: 65%, Kfsp: 15%, Ms: 5% | Plag: 85% Kfsp: 15% | Type B | aggregates, grain size 30 µm - 500 µm, sutured grain boundaries, undulous extinction |
| FH27 | Weitental | 46°51'13.0" N 11°42'52.1" E | Qz: 20%, Plag: 65%, Kfsp: 5%, Ms: 10% | Plag: 95% Kfsp: 5% | Type B | several mm long and a few tens of µm wide layers; recrystallized grains < 20µm in diameter |
| FH30 | Weitental | 46°51'20.8" N 11°42'55.5" E | Qz: 30%, Plag: 40%, Kfsp: 20%, Ms: 10% | Plag: 50% Kfsp: 50% | Type B | aggregates, grain size 30 µm - 300 µm, sutured grain boundaries, undulous extinction |
| FH39 | Weitental | 46°51'15" N 11°42'54.7" E | Qz: 10%, Plag: 55%, Kfsp: 25%, Ms: 10% | Plag: 30% Kfsp: 70% | Type B | several mm long, narrow layers; recrystallized grains < 20µm in diameter |
| FH47 | Uttenheim | 46°52'10.4" N 11°56'23.9" E | Qz: 30%, Plag: 40%, Kfsp: 20%, Ms: 10% | Plag: 55% Kfsp: 45 % | Type A | aggregates; coarse grained (400 of µm), sutured grain boundaries with 10 µm long, straight segments |
| FH60 | Weitental | 46°51'37" N 11°51'33.5" E | Qz: 15%, Plag: 65%, Kfsp: 10%, Ms: 10% | Plag: 75% Kfsp: 25% | Type A | layers < 1mm long; grain size 50 - 300 µm, sutured grain boundaries |
| FH94 | Weitental | 46°52'8.2" N 11°50'42.5" E | Qz: 20%, Plag: 50%, Kfsp: 25%, Ms: 5% | Plag: 35% Kfsp: 65% | intermediate | layers < 1mm long; grain size 50 - 300 µm, ~ 10 µm in shear zones |