# Peer review of "Deformation of feldspar at greenschist facies conditions – the record of mylonitic pegmatites from the Pfunderer Mountains, Eastern Alps"

_Solid Earth, 2018_

## Referee Comment (RC1) · Anonymous Referee #1 · 24 Sep 2018

Comments on the manuscript: ÂńDeformation of feldspar at greenschist facies...Âż by Hentschel, Trepmann, and Janots.

The manuscript describes low temperature mylonites in quartz-feldspar rocks from the Alps. The rocks represent an example of semi-brittle deformation, and there still is a great lack of data and interpretation of such microstructures. Given that such rocks represent one of the strongest portions of the crust or even lithosphere, it is a very important study for understanding the deformation of these strong rocks in the brittle to viscous transition region. The analysis of the microstructures has been carried out carefully and with a lot of detail. The data base is excellent and of very good qual-

ity. The interpretations are for the larger part sound and justified by the data base. Some concepts need to be clarified and explained better in the text. In particular, the relationship between cracking and recrystallization or low temperature plasticity and recrystallization as presented in the text contain some virtually contradictory or at least inconsistent statements. These will be minor to moderate revisions which are required before publication, which I definitely recommend for this manuscript.

Detailed comments:

Abstract, line 10: Better: "...replacement is interpreted to take place by..." line 10: Better: "chemical metastability" instead of "solubility difference", as that term is more general Line 11: omit "in contrast" line 15: "dislocation glide and strain-induced grain boundary migration" – see general comments and comments below concerning this term p.3, line 16: Fig. 1 e,g do not exist, only Fig. 1 p.3, line 28: What is the connection of tertiary ages with the rest of this text? p.4, line 11: insert commas after "argued" and "studies" p.4, line 14: insert "and" after comma p.4, line 15. "mineral" instead of "mineralogical" p.5, line 10: Are the grain sizes given as diameters of equivalent spheres or circles? Mean or mode of the grain size distribution? Please state more details of the grain size analysis. p.5, line 16: omit the sentence: "Feldspar...". This is a repetition, the situation is better explained below in the text. p.5, line 17: better: "...and rarely shows perthitic..." p.5, line 18: better: "...with Ab95-86 is present and in these grains zoisite..." p.5, line 25: Omit "In contrast" at beginning of sentence p.5, line 27: "affected" instead of "influenced" p.6, line 5: better: "...are irregular and rather..." p.6, lines 7-8: I think that there is some indication for host control for the upper left hand quadrant (compare Fig. 4d with 4f). Many of the new grains have an orientation which is vaguely similar to the clast, whereas this is clearly not the case of the other pole figures (4e, g). p.6, line 18: The term "sawtooth-shaped" is not very good. Sawtooth usually implies some asymmetry in the teeth shape, like "monoclinic" shapes. Perhaps it is better to use "cuspate-lobate" or just "lobate" as a descriptive term for these microstructures. p.6, line 19: "into" instead of "through" K-feldspar. p.6,

line 19: "lobate" instead of "curved" grain boundaries p.6, line 24: What do you mean by this sentence? That the cracks terminate at the albite grains or that the albite grains are separated from the host clast? Please explain this better. p.6, line 28: "aggregate" instead of "aggregates" p.7, line 3-4: better: "…that they represent healed cracks … misorientation rather than subgrains (Fig. 8a)." p.7, line7: "… (Fig. 8e), particularly for correlated grain boundaries." p.7, line 17: space after "compositional" and "which" instead of "whis" p.7, line 19: omit "which is" p.7, line 24: "elongated" instead of "lens shaped" (lens is a 3-D term) p.8, line 23: "of" instead of "on" p.8, lines 22-23: The apatite inclusions are interesting. It is difficult to see the apatite inclusions in the K-spar in the images of Fig. 6, but they seem to be there in some cases. Is it possible that the apatite inclusions are also present in the K-spar and can be used to mark the former clast outline of the K-spar grains? p.8, line 24: o.k, the replacement is not directly related to strain, but the stresses will be highest at the grain boundary, so that in a deforming aggregate, the K-spar will be replaced at the highest stress sites. In addition, it is, generally speaking, the higher free energy state of K-spar than albite. Of course, the higher free energy state will result in a higher solubility, but to express it as solubility is a bit unusual as the solubility depends, among other factors, on the fluid composition, which is unknown here. p.8, lines 29-31. The fact that there are albite grains at the boundary of the K-spar clasts (clear replacement structures, Fig. 5b) and that there are K-spar clasts inside the fractures (Fig. 5d), it is obvious that K-spar is replaced by albite. It may be possible that, in addition to the replacement, some albite might also precipitate from a fluid, but it is not necessarily "more likely" (as expressed in the text) than the replacement, for which there is clear evidence. p.9, lines 2-3: The bending may well be results of microcracking, as outlined in Tullis Yund 1987. So, it is not necessarily the result of plasticity. p.9, line 9: Dislocation glide combined with recrystallization (e.g. strain induced grain boundary migration) constitutes, by definition, dislocation creep. Phrased in the way it is written here, the statement is neither correct nor what you want to say. It should be made clear (also in the following discussion section) that the two events (e.g. cracking or glide of dislocations and

the replacement/re- or neocrystallization) are different events or episodic processes, otherwise the combined processes would constitute dislocation creep. p.9, lines 10-12: I agree with this statement, and you are showing in Fig 11 and 13 that there are chemical differences in grains and overgrowth rims. So, chemical effects will be part of the driving potential. p.9, 12-14: As pointed out above, the bending may be the result of microcracking. In addition, the discrete boundaries of misorientation are visible in Fig. 7a (lower arrow marks a discrete misorientation boundary), and in Fig. 8a (many sharp boundaries between dark and light blue). Furthermore, the fragmentation of the albite clast is clearly visible in the Figs. 7 and 8). The brittle deformation induces defects, too. So, certainly low temperature glide processes may occur, but the evidence shown documents primarily cracking processes. p.9, line 28-30: Strain induced grain boundary migration is a recovery or recrystallization mechanism and thus would be part of dislocation creep. Again, as pointed out above, one has to stress the fact that the processes do not occur simultaneously or are not coupled, because dislocation creep is excluded here (for good reasons). p.9, line 32: The "micro-crush zones" point to an important term in this context: "semi-brittle" deformation. I think that this term is perfectly applicable and includes the cracking and replacement/recrystallization aspects. p.10, line 5: omit "in contrast" – this is the start of a new chapter. p.10, lines 16-18: Myrmekitization typically does not occur below 550C, because an intermediate plagioclase composition is required for that. p.10, line21-23: Why only precipitation and not partly replacement? The albite replacing K-spar forms randomly oriented grains (Fig. 5). p.10, lines 28-31: Do you refer to phase mixing by grain boundary sliding? This mechanism is not very effective in producing mixing, and nucleation is far more efficient for that. As you have precipitation (including nucleation?), the mixing in the polyphase material may well be produced by this process. The question is: why is the monophase albite aggregate a single phase material? p.11, lines 28: It seems necessary to include at least a short discussion about what may cause the difference between type A and B microstructures. As everything is documented carefully and in detail, the reader is left without a conclusion concerning these differences. p.12,

lines1-2: What is the difference between "strain-induced replacement of albite with granular flow" and "dissolution precipitation creep"? The old albite (or K-spar) has to be dissolved in some way, and the replacement corresponds to a precipitation. So, given the fact that chemical changes are involved, it still is a type of dissolution precipitation creep process. p.13, lines 1-3: same as p.12, lines 1-2.

Please also note the supplement to this comment:
https://www.solid-earth-discuss.net/se-2018-81/se-2018-81-RC1-supplement.pdf

―――――――――――――――――――――

---

## Referee Comment (RC2) · Dr. Kilian (Referee) · 1 Oct 2018

The manuscript by Hentschel and co-workers reports on microstructures in deformed pegmatites in the Eastern Alps. The intention is to find microstructures which can be related to the active deformation processes and mechanisms in those feldspar dominated rocks. Since these rocks seem to be deformed under greenschist facies conditions, crystal plastic processes (dislocation glide and climb) are expected to be of minor importance and fracturing and dissolution-precipitation and reaction-related processes are those which are usually to be encountered. Since feldspar is an important

mineral but with regard to many processes still quite poorly understood, it is a useful study to gain more insights on processes involved in the deformation behaviour of this group of minerals. The data of the manuscript is overall good, the manuscript is generally and in most parts understandably written and in many parts sufficiently concise. The argumentation and discussion is mostly free of contradictions but in certain parts, clarifications and corrections are required. Comparing the presented data with the interpretations done by the authors, there is room for improvement as some - which will be outline further below - interpretations are rather of speculative nature (although, most likely what one would expect in such a rock and hence in general not surprising). A few concepts presented need to be clarified as in the current way, they might/can not be correct. A few parts seem somewhat lengthy and could benefit from streamlining while e.g. the methods section clearly requires an extended explanation with respect to some of the data shown. Also, the discussion and associated interpretations might benefit from a clearer separation of interpretations based on data and speculations (in part lifted from the literature).

I suggest this manuscript is worth publishing but would strongly benefit from a number of (smaller) enhancements and corrections, which in total suggest major revisions.

1) "Rheologically dominant processes" and strain The author set the scope of the manuscript to identify the rheologically governing processes (eg.  p1,l3; p3,l13, p11,l29ff). It is noted that monophase layers define the "mylonitic microstructure and clearly correlate with strain". However, it remains unclear 1) how strain is determined (overall in the manuscript when reference is taken to "high strain" or "low strain") and b) which is the rheologically governing process. Dissolution-precipitation creep is a deformation mechanism, granular flow can be a mechanism or a process, quartz layers deforming by dislocation creep are another ingredient to bulk rheology of a rock. Which one out of all of these mechanisms in now dominant, is from my point of view still very open - and for a given mechanism, which process dominates the rheology is also not accessed. For example, does "dissolution-precipitation creep with granular flow" mean

it is grain boundary sliding in the sense of Rachinger sliding accommodated by dpc, or dpc accommodated by Lifshitz sliding - and in any of those mechanisms, which is the rate limiting, and hence rheology controlling process? Any answer to this sort of question is what one would expect under a section "Rheologically dominant processes". Maybe the authors meant to actually focus their interpretation on which mechanisms are most likely dominating the formation of a dynamic microstructure? How this relates to bulk rheology within a polyphase rock, evolving microstructurally and mineralogically with strain is then a somewhat different question.

An outcome of the authors interpretation - monophase layers apparently appear most highly deformed - is really interesting and might deserve some discussion. Usually it can be observed that phase mixing of materials undergoing any sort of interface-involved deformation process results in weakening or strain localisation. Why dos this rock behave differently?

2) Please clarify the concept of strain (and related terms) in relation to microstructural observations. "Fractures formed at a low angle to the shortening direction" (p12,l15) and "main shortening direction indicated by the foliation" (p1,6) need clarification. If bulk shortening direction refers to finite strain - as suggested by the reference to the foliation -, it needs an explanation, since this is not what one would expect. Why would fractures form or why would clasts (prone to rotation) containing fractures formed in relation to finite strain? Fractures usually for with some relation to the stress field, i.e. relate to instantaneous strain. If bulk shortening direction refers to the instantaneous shortening direction - which would be physically reasonable (and fractured grains might still be prone to later rotation) - they'd be unrelated to foliation, which seems to be a finite strain feature. Unless the rocks deformed by coaxial progressive deformation (which seems not the case) - a foliation defined by the xy-plane of finite strain is unrelated to the ISA and hence principal stress directions. So, a) the relation of fractures to fabric elements and those relations to finite strain or kinematic directions need clarification and b) I did not find any data actually on the distribution on the trends of fractures

- so any interpretation based on those relations are rather speculations or somewhat vague? Similarly, (e.g. p11, l23): "growth parallel to the stretching lineation" , the stretching lineation is finite strain, why would a grain grow towards this direction?

Similarly "sites of shortening" appears multiple times in the text should refer rather to to e.g. contractional quadrant (in relation to prophyroclasts), surfaces at a high angle with respect to the inferred principal shortening direction or similar, but I'd argue a site of shortening is something like a point, and hence it does not make sense to refer to shortening of a point.

3) Dislocation glide in albite: Bending of kfs is suggested to be mainly due to microfracturing while bending of albite porphyroclasts should primarily relate to dislocation glide. While microfracturing in Kfs might have been identified in the SEM or thinsection (?), I do not see on which data, the absence of microfracturing in favour of dislocation glide in albite is based on? How was microfracturing in ab excluded?

4) Absence of an orientation relation between ab and kfs (p6,l21): The authors present polefigures for three crystal directions (a partial representation of the full crystal orientation e.g.Fig 6e) to discard an orientation relationship between e.g. kfs and albite. However, pole figures are not the suitable object to explore such relationships. Most easily, orientation relations are explored in misorientation space (for example see Krakow et al, 2017). Additionally, as far as I can tell from Fig. 6e, there is quite a lot of coincidence between kfs and ab directions in the pole figures already, so how comes that such a conclusion is drawn?

5) Kfs replacement is independent on specific direction and hence not directly related to strain (p8,l23): How is the rotation of porphyroclasts excluded? I do not see a strong argument here, also no quantitative data to support or reject this claim.

6) Interface-coupled dissolution-precipitation: Conceptually, it has been demonstrated in mostly static environments (see: references in (2) p8,l13ff refer to static features mostly without any deformation involved) and one could argue, that it might be unrelated to strain. However, the opposite argument - because it is apparently independent on (the last state of) strain, it should be icdpc is not tested (see comment on rotating prophyroclasts).

7) "end-member matrix microstructures which correlate with strain" (p1, l18; p7, l13): While the microstructural differences are clearly present, I do not understand where the relation to strain could be established. How was strain measured? How could it be said that one is more strained than the other? Also, do those occur only different samples from different locations - as far as it seems in the way presented here - or could both also be found within the same sample?

8) A few missing explanations in methods and or /figure captions: - how was grain size established - why was frequency distribution and frequency mean chosen over area weighted mean? - how was twinning dealt with in ebsd data wrt grain size or other grain related measures - why are point plots chosen over properly contoured pole figures. In many cases point plots may not be very useful. - Misorientation angle profiles: Misor. angle to origin - please specify what is meant with the various occurrences of "relative misorientation (angle or map)", " internal misorientation (angle)" - the authors note that orientation contrasts camouflages subtle compositional differences in the BSE images, - just adding that the latter then should be, what is seen in CL - so why are then EBSD polished section used for BSE analysis to begin with, if this is a known problem? - How were apatite needles identified? P signal in EDX?

Notes on Figures: Fig. 1: Great to see where samples come from, however out of all of these, only 4 appear in the text. Were the other not suitable or were the selected samples the ones that fit the observation?

Fig. 2: Fractures oriented at small angle to shortening direction - where should that be ? (please indicate shear sense); abbreviation Pl not in the image - see also comments above on fracture orientation

Fig. 3: Unclear what this figure adds to the overall story of the manuscript. Is it needed?

Fig. 4: Pole plots (d,e,g) cannot should be properly contoured. If the message should be, that they are all different, not to distinguish from uniform etc... a proper contouring is needed as pole plots are hard/not to interpret for this purpose. Why are only pole figures plotted for poles to planes and not for directions? Maybe plotting IPDFs for a reasonable reference direction might be even more telling. "relative misorientation map" -> misorientation angle; also relative to what? An arbitrary reference orientation? Grain size histograms: Why are bins chosen to be so narrow that many of them have populations of just one or two grains? Also, please indicate total number. What is the reasoning for the choice of frequency distribution instead if area fraction?

Fig. 5: (e) Pole figures are not very suitable to establish/discard any orientation relationship between the two phases. Maybe colorcoding the misorientation angles might be more telling - or better, colorcoding either for the full misorientation or e.g. misorientation axis might be more telling.

Fig. 5/6: Could it be that the albite growing into kfs is larger than the matrix albite?

Fi. 7: "bent and kinked" Where do I see the difference? (f) What is the bright phase ? Apatite? Some other Ca-phase? It seems that it grows over the clast-new grains boundary (vertical one at the left side).

Fig. 8: (a) relative misorientation -> angle ; also relative to what? As noted in the text, I do not see the necessity that the core-rim orientation gradient in the fragmented clast should relate to crystal plasticity. (a)-(b) Why is the choice of grains different. Also, if in (a) only the central big grain is displayed, why does it seem that in (b) several grains occupy the same area? Red lines being low angle boundaries: In (a), they are barely visible, in (b) it looks like they follow direction which could be consistent with the trend of albite twin boundaries - see also the misorientation profile. Also, comparing (a) and (b), again the segmentation seems to be different i.e. in (a) some of the "low angle boundaries" seem to be actually grain boundaries. So maybe something n the segmentation/ handling of twin boundaries went wrong? Please clarify. (c) what is the

colorcoding of points in the pole plot? (d) a proper contouring might be nicer. (e) Grain size histogram -> see comment on Fig. 4 (g,h) please indicate that this is most likely misorientation angle to origin Fig. 9: Do both matrix types also occur in one and the same sample? Here it's FH5 and CT599 which come from different locations. Any systematics about their occurrences?

Fig. 10: (d) please provide number of grains, what is contoured (1 point per grain or all points) While contouring is much better than the point plots in Fig.9d, it looks like a broader kernel might be more appropriate. (e) Grain size histogram -> see comment on Fig. 4

Fig. 11: (c) So orientation contrast camouflages compositional contrast, so what should be learn from the image? That we can see something in the CL (d) what we might have seen in the BSE if the sample wouldn't have had EBSD-quality polishing?

Fig. 12: Where do color artefacts (center lower part and lower left) in (b) come from? (c) please use a proper kernel for contouring (e) misorientation angle distribution of "albite" Pole figures of pixels or 1 point-per grain? How many data points? It looks like both, ab and kfs is colorcoded in the ipf map: Is that useful? How should one distinguish both there? "maximum mud . . .": Maxima of pole figures are often relatively meaningless, especially if a relatively arbitrary kernel seems to be chosen or multiple maxima exist.The 2-norm of the pole figure (sometimes called pfJ) or any other measure that suits the symmetry and application might be better, or any of proper measures for orientation distribution functions.

Fig. 13: "Preferred growth parallel to stretching lineation" Why would it grow parallel to the finite stretching direction - unless the pure shear p.d. contribution is very large - shouldn't it grow grow parallel to the extending ISA and eventually rotate?

All figures, where a shear sense is available but not provided, should have nice arrows indicating the shear sense.

[Figure]

General notes on figures: Please make sure the reproduced quality will be better than in the manuscript. I assume that the authors submitted high quality figures - and I am aware of the eagerness of file size reduction at the cost of quality at the side of the Copernicus graphics office/ layout people - so please double check later, that the quality of figures remains very good.

A few more notes: p1, l12: Doesn't kinking and twinnign indicate that glide can't be too effective in accommodating deformation?

p1. l21: layers ... parallel to the foliation rather than lineation

p3,l10ff: The last paragraph of the introduction reads like a conclusion, or at least mentions the processes which are later interpreted based on specific microstructures. Is that intentional?

p5. l7: Was ebsd da cleaned of orientation noise? That's usually a good idea before doing KAM/ gKAm

p5, l17: sentence

p5, l30: dilation or extension? (also in other places, please clarify why you think it is dilation and not simply not sites of e.g. lower P)

p6, l33, p7 1ff: Quantifying lattice bending using a misorientation angle wrt origin as a function of distance is not very satisfying since this may only make sense if it can be reasonably assumed that all misorientation is realised around the same axis and rotations remain so low (or at a given symmetry element) that crystal symmetry does not yet matter.

p7, l2: LAB parallel to shortening direction: anything quantitative on that? Also, where is the shortening direction?

p7, l22: What are (monophase) layers composed of aggregates.

p7, l29: How were traces of planes related to real 3d boundary planes?

p8, l2: i.. not show an internal orientation contrast ...

p9,l11-15 (but also elsewhere): Observations and interpretations of the authors are mixed with references to the literature in a way making it hard to figure what information is claimed by the authors and what comes from literature. These sections can benefit from a more clear separation of citation and authors interpretation.

p9. l21: influence of water on diffusion e.g. R&D2004: this most likely relates to gb-transport phenomena, at least it was never demonstrated that it is intracrystalline diffusion, hence it's a bit of a brave jump to speculate on climb enhanced by the presence of fluid - or the absence

p9,l15-26: this is a collection of citations in relation to the inability of dislocation climb and the sluggishness of diffusion in the absence of a hydrous fluid. However, this section might be better placed into the introduction.

p9,l30: "as opposed to solid state grains boundary migration": please explain/clarify; there needs to be transport across the boundary in each case

p10. l1ff(and earlier): While all reasonable in very general terms and something one would expect for such a rock, here a few ingredients to the interpretation are somewhat speculative: a) glide and b) strain induced gbm are not demonstrated. While both may be likely, here it remains a speculation since it is not backed by any (semi) quantitative data

p10. l8: reaction of fracture to crystal directions: a) How was this investigated? and B) is there any data on that?

p10, l19: Dilation: Please explain, it this true dilation or low P sites or surfaces near orthogonal to extensional directions?

p11,l8: albite aggregates instead of albite taking up some deformation

p11, l15: grain boundary sliding: while one can see a few straight boundaries in Fig.

13, a) why should they indicate gbs b) how frequent are those compared to others ?
Anything more convincing on gbs?

p11, l19: Hildyard needs year

p11, l25: "Microstructure correlates with strain": again, where does strain come from?
How does such a "correlation" manifest? Simply elongated vs more equiaxed grains?

p11, l25: "The higher ..." Sentence

p11. l27: growth parallel to the stretching lineation: While this does not make a lot
of sense for non-coaxial p.d. (see comment 2), why preferred growth? Preferred by
what? Crystallography? Where should the "dilation" come from? Anything tested on
that? What is the CPO of the most elongated grains, or which crystal direction is
parallel to to the maximum grain elongation direction?

Entire section 5.6 does not allow me to understand which by now is the process that
dominates rheology.

p12, l1: "granular flow" (here and elsewhere) please define your understanding of gran-
ular flow within the context of a mylonite. Or do you refer to grain boundary sliding in
the sense of Rachinger sliding?

p12, l7: why probably?

p12. l7: Please enlighten (probably not in the conclusion) why the lobate boundaries
between newly grown albite and kfs should be chemical disequilibrium and not due to
other driving forces, i.e. gb-width, porosity variations in kfs, defect densities etc.?

p12, l19: Why would glide drive gs-reduction in this combined mixture of mechanisms?

p12,l20: "observed tendency of slightly enriched Na-content. . ." Any data on that?

p12,l22: Why subordinate? The balance between chemical driving force vs. e.g. strain
energy depends on a lot of variables. For some variables we might have good estimates while for others, we are simply guessing, i.e. dislocation density and elastic energy added by dislocations during deformation etc.

Rüdiger Kilian

Please also note the supplement to this comment:
https://www.solid-earth-discuss.net/se-2018-81/se-2018-81-RC2-supplement.pdf

---

## Author Comment (AC1) · 7 Nov 2018

***Response to referee comments RC1 Solid Earth se-2018-81***

*General comment: Particular, the relationship between cracking and recrystallization or low temperature plasticity and recrystallization as presented in the text contain some virtually contradictory or at least inconsistent statements.*

As the sequence of associated microcracking and dislocation glide of albite (low-temperature plasticity) followed by growth (by strain-induced grain boundary migration and overgrowth parallel to the extensional direction) is one of the major points of this study, we will phrase this sequence more carefully to avoid misunderstandings (see comment to points 32, 35).

*1. Abstract, line 10: Better: "…replacement is interpreted to take place by…"*

We will change this.

*line 10: Better: "chemical metastability" instead of "solubility difference", as that term is more general*

We will follow the suggestions by rephrasing the sentences accordingly when revising the manuscript.

*2. Line 11: omit "in contrast"*

We will omit "in contrast".

*3. line 15: "dislocation glide and strain-induced grain boundary migration" – see general comments and comments below concerning this term*

See general comment and response to points 32 and 35.

*4. p.3, line 16: Fig. 1 e,g do not exist, only Fig. 1*

This was a typesetting error, the e.g. introduces the cited references: (Fig. 1; e.g., Hofmann et al., 1983; …).

*5. p.3, line 28: What is the connection of tertiary ages with the rest of this text?*

Tertiary ages are mentioned in the geologic context. They reveal that the rocks north of the DAV were deformed during alpine metamorphism, in contrast to rocks south of the DAV.

*6. p.4, line 11: insert commas after "argued" and "studies"*

We will add the commas.

*7. p.4, line 14: insert "and" after comma*

We will insert "and" after comma.

*8. p.4, line 15. "mineral" instead of "mineralogical"*

We agree.

*9. p.5, line 10: Are the grain sizes given as diameters of equivalent spheres or circles? Mean or mode of the grain size distribution? Please state more details of the grain size analysis.*

*We were referring to the diameter of a circle with equivalent diameter. We will now describe the area normalized grain size as requested by referee#2 (see comments to RC2, point 12). We will describe the grain size analysis in more detail in the revised manuscript.*

*10. p.5, line 16: omit the sentence: "Feldspar…". This is a repetition, the situation is better explained below in the text.*

We omit this sentence in the revised text.

*11. p.5, line 17: better: "…and rarely shows perthitic…"*

We will rephrase this sentence accordingly.

*12. p.5, line 18: better: "…with Ab95-86 is present and in these grains zoisite…"*

We will rephrase this sentence accordingly.

*13. p.5, line 25: Omit "In contrast" at beginning of sentence*

We agree and will omit "in contrast".

*14. p.5, line 27: "affected" instead of "influenced"*

We will rephrase this sentence accordingly.

*15. p.6, line 5: better: "…are irregular and rather…"*

We will rephrase this sentence accordingly.

*16. p.6, lines 7-8: I think that there is some indication for host control for the upper left hand quadrant (compare Fig. 4d with 4f). Many of the new grains have an orientation which is vaguely similar to the clast, whereas this is clearly not the case of the other pole figures (4e, g).*

We will describe this observation more specifically in the revised manuscript.

*17. p.6, line 18: The term "sawtooth-shaped" is not very good. Sawtooth usually implies some asymmetry in the teeth shape, like "monoclinic" shapes. Perhaps it is better to use "cuspate-lobate" or just "lobate" as a descriptive term for these microstructures.*

We adopted this term from Norberg et al. (2011). However, we agree that the term might be problematic, especially as there is no host-control on the dissolution and reprecipitation, which could lead to asymmetric "teeth". The term "lobate" might rather associate to roundish / smoothly curved boundaries, which is not the case here. Therefore, we will use the suggested term "cuspate".

*18. p.6, line 19: "into" instead of "through" K-feldspar.*

We will rephrase this sentence accordingly.

*19. p.6, line 19: "lobate" instead of "curved" grain boundaries*

We will use "lobate" accordingly.

*20. p.6, line 24: What do you mean by this sentence? That the cracks terminate at the albite grains or that the albite grains are separated from the host clast? Please explain this better.*

We mean that microcracks terminate at new albite grains, which suggests that the albite grains formed after the cracks. We will rephrase this sentence to describe this observation more clearly.

*21. p.6, line 28: "aggregate" instead of "aggregates"*

We will correct this mistake.

*22. p.7, line 3-4: better: "…that they represent healed cracks … misorientation rather than subgrains (Fig. 8a)."*

We will rephrase this sentence accordingly.

*23. p.7, line7: "… (Fig. 8e), particularly for correlated grain boundaries."*

We will add this specification.

*24. p.7, line 17: space after "compositional" and "which" instead of "whis"*

We will correct these typos.

*25. p.7, line 19: omit "which is"*

We will omit "which is".

*26. p.7, line 24: "elongated" instead of "lens shaped" (lens is a 3-D term)*

We rephrase the sentence accordingly.

*27. p.8, line 23: "of" instead of "on"*

We will correct this.

*28. p.8, lines 22-23: The apatite inclusions are interesting. It is difficult to see the apatite inclusions in the K-spar in the images of Fig. 6, but they seem to be there in some cases. Is it possible that the apatite inclusions are also present in the K-spar and can be used to mark the former clast outline of the K-spar grains?*

Indeed, the apatite inclusions can be traced into the K-feldspar. This indicates that not only albite replaced K-feldspar, but there was also precipitation of K-feldspar. We will add this

observation in the revised manuscript. Whether the apatite inclusions can be used to outline the original shape of the K-feldspar is, however, from our point of view too vague.

*29. p.8, line 24: o.k, the replacement is not directly related to strain, but the stresses will be highest at the grain boundary, so that in a deforming aggregate, the K-spar will be replaced at the highest stress sites. In addition, it is, generally speaking, the higher free energy state of K-spar than albite. Of course, the higher free energy state will result in a higher solubility, but to express it as solubility is a bit unusual as the solubility depends, among other factors, on the fluid composition, which is unknown here.*

With the term solubility difference, we want to stress not only the driving force for dissolution of the K-fsp, but also the formation of albite. The higher free energy state depends on many unknown factors as well.

*30. p.8, lines 29-31. The fact that there are albite grains at the boundary of the K-spar clasts (clear replacement structures, Fig. 5b) and that there are K-spar clasts inside the fractures (Fig. 5d), it is obvious that K-spar is replaced by albite. It may be possible that, in addition to the replacement, some albite might also precipitate from a fluid, but it is not necessarily "more likely" (as expressed in the text) than the replacement, for which there is clear evidence.*

We will rephrase the text accordingly.

*31. p.9, lines 2-3: The bending may well be results of microcracking, as outlined in Tullis & Yund 1987. So, it is not necessarily the result of plasticity.*

In bent albite grains we did not find evidence of microcracks at light-optical scale and SEM-scales, yet some influence of microcracking can certainly not be excluded. However, to explain the observation of a continuously bent crystal solely by brittle deformation would be from our point of view too speculative. See also comments to points 34 and referee #2. We will discuss this when revising the manuscript more comprehensively.

*32. p.9, line 9: Dislocation glide combined with recrystallization (e.g. strain induced grain boundary migration) constitutes, by definition, dislocation creep. Phrased in the way it is written here, the statement is neither correct nor what you want to say. It should be made clear (also in the following discussion section) that the two events (e.g. cracking or glide of dislocations and the replacement/re- or neocrystallization) are different events or episodic processes, otherwise the combined processes would constitute dislocation creep.*

We fully agree, we mean a sequence of events, i.e. fracturing and dislocation glide followed by growth (e.g. strain-induced grain boundary migration or overgrowth parallel to the extensional direction). Indeed, this is one of our main point. We will carefully rephrase the text to make our main point more clearly. Please see general comment and point 35.

*33. p.9, lines 10-12: I agree with this statement, and you are showing in Fig 11 and 13 that there are chemical differences in grains and overgrowth rims. So, chemical effects will be part of the driving potential.*

We agree.

*34. p.9, 12-14: As pointed out above, the bending may be the result of microcracking. In addition, the discrete boundaries of misorientation are visible in Fig. 7a (lower arrow marks a discrete misorientation boundary), and in Fig. 8a (many sharp boundaries between dark and light blue). Furthermore, the fragmentation of the albite clast is clearly visible in the Figs. 7 and 8). The brittle deformation induces defects, too. So, certainly low temperature glide processes may occur, but the evidence shown documents primarily cracking processes.*

We agree that cracking is clearly documented by the albite microstructures, as described in chapter 4.3 and 5.2. Plagioclase is showing characteristically a mixture of new grains (strain-free) and fragments (twinned, bent, see Fig. 7 and 8) along boundaries perpendicular to the finite shortening direction. Microcracking can produce dislocations but also dislocation glide can cause micro fracturing. Pile up of dislocations during dislocation glide with ineffective dislocation climb (and thus ineffective recovery) causes strain hardening finally leading to brittle fracturing. The relative role of microcracking versus dislocation glide is clearly difficult to assess from our "post-mortem" approach. Yet, qualitatively, bent and twinned grains without any evidence of microcracks on the light-optical and SEM-scales (as observed here for plagioclase) would indicate that dislocation glide plays a more important role than indicated by healed and sealed intragranular microcracks at low angle to the shortening direction of the finite strain ellipsoid visible on both light-optical and SEM scales (as observed here for K-feldspar, Fig. 5). We will stress this point in our discussion throughout the manuscript and especially in the discussion (Chapters 5.2 and 5.3).

*35. p.9, line 28-30: Strain induced grain boundary migration is a recovery or recrystallization mechanism and thus would be part of dislocation creep. Again, as pointed out above, one has to stress the fact that the processes do not occur simultaneously or are not coupled, because dislocation creep is excluded here (for good reasons).*

We fully agree and will sharpen and stress our arguments for this sequence of microfracturing and associated dislocation glide followed by grain boundary migration. Please see general comment and point 32.

*36. p.9, line 32: The "micro-crush zones" point to an important term in this context: "semibrittle" deformation. I think that this term is perfectly applicable and includes the cracking and replacement/recrystallization aspects.*

We agree.

*37. p.10, line 5: omit "in contrast" – this is the start of a new chapter.*

We will omit "in contrast".

*38. p.10, lines 16-18: Myrmekitization typically does not occur below 550C, because an intermediate plagioclase composition is required for that.*

We agree.

*39. p.10, line21-23: Why only precipitation and not partly replacement? The albite replacing Kspar forms randomly oriented grains (Fig. 5).*

We agree that replacement might occur also in strain shadows. Yet, the (micro-)fabrics indicate shortening perpendicular to the foliation and dilation parallel to the stretching lineation of the finite strain ellipsoid. The polyphase aggregates in strain shadows are taken to indicate precipitation of material that has been dissolved from areas at high angle to the shortening direction, and thus resulting in dilation. Yet, replacement probably occurs as well, which will be mentioned now in addition in the revised manuscript.

*40. p.10, lines 28-31: Do you refer to phase mixing by grain boundary sliding? This mechanism is not very effective in producing mixing, and nucleation is far more efficient for that. As you have precipitation (including nucleation?), the mixing in the polyphase material may well be produced by this process.*

We fully agree, as stated in the text: "In the mylonitic pegmatites reported here, however, no indication of active "phase mixing" is observed and we attribute the occurrence of a polyphase matrix to precipitation." Precipitation includes nucleation, i.e. not only replacement. We refer to Fliervoet (1995), who describes mechanical phase-mixing, though this author does not present a clear explanation of the process. Yet, we argue that we here do not see evidence of any active phase-mixing.

*The question is: why is the monophase albite aggregate a single phase material?*

The next sentence: "Also, the highest strain in the mylonitic pegmatites is associated not with a polyphase matrix but with the monophase quartz and feldspar layers." is used as connecting passage to discuss the monophase albite aggregates in the following Chapter 5.5.

*42. p.11, lines 28: It seems necessary to include at least a short discussion about what may cause the difference between type A and B microstructures. As everything is documented carefully and in detail, the reader is left without a conclusion concerning these differences.*

Type A: isometric fine-grained albite correlated with coarse quartz grains are interpreted to indicate lower strain.

Type B: elongate shape of albite grains are explained by growth parallel to the extensional direction during deformation (i.e. the stretching lineation of the finite strain ellipsoid), i.e. overgrowth at sites of dilation and together with fine-grained quartz interpreted to indicate overall higher strain.

We will discuss this difference more comprehensively.

*43. p.12, lines1-2: What is the difference between "strain-induced replacement of albite with granular flow" and "dissolution precipitation creep"? The old albite (or K-spar) has to be dissolved in some way, and the replacement corresponds to a precipitation. So, given the fact that chemical changes are involved, it still is a type of dissolution precipitation creep process.*

The difference can be expressed as follows:

"Dissolution precipitation creep" refers to dissolution at areas at high angle to the shortening direction and precipitation with nucleation at areas at high angle to the extensional direction (stretching lineation of the finite strain ellipsoid), which usually results in polyphase aggregates in strain shadows.

"Strain-induced replacement of albite" refers to fracturing and dislocation glide in areas at high angle to the shortening direction followed by growth by strain-induced grain boundary and involving precipitation to form grains with high aspect ratio with the long axes parallel to the stretching lineation. The results are monophase aggregates.

Because of the characteristically different microstructures and the characteristic sequence for "Strain-induced replacement of albite" (see points 32, 35, 36) with fracturing and dislocation glide followed by growth involving precipitation (including chemical driving forces in addition to strain) we feel that this difference is important. We will strengthen this difference when revising the manuscript, as this is one of our main points.

---

## Author Comment (AC2) · 7 Nov 2018

***Response to referee comments RC2 Solid Earth se-2018-81***

*1) "Rheologically dominant processes" and strain The author set the scope of the manuscript to identify the rheologically governing processes (eg. p1,l3; p3,l13, p11,l29ff). It is noted that monophase layers define the "mylonitic microstructure and clearly correlate with strain". However, it remains unclear 1) how strain is determined (overall in the manuscript when reference is taken to "high strain" or "low strain") and b) which is the rheologically governing process. Dissolution-precipitation creep is a deformation mechanism, granular flow can be a mechanism or a process, quartz layers deforming by dislocation creep are another ingredient to bulk rheology of a rock. Which one out of all of these mechanisms in now dominant, is from my point of view still very open - and for a given mechanism, which process dominates the rheology is also not accessed. For example, does "dissolution-precipitation creep with granular flow" mean - so any interpretation based on those relations are rather speculations or somewhat vague? Similarly, (e.g. p11, l23): "growth parallel to the stretching lineation" , the stretching lineation is finite strain, why would a grain grow towards this direction?*

We agree that with our "post-mortem" approach it is difficult to quantitatively judge the role of several fundamental deformation mechanisms and associated processes. The goal of this study is to correlate specific microstructures to different processes to evaluate how the microstructure evolved. From our findings, we discuss some aspects on the rheological behaviour through the deformation history. We will phrase this more carefully when revising the manuscript; specifically we will change the heading of Chapter 5.6.

*2) Similarly "sites of shortening" appears multiple times in the text should refer rather to to e.g. contractional quadrant (in relation to prophyroclasts), surfaces at a high angle with respect to the inferred principal shortening direction or similar, but I'd argue a site of shortening is something like a point, and hence it does not make sense to refer to shortening of a point.*

We will rephrase "sites of shortening". We used "site" rather in the sense of a volume and were not referring to a point. We usually refer to a crystalline volume close to boundaries of porphyroclasts at high angle to the shortening direction indicated by the plane normal to the foliation, *z*, of the finite strain ellipsoid. We will explain and phrase this now more carefully when revising the manuscript.

*3) Dislocation glide in albite: Bending of kfs is suggested to be mainly due to microfracturing while bending of albite porphyroclasts should primarily relate to dislocation glide. While microfracturing in Kfs might have been identified in the SEM or thinsection (?both, see Fig. 5), I do not see on which data, the absence of microfracturing in favour of dislocation glide in albite is based on? How was microfracturing in ab excluded?*

We did not exclude microfracturing of albite. In chapters 4.3 and 5.2 we describe albite porphyroclasts that show characteristically a mixture of fragments (twinned, bent, see Fig. 7 and 8) and strain-free new grains along boundaries perpendicular to the shortening direction of the finite strain ellipsoid, resembling "micro-crush zones" described in Tullis and Yund (1987). Microcracking can produce dislocations but also dislocation glide can cause micro fracturing. Pile up of dislocations during dislocation glide with ineffective dislocation climb (thus ineffective recovery) can cause strain hardening finally leading to brittle fracturing. The relative role of microcracking versus dislocation glide is clearly difficult to assess from natural microstructures. Yet, qualitatively, bent and twinned albite porphyroclasts without any evidence of microcracks on the light-optical and SEM-scales together with the albite replacement in "micro-crush zones" at boundaries at high angle to the shortening direction would indicate that dislocation glide plays a more important role for their formation in comparison to the formation of healed and sealed intragranular microcracks at low angle to the shortening direction of the finite strain ellipsoid visible on both light-optical and SEM scales (as characteristically observed here for K-feldspar, Fig. 5). We will strengthen this point throughout the manuscript. See also comments to referee #1 (points 31 and 34) and point 38 below.

*4) Absence of an orientation relation between ab and kfs (p6,l21): The authors present polefigures for three crystal directions (a partial representation of the full crystal orientation e.g.Fig 6e) to discard an orientation relationship between e.g. kfs and albite. However, pole figures are not the suitable object to explore such relationships. Most easily, orientation relations are explored in misorientation space (for example see Krakow et al, 2017). Additionally, as far as I can tell from Fig. 6e, there is quite a lot of coincidence between kfs and ab directions in the pole figures already, so how comes that such a conclusion is drawn?*

We agree that full misorientation space would probably be most telling. For triclinic minerals, this is unfortunately not easy to visualize. In the revised Fig. 5 (which the referee is probably referring to), we will present angle and axis distribution separately and also color-code the phase boundaries according to their misorientation angles. Independent on the way to visualize the EBSD data, a systematic crystallographic relationship between the new grains (green) and the original K-feldspar clast (yellow) is not evident.

*5) Kfs replacement is independent on specific direction and hence not directly related to strain (p8,l23): How is the rotation of porphyroclasts excluded? I do not see a strong argument here, also no quantitative data to support or reject this claim.*

We do not exclude rotation of porphyroclasts, which to some extent appears likely in mylonites. However, we argue that the replacement is not influenced by the orientation of the porphyroclast in the strain field, as the cuspate-phase boundaries to albite occur symmetrically at the boundary of the porphyroclast. If replacement would be significantly influenced by a specific orientation to the strain field, such a symmetric pattern would be difficult to explain by rotation of the porphyroclast with respect to the foliation. Furthermore, the replacements are cut-off by microcracks that occur exclusively at low angle to the shortening direction of the finite strain ellipsoid (Figs. 5; 6). In addition, the elongate shape of the K-feldspar porphyroclasts with the long axis being parallel to the stretching lineation of the finite strain ellipsoid, excludes major rotation of the porphyroclast independently to its surrounding after the formation of these microstructures.

*6) Interface-coupled dissolution-precipitation: Conceptually, it has been demonstrated in mostly static environments (see: references in (2) p8,l13ff refer to static features mostly without any deformation involved) and one could argue, that it might be unrelated to strain. However, the opposite argument - because it is apparently independent on (the last state of) strain, it should be icdpc is not tested (see comment on rotating prophyroclasts).*

The observed cuspate phase boundary of the K-feldspar to albite indicates replacement by dissolution-precipitation processes at the specific phase boundary, i.e. interface-coupled dissolution-precipitation.

*7) "end-member matrix microstructures which correlate with strain" (p1, l18; p7, l13): While the microstructural differences are clearly present, I do not understand where the relation to strain could be established. How was strain measured? How could it be said that one is more strained than the other? Also, do those occur only different samples from different locations - as far as it seems in the way presented here - or could both also be found within the same sample?*

The correlation is, that albite layers with elongated grains and SPO occur in samples, where monophase quartz layers are narrow (a few tens of μm) and several mm long, where the quartz aggregate is fine-grained (several μm in diameter). As opposed to the fine-grained (several μm in diameter) monophase albite layers with isometric grains that occur to coarse-grained (several tens of μm in diameter) quartz layers that have width of hundreds of μm. These microstructures are sample specific. The variations in the width of alternating monophase albite and quartz layers in thin section, correlate with mesoscopic observation of hand specimen and in the field  by the width and spacing of the foliation planes defined by elongate mineral assemblages and is interpreted to reflect strain. We rephrase this correlation to

be interpreted to reflect strain. The distribution of the endmember microstructures will be shown in Fig. 1 and listed in an additional (supplementary?) table (see comments to points 9, 17).

*8) A few missing explanations in methods and or /figure captions: - how was grain size established - why was frequency distribution and frequency mean chosen over area weighted mean?*

We also checked area weighted, which is now changed in all grain diameter histograms

*how was twinning dealt with in ebsd data wrt grain size or other grain related measures*

For grain reconstruction a thresholding value of 10° was used. For grain reconstruction, Dauphiné twin boundaries in quartz are neglected. Evaluating albite grain boundaries in full misorientation space (Krakow et al., 2017) revealed that almost all twins correspond to the albite law and some to the pericline law. Also for grain reconstruction of albite, these twins are neglected by merging along twin boundaries. To evaluate also mean grain orientations, requires to use a higher symmetry, which contains the symmetry element responsible for twinning, which is the point group 121 for albite and 622 for quartz. The mean orientation of the "higher symmetry" grain is the modal orientation of the "lower symmetry" grain. Using the higher symmetry yields the same grain reconstruction result as merging along twin boundaries.

*why are point plots chosen over properly contoured pole figures. In many cases point plots may not be very useful.*

We will use contoured pole figures in the new figures

*Misorientation angle profiles: Misor. angle to origin - please specify what is meant with the various occurrences of "relative misorientation (angle or map)", " internal misorientation (angle)"*

We mean the angle to a reference point or to mean orientation, respectively, as will be described in the caption

*the authors note that orientation contrasts camouflages subtle compositional differences in the BSE images, - just adding that the latter then should be, what is seen in CL - so why are then EBSD polished section used for BSE analysis to begin with, if this is a known problem?*

We checked the grey-scale contrast whether it is derived by compositional or orientation contrasts using EDS and EBSD measurements. BSE orientation contrast is present when there is a difference in orientation, irrespective whether Syton-polished or not.

*How were apatite needles identified? P signal in EDX?* Yes.

We will add the information in the methods-chapter in the revised manuscript.

*9) Notes on Figures: Fig. 1: Great to see where samples come from, however out of all of these, only 4 appear in the text. Were the other not suitable or were the selected samples the ones that fit the observation?*

All samples were carefully analysed and the systematic and characteristic observations are described. We do not want to present the same characteristic and systematic observations from all analysed samples. We will add a table (see comments to points 7, 17), which gives some more overview on our comprehensive data.

*10) Fig. 2: Fractures oriented at small angle to shortening direction - where should that be ? (please indicate shear sense); abbreviation Pl not in the image - see also comments above on fracture orientation*

We will add in the figure caption that the fracture is indicated by the white arrow in Fig. 2 b and also changed Pl (plagioclase) to Ab (albite).

*11) Fig. 3: Unclear what this figure adds to the overall story of the manuscript. Is it needed?*

The figure is intended to give the reader some context on the Alpine metamorphic mineral assemblages. We feel that this information is important even though it does not directly relate to the investigation of feldspar deformation.

*12) Fig. 4: Pole plots (d,e,g) cannot should be properly contoured. If the message should be, that they are all different, not to distinguish from uniform etc… a proper contouring is needed as pole plots are hard/not to interpret for this purpose. Why are only pole figures plotted for poles to planes and not for directions? Maybe plotting IPDFs for a reasonable reference direction might be even more telling. "relative misorientation map" -> misorientation angle; also relative to what? An arbitrary reference orientation? Grain size histograms: Why are bins chosen to be so narrow that many of them have populations of just one or two grains? Also, please indicate total number. What is the reasoning for the choice of frequency distribution instead if area fraction?*

We will present contoured plots, calculated from the ODF, with texture index and pole figure strength as measured. We also give contour intervals now. ODFs were calculated for the mean orientation of grains. Relative misorientation in the map is towards mean orientation, we mention this now in the text.  We revised the histograms and now show area fractions.

*13) Fig. 5: (e) Pole figures are not very suitable to establish/discard any orientation relationship between the two phases. Maybe colorcoding the misorientation angles might be more telling - or better, colorcoding either for the full misorientation or e.g. misorientation axis might be more telling.*

See point 4): We agree that full misorientation space would probably be most telling. For triclinic minerals this is unfortunately not easy to visualize, so we chose to use to present angle and axis distribution separately and also color-coded the phase boundaries according to their misorientation angles.

*14) Fig. 5/6: Could it be that the albite growing into kfs is larger than the matrix albite?*

From our observation, both populations have a similar size distribution. However, we feel that there are too few albite grains replacing K-feldspar to make a meaningful analysis.

*15) Fi. 7: "bent and kinked" Where do I see the difference?*

Indeed, in this image the change in orientation is rather continuous (i.e., "bent"), however, rarely more abrupt changes in orientation occur, which rather resembles "kinking". Yet, because bending is much more common, we will no longer refer to "kinking".

*(f) What is the bright phase ? Apatite? Some other Ca-phase? It seems that it grows over the clast-new grains boundary (vertical one at the left side).*

The bright phase is zoisite, which grows in the rare cases of plagioclase with An-contents up to 14 %. This information will be added in the caption. The zoisite grain is actually fractured at the boundary between clast and new grain. From the positions of other zoisite grains, it is very likely that the zoisite formed before the new grains.

*16) Fig. 8: (a) relative misorientation -> angle ; also relative to what?*

Sorry this was mistake, it relative to the mean orientation, this information will be added.

*As noted in the text, I do not see the necessity that the core-rim orientation gradient in the fragmented clast should relate to crystal plasticity.*

(see comments to point 3 and comments to referee #1, points 31 and 34).

*(a)-(b) Why is the choice of grains different. Also, if in (a) only the central big grain is displayed, why does it seem that in (b) several grains occupy the same area?*

This is a misunderstanding, the choice of grains is not different. In (a) not only the central big grain is colorized, but every albite grain within a maximum misorientation of 30°. We did this exactly because the porphyroclast was fragmented into several grains. (Although, there is still one central grain, which we also used for the pole plot in (c). We will clarify this in the revised manuscript.

*Red lines being low angle boundaries: In (a), they are barely visible, in (b) it looks like they follow direction which could be consistent with the trend of albite twin boundaries - see also the misorientation profile. Also, comparing (a) and (b), again the segmentation seems to be different i.e. in (a) some of the "low angle boundaries" seem to be actually grain boundaries. So maybe something n the segmentation/ handling of twin boundaries went wrong? Please clarify.*

Sorry, yes, we made a mistake in labelling, will correct it and will more clearly display the LAGBs (see comment to point 30). The segmentation is now corrected. The general information is not affected by this.

*(c) what is the colorcoding of points in the pole plot?*

The color-coding is the same as in (a)

*(d) a proper contouring might be nicer.*

As we only want to show the orientation of the porphyroclast, we do not think contouring would add any information.

*(e) Grain size histogram -> see comment on Fig. 4 (g,h)*

We revised the histograms and now show area fractions.

*please indicate that this is most likely misorientation angle to origin.*

We will indicate this.

*17) Fig. 9: Do both matrix types also occur in one and the same sample? Here it's FH5 and CT599 which come from different locations. Any systematics about their occurrences?*

The microstructure type B is more common than the microstructure type A. The distribution will be shown in Fig. 1 and in an additional table, we will state this information more clearly in the revised text (see comments to points 7, 9).

*18) Fig. 10: (d) please provide number of grains, what is contoured (1 point per grain or all points) While contouring is much better than the point plots in Fig.9d, it looks like a broader kernel might be more appropriate. (e) Grain size histogram -> see comment on Fig. 4*

We will provide the number of grains and we recalculated the contouring with a more appropriate (and broader) kernel, determined by the cross-validation approach provided by mtex.

*19) Fig. 11: (c) So orientation contrast camouflages compositional contrast, so what should be learn from the image? That we can see something in the CL (d) what we might have seen in the BSE if the sample wouldn't have had EBSD-quality polishing?*

The BSE signal is showing both, orientation and chemical contrast, independent of Syton-polishing (see also comment to point 7). Orientation contrast does not camouflage compositional contrast inside single grains in this case, but leads to an additional contrast between grains. CL images shows internal structures, which do not cause a strong enough contrast in BSE-images.

*20) Fig. 12: Where do color artefacts (center lower part and lower left) in (b) come from?*

We think the color artefacts come from a high density of unusual twins (not albite/pericline), which are present in these grains. We could not find these twins at any other occasion.

*(c) please use a proper kernel for contouring (e) misorientation angle distribution of "albite" Pole figures of pixels or 1 point-per grain? How many data points? It looks like both, ab and kfs is colorcoded in the ipf map: Is that useful? How should one distinguish both there? "maximum mud …": Maxima of pole figures are often relatively meaningless, especially if a relatively arbitrary kernel seems to be chosen or multiple maxima exist. The 2-norm of the pole figure (sometimes called pfJ) or any other measure that suits the symmetry and application might be better, or any of proper measures for orientation distribution functions.*

We will present contoured plots, calculated from the ODF, with texture index and pole figure strength as measured. We also will give contour intervals. ODFs were calculated for the mean orientation of grains. The one Kfs grain will not be colored in IPF-colors any more.

*21) Fig. 13: "Preferred growth parallel to stretching lineation" Why would it grow parallel to the finite stretching direction - unless the pure shear p.d. contribution is very large shouldn't it grow parallel to the extending ISA and eventually rotate? All figures, where a shear sense is available but not provided, should have nice arrows indicating the shear sense.*

We rephrase that the grains grow parallel to the extensional (or dilatational) direction during deformation, in the finite strain state represented by the stretching lineation (x) (see also comments to points 28, 46). Where a shear sense indicator is present, the shear sense is presented by arrows.

*22) General notes on figures: Please make sure the reproduced quality will be better than in the manuscript. I assume that the authors submitted high quality figures - and I am aware of the eagerness of file size reduction at the cost of quality at the side of the Copernicus graphics office/ layout people - so please double check later, that the quality of figures remains very good.*

Thank you, we will take care of that.

*23) A few more notes: p1, l12: Doesn't kinking and twinnign indicate that glide can't be too effective in accommodating deformation?*

Twinning involves glide of dislocations (e.g., Groshong 1988).

*24) p1. l21: layers … parallel to the foliation rather than lineation*

We will correct this.

*25) p3,l10ff: The last paragraph of the introduction reads like a conclusion, or at least mentions the processes which are later interpreted based on specific microstructures. Is that intentional?*

We will rewrite this part.

*26) p5. l7: Was ebsd da cleaned of orientation noise? That's usually a good idea before doing KAM/ gKAm*

Data was cleaned with a half-quadratic filter before gkam. This will be more comprehensively described in the methods section.

*27) p5, l17: sentence*

We will change that sentence to "The K-feldspar is Na-poor (<10%) and rarely shows perthitic exsolution." See also our comment to the first referee (point 11.).

*28) p5, l30: dilation or extension? (also in other places, please clarify why you think it is dilation and not simply not sites of e.g. lower P)*

During dissolution precipitation creep, "sites" at high angle to the extensional direction are "sites" where new material is precipitated for example in veins or strain shadows, causing dilation in this specific direction and represented in the finite strain state by the stretching lineation (e.g., Groshong, 1988; Passchier and Trouw, 2005, Wassmann and Stöckhert, 2013). In this specific sentence, we will refer to "strain shadows" instead of "areas of dilation".

Groshong, H., 1988. Low-temperature deformation mechanisms and their interpretation. Geol. Soc. Am. Bull. 100, 1329–1360.

Passchier, C.W., Trouw, R.A.J., 2005. Micro-Tectonics. Springer, Heidelberg 159–187.

Wassmann, S., Stöckhert, B., 2013. Rheology of the plate interfaced - dissolution precipitation creep in high pressure metamorphic rocks. Tectonophysics 608, 1-29.

*29) p6, l33, p7 1ff: Quantifying lattice bending using a misorientation angle wrt origin as a function of distance is not very satisfying since this may only make sense if it can be reasonably assumed that all misorientation is realised around the same axis and rotations remain so low (or at a given symmetry element) that crystal symmetry does not yet matter.*

We give the misorientation angle along a distance in addition to the gKam value, as we find this information more intuitively and indeed it refers to the continuous bending of a crystal that is already visible in polarized light micrograph.

*30) p7, l2: LAB parallel to shortening direction: anything quantitative on that?*

We will display the LAGB in Fig. 8 (see comment point 16).

*Also, where is the shortening direction?*

We indicate in all Figures the shortening and stretching directions (z and x) of the finite strain ellipsoid inferred from normal to the foliation and the stretching lineation on sample scale.

*31) p7, l22: What are (monophase) layers composed of aggregates.*

Monophase means just one mineral phase, aggregate means it is composed of different grains (of the same phase, but this is already included in the term "grain")

32) p7, l29: How were traces of planes related to real 3d boundary planes?

We will make clear, that these straight segments can be parallel to traces of (001) and (010) cleavage planes.

*33) p8, l2: i.. not show an internal orientation contrast ...*

Yes, internal, we agree.

*34) p9,l11-15 (but also elsewhere): Observations and interpretations of the authors are mixed with references to the literature in a way making it hard to figure what information is claimed by the authors and what comes from literature. These sections can benefit from a more clear separation of citation and authors interpretation.*

We will carefully revise the text accordingly.

*35) p9. l21: influence of water on diffusion e.g. R&D2004: this most likely relates to gbtransport phenomena, at least it was never demonstrated that it is intracrystalline diffusion, hence it's a bit of a brave jump to speculate on climb enhanced by the presence of fluid - or the absence*

This is a misunderstanding, we did not mean to speculate on the enhancement of climb by the presence of fluid. We referred to findings from the literature and will transfer a few of these aspects into the introduction (see point below).

*36) p9,l15-26: this is a collection of citations in relation to the inability of dislocation climb and the sluggishness of diffusion in the absence of a hydrous fluid. However, this section might be better placed into the introduction.*

We agree and will place a few of these aspects into the introduction.

*37) p9,l30: "as opposed to solid state grains boundary migration": please explain/clarify; there needs to be transport across the boundary in each case*

Our point was to stress the role of dissolution-precipitation as opposed to for example climb-involved subgrain rotation recrystallization (Drury and Urai, 1990; Schenk and Urai, 2005; Stipp and Kunze, 2008). We will clarify this in the revised manuscript.

*38) p10. l1ff(and earlier): While all reasonable in very general terms and something one would expect for such a rock, here a few ingredients to the interpretation are somewhat speculative: a) glide and b) strain induced gbm are not demonstrated. While both may be likely, here it remains a speculation since it is not backed by any (semi) quantitative data*

Here, we disagree, the indication of dislocation glide is not speculative (see comments to referee #1 (points 31 and 34 and point 3 above). Dislocation glide is demonstrated, e.g., by twinning (which involves glide of dislocations, e.g., Groshong 1988, see point 23) and the continuous bending of the crystal lattice. Even the formation of "micro-crush zones" sensu Tullis and Yund (1987) involves dislocation glide. It would be much more speculative to argue that continuous bending of a crystal is purely brittle, especially without any evidence of fracturing on SEM and polarized-light microscopic scales. We will discuss this in some detail in Chapter 5.2. Strain-induced grain boundary migration is likely by the presence of new grains that are basically strain-free and not represent fragments of the original clasts, we will further strengthen this important point when revising the manuscript. However, also overgrowth of grains by precipitation parallel to the extensional direction during deformation will be important in addition to strain-induced grain-boundary migration.

*39) p10. l8: reaction of fracture to crystal directions: a) How was this investigated? and B) is there any data on that?*

We will phrase this more carefully when revising the manuscript. For K-feldspar, we investigated this by comparing the orientation of microcracks to EBSD data. See, for example Fig. 5. Fractures in K-feldspar are clearly related to the axes of the finite strain ellipsoid and not to the crystallography of the crystal. For fragmentation of plagioclase in the "micro-crush" zones, however, cleavage fractures might indeed play a role.

*40) p10, l19: Dilation: Please explain, it this true dilation or low P sites or surfaces near orthogonal to extensional directions?*

See comment to comment to point 28.

*41) p11,l8: albite aggregates instead of albite taking up some deformation*

The aggregates formed from strained albite porphyroclasts…

*42) p11, l15: grain boundary sliding: while one can see a few straight boundaries in Fig. 13, a) why should they indicate gbs b) how frequent are those compared to others ? Anything more convincing on gbs?*

We indeed do not have further microstructural evidence on granular flow except of the fine-grained albite layers deflected around porphyroclasts with minor straight boundaries but mostly lobate boundaries (Fig. 13). We will delete "sliding of grains relative to each other" in that sentence and leave it to "granular flow (e.g. Behrmann and Mainprice, 1987; Stünitz and Fitz Gerald, 1993; Jiang et al., 2000) probably has played a role in the formation of the fine-grained albite matrix."

*43) p11, l19: Hildyard needs year*

We will add it (Hildyard et al., 2011).

*44) p11, l25: "Microstructure correlates with strain": again, where does strain come from? How does such a "correlation" manifest? Simply elongated vs more equiaxed grains?*

Please see comment to point 7).

*45) p11, l25: "The higher ..." Sentence*

We will correct the sentence.

*46) p11. l27: growth parallel to the stretching lineation: While this does not make a lot of sense for non-coaxial p.d. (see comment 2), why preferred growth? Preferred by what? Crystallography? Where should the "dilation" come from? Anything tested on that? What is the CPO of the most elongated grains, or which crystal direction is parallel to the maximum grain elongation direction?*

The observation is: The long axes of grains is parallel to the stretching lineation of the finite strain ellipsoid. There is no preferred crystallographic orientation of grains with high aspect ratio (see also comments to points 21 and 28). We will rephrase the sentence on p11., l. 27 to: "Therefore, we suggest that the albite grains preferentially grew in the direction parallel to the extensional direction during deformation, resulting in a higher aspect ratio of grains with the long axis of grains parallel to the stretching lineation of the finite strain ellipsoid."

*47) Entire section 5.6 does not allow me to understand which by now is the process that dominates rheology.*

We will change the caption heading, see comment to point 1.

*48) p12, l1: "granular flow" (here and elsewhere) please define your understanding of granular flow within the context of a mylonite. Or do you refer to grain boundary sliding in the sense of Rachinger sliding?*

Please see comments to point 42).

*49) p12, l7: why probably?*

We agree and omit "probably", as this is indeed too speculative.

*50) p12. l7: Please enlighten (probably not in the conclusion) why the lobate boundaries between newly grown albite and kfs should be chemical disequilibrium and not due to other driving forces, i.e. gb-width, porosity variations in kfs, defect densities etc.?*

We agree that not only chemical disequilibrium but also other factors do play a role for the replacement and will discuss this more comprehensively in the section 5.1. However, as albite does replace K-feldspar, the chemical driving force is an important factor.

*51) p12, l19: Why would glide drive gs-reduction in this combined mixture of mechanisms?*

As discussed in section 5.2, dislocation glide in association with microfractures in the sense of low-T plasticity is causing a reduction in grain size similar to the "micro-crush" zone in Tullis and Yund, 1987 (see comments to points 3, 38).

*52) p12,l20: "observed tendency of slightly enriched Na-content..." Any data on that?*

We mention the range of compositions of both the porphyroclast and new grains (p5,18; p6,l29-30). They overlap with a slight tendency for new grains to be more Na-rich.

*53) p12,l22: Why subordinate? The balance between chemical driving force vs. e.g. strain energy depends on a lot of variables. For some variables we might have good estimates while for others, we are simply guessing, i.e. dislocation density and elastic energy added by dislocations during deformation etc.*

It is true that many variables are not known and we do not try to ignore this problem. Estimates on the influence of chemical driving forces, consider the chemical differences that we observe as too small to play a significant role (e.g., Stünitz, 1998). We discuss this in Chapter 5.1.

---

## Author Response (AR1)

***Response to referee comments RC1 Solid Earth se-2018-81***

The authors gratefully acknowledge the critical and constructive comments by the referee. In the following, we respond to each of the points raised. The *comments by the referee are given in cursive characters*, our response in blue and the specific changes in the revised manuscript in red. You will find an annotated pdf-file of our revised manuscript below, in which all changes are correlated to the reviewer's comments.

*General comment: Particular, the relationship between cracking and recrystallization or low temperature plasticity and recrystallization as presented in the text contain some virtually contradictory or at least inconsistent statements.*

As the sequence of associated microcracking and dislocation glide of albite (low-temperature plasticity) followed by growth (by strain-induced grain boundary migration and formation of albite growth rims resulting in a SPO) is one of the major points of this study. We phrased this sequence more carefully, especially in the rewritten chapter 5.6., to avoid misunderstandings (see comment to points 32, 35).

*1. Abstract, line 10: Better: "…replacement is interpreted to take place by…"*

We changed this accordingly.

*line 10: Better: "chemical metastability" instead of "solubility difference", as that term is more general*

As formation of these microstructures involve dissolution-precipitation processes, we decided to keep the term "solubility difference", see point 29.

*2. Line 11: omit "in contrast"*

We omitted "in contrast".

*3. line 15: "dislocation glide and strain-induced grain boundary migration" – see general comments and comments below concerning this term*

See general comment and response to points 32 and 35.

*4. p.3, line 16: Fig. 1 e,g do not exist, only Fig. 1*

This was a typesetting error, the e.g. introduces the cited references: (Fig. 1; e.g., Hofmann et al.,1983; …), it is now corrected.

*5. p.3, line 28: What is the connection of tertiary ages with the rest of this text?*

Tertiary ages are mentionedinthe geologic context. They reveal that the rocks north of the DAV were deformed during alpine metamorphism, in contrast to rocks south of the DAV.

*6. p.4, line 11: insert commas after "argued" and "studies"*

We added the commas.

*7. p.4, line 14: insert "and" after comma*

We inserted "and" after comma.

*8. p.4, line 15. "mineral" instead of "mineralogical"*

We changed this accordingly.

*9. p.5, line 10: Are the grain sizes given as diameters of equivalent spheres or circles? Mean or mode of the grain size distribution? Please state more details of the grain size analysis.*

We were referring to the diameter of a circle with equivalent diameter. Wenow describethe area normalized grain size as requested by referee#2 (see comments to RC2, point 12).

We describe the grain size analysis in more detail in the chapter methods revised manuscript. "Grain size analysis was by area normalization excluding border grains. The mean and median of the area distribution are given."

*10. p.5, line 16: omit the sentence: "Feldspar…". This is a repetition, the situation is better explained below in the text.*

We omitted this sentence in the revised text.

*11. p.5, line 17: better: "…and rarely shows perthitic…"*

We rephrased this sentence accordingly.

*12. p.5, line 18: better: "…with Ab95-86 is present and in these grains zoisite…"*

We rephrased this sentence accordingly.

*13. p.5, line 25: Omit "In contrast" at beginning of sentence*

We agree and omitted "in contrast".

*14. p.5, line 27: "affected" instead of "influenced"*

We rephrased this sentence accordingly.

*15. p.6, line 5: better: "…are irregular and rather…"*

We rephrased this sentence accordingly.

*16. p.6, lines 7-8: I think that there is some indication for host control for the upper left hand quadrant (compare Fig. 4d with 4f). Many of the new grains have an orientation which is vaguely similar to the clast, whereas this is clearly not the case of the other pole figures (4e, g).*

Please note that we carefully checked single grain measurements and originally displayed scattered pole figures, but now decided to display density plots, following the suggestions of referee 2. Density plots have been recalculated from odf after segmentation with space group 121. Texture index and pfj are given. All observations can be summarized as follows:

Chapter 4.1. "EBSD measurements of albite in strain shadows were analysed comparing single grain orientations with that of the host, comparing pole figures of scattered measurements as well as density plots recalculated from ODF (Fig. 4d-g). The EBSD data reveal no obvious orientation relationship between new grains within aggregates or a specific relationship between new grains and porphyroclasts, although some new grain orientations might correlate with that of the clast (compare to Fig. 4 f)."

Chapter 5.4. "That few grain orientations in strain shadows are correlating with that of the host crystal is interpreted to be due to the presence of some fragments of the host crystal (Fig. 4d)."

*17. p.6, line 18: The term "sawtooth-shaped" is not very good. Sawtooth usually implies some asymmetry in the teeth shape, like "monoclinic" shapes. Perhaps it is better to use "cuspate-lobate" or just "lobate" as a descriptive term for these microstructures.*

We adopted this term from Norberg et al. (2011). However, we agree that the term might be problematic, especially as there is no host-control on the dissolution and reprecipitation, which could lead to asymmetric "teeth".The term "lobate" might rather associate to roundish / smoothly curved boundaries, which is not the case here. Therefore, we now use the suggested term "cuspate".

*18. p.6, line 19: "into" instead of "through" K-feldspar.*

We rephrased this sentence accordingly.

*19. p.6, line 19: "lobate" instead of "curved" grain boundaries*

We used "lobate" accordingly.

*20. p.6, line 24: What do you mean by this sentence? That the cracks terminate at the albite grains or that the albite grains are separated from the host clast? Please explain this better.*

"Healed microcracks terminate at new albite grains, which therefore formed after fracturing (arrows in Fig. 5c, d). "

*21. p.6, line 28: "aggregate" instead of "aggregates"*

We corrected this mistake.

*22. p.7, line 3-4: better: "…that they represent healed cracks … misorientation rather than subgrains (Fig. 8a)."*

We rephrased this sentence accordingly.

*23. p.7, line7: "… (Fig. 8e), particularly for correlated grain boundaries."*

We added this specification.

*24. p.7, line 17: space after "compositional" and "which" instead of "whis"*

We corrected these typos.

*25. p.7, line 19: omit "which is"*

We omitted "which is".

*26. p.7, line 24: "elongated" instead of "lens shaped" (lens is a 3-D term)*

We rephrased the sentence accordingly.

*27. p.8, line 23: "of" instead of "on"*

We corrected this.

*28. p.8, lines 22-23: The apatite inclusions are interesting. It is difficult to see the apatite inclusions in the K-spar in the images of Fig. 6, but they seem to be there in some cases. Is it possible that the apatite inclusions are also present in the K-spar and can be used to mark the former clast outline of the K-spar grains?*

Indeed, the apatite inclusions can be traced into the K-feldspar. This indicates that not only albite replaced K-feldspar, but there was also precipitation of K-feldspar. We will add this observation in the revised manuscript. Whether the apatite inclusionscan be used to outline the original shape of the K-feldspar is, however, from our point of view too vague.

"The apatite inclusions are in some places also present in the K-feldspar (Fig. 6b, arrows). "

*29. p.8, line 24: o.k, the replacement is not directly related to strain, but the stresses will be highest at the grain boundary, so that in a deforming aggregate, the K-spar will be replaced at the highest stress sites.*

We agree.

Last paragraph chapter 5.1. "The K-feldspar replacement is independent on the orientation of the boundary to the foliation and stretching lineation and is therefore interpreted to be not directly related to the strain field during deformation, not excluding some influence of higher strain along the boundary compared to within the crystal."

*In addition, it is, generally speaking, the higher free energy state of K-spar than albite. Of course, the higher free energy state will result in a higher solubility, but to express it as solubility is a bit unusual as the solubility depends, among other factors, on the fluid composition, which is unknown here.*

With the term solubility difference, we want to stress not only the driving force for dissolution of the K-fsp, but also the formation of albite. The higher free energy state depends on many unknown factors as well, see point 1.

*30. p.8, lines 29-31. The fact that there are albite grains at the boundary of the K-spar clasts (clear replacement structures, Fig. 5b) and that there are K-spar clasts inside the fractures (Fig. 5d), it is obvious that K-spar is replaced by albite. It may be possible that, in addition to the replacement, some albite might also precipitate from a fluid, but it is not necessarily "more likely" (as expressed in the text) than the replacement, for which there is clear evidence.*

We agree and deleted "more likely".

*31. p.9, lines 2-3: The bending may well be results of microcracking, as outlined in Tullis&Yund 1987. So, it is not necessarily the result of plasticity.*

In bent albite grains we did not find evidence of microcracks at light-optical scale and SEM-scales, yet some influence of microcracking can certainly not be excluded. However, to explain the observation of a continuously bent crystal solely by brittle deformation would be from our point of view too speculative. See also comments to points 34 and referee #2. We will discuss this when revising the manuscript more comprehensively.

"Bent twins are corresponding to an undulous extinction indicating a continuous internal misorientation, which is usually taken to result from the presence of geometrically necessary dislocations (e.g., Nicolas and Poirier, 1976; Wheeler et al., 2009), though some microcracking might also be involved, as pointed out by Tullis and Yund (1987). For albite this continuous internal misorientation is not associated to distributed healed microfractures, as observed for K-feldspar (compare Figs. 5 and 7), which indicates the relative higher importance of dislocation glide for the deformation of albite compared to K-feldspar."

*32. p.9, line 9: Dislocation glide combined with recrystallization (e.g. strain induced grain boundary migration) constitutes, by definition, dislocation creep. Phrased in the way it is written here, the statement is neither correct nor what you want to say. It should be made clear (also in the following discussion section) that the two events (e.g. cracking or glide of dislocations and the replacement/re- or neocrystallization) are different events or episodic processes, otherwise the combined processes would constitute dislocation creep.*

We fully agree, we mean a sequence of events, i.e. fracturing and dislocation glide followed by growth (e.g. strain-induced grain boundary migration or growth rims resulting in a SPO). Indeed, this is one of our main point. Please see general comment and point 35.

"Strain-induced grain boundary migration following dislocation glide and microfracturing is consistent with an orientation scatter around the orientation of the host porphyroclast (Fig. 8d, f)."

*33. p.9, lines 10-12: I agree with this statement, and you are showing in Fig 11 and 13 that there are chemical differences in grains and overgrowth rims. So, chemical effects will be part of the driving potential.*

We agree.

*34. p.9, 12-14: As pointed out above, the bending may be the result of microcracking. In addition, the discrete boundaries of misorientation are visible in Fig. 7a (lower arrow marks a*

*discrete misorientation boundary), and in Fig. 8a (many sharp boundaries between dark and light blue). Furthermore, the fragmentation of the albite clast is clearly visible in the Figs. 7 and 8). The brittle deformation induces defects, too. So, certainly low temperature glide processes may occur, but the evidence shown documents primarily cracking processes.*

We agree that cracking is clearly documented by the albite microstructures, as described in chapter 4.3 and 5.2. Albite is showing characteristically a mixture of new grains (strain-free) and fragments (twinned, bent, see Fig. 7 and 8) along boundaries parallel to the foliation. Microcracking can produce dislocations but also dislocation glide can cause micro fracturing. Pile up of dislocations during dislocation glide with ineffective dislocation climb (and thus ineffective recovery) causes strain hardening finally leading to brittle fracturing. The relative role of microcracking versus dislocation glide is clearly difficult to assess from our "post-mortem" approach. Yet, qualitatively, bent and twinned grains without any evidence of microcracks on the light-optical and SEM-scales (as observed here for plagioclase) would indicate that dislocation glide plays a more important role than indicated by healed and sealed intragranular microcracks at high angle to the stretching lineation visible on both light-optical and SEM scales (as observed here for K-feldspar, Fig. 5).

We stressed this point in our discussion throughout the manuscript and especially in the discussion (Chapters 5.2 and 5.3), see point 31.

*35. p.9, line 28-30: Strain induced grain boundary migration is a recovery or recrystallization mechanism and thus would be part of dislocation creep. Again, as pointed out above, one has to stress the fact that the processes do not occur simultaneously or are not coupled, because dislocation creep is excluded here (for good reasons).*

We fully agree and will sharpen and stress our arguments for this sequence of microfracturing and associated dislocation glide followed by grain boundary migration. Please see general comment and point 32.

new version:

"We suggest that albite porphyroclasts deform in the regime of low-temperature plasticity, where dislocation climb is ineffective and where dislocation glide leads to strain hardening and microfracturing. Additionally, dislocations can be induced by microfracturing (Tullis and Yund, 1987). Subsequently, grains grow by strain-induced grain boundary migration, where crystalline volume with higher strain energy is dissolved and strain-free crystalline volumes precipitated, as was also found by Tullis and Yund (1987)."

*36. p.9, line 32: The "micro-crush zones" point to an important term in this context: "semibrittle" deformation. I think that this term is perfectly applicable and includes the cracking and replacement/recrystallization aspects.*

We agree.

*37. p.10, line 5: omit "in contrast" – this is the start of a new chapter.*

We omitted "in contrast".

*38. p.10, lines 16-18: Myrmekitization typically does not occur below 550C, because an intermediate plagioclase composition is required for that.*

We agree.

*39. p.10, line21-23: Why only precipitation and not partly replacement? The albite replacing Kspar forms randomly oriented grains (Fig. 5).*

We agree that replacement might occur also in strain shadows. Yet, the (micro-)fabrics indicate shortening perpendicular to the foliation and dilation/extension parallel to the stretching lineation of the finite strain ellipsoid. The polyphase aggregates in strain shadows are taken to indicate precipitation of material that has been dissolved from boundaries parallel to the foliation. Yet, replacement probably occurs as well, which is now mentioned in addition in the revised manuscript.

"Additionally, some replacement might also occur in strain shadows."

*40. p.10, lines 28-31: Do you refer to phase mixing by grain boundary sliding?This mechanism is not very effective in producing mixing, and nucleation is far more efficient for that. As you have precipitation (including nucleation?), the mixing in the polyphase material may well be produced by this process.*

We fully agree, as stated in the text: "In the mylonitic pegmatites reported here, however, no indication of active "phase mixing" is observed and we attribute the occurrence of a polyphase matrix to precipitation." Precipitation includes nucleation, i.e. not only replacement. We refer to Fliervoet (1995), who describes mechanical phase-mixing, though this author does not present a clear explanation of the process. Yet, we argue that we here do not see evidence of any active phase-mixing.

We state this more clearly in Chapter 5.5, second paragraph of the revised manuscript.

*The question is: why is the monophase albite aggregate a single phase material?*

The next sentence: "Also, the highest strain in the mylonitic pegmatites is associated not with a polyphase matrix but with the monophase quartz and feldspar layers." is used as connecting passage to discuss the monophase albite aggregates in the following Chapter 5.5.

*42. p.11, lines 28: It seems necessary to include at least a short discussion about what may cause the difference between type A and B microstructures. As everything is documented carefully and in detail, the reader is left without a conclusion concerning these differences.*

Chapter 5.5, last two paragraphs "Quartz layers of coarse recrystallized grains systematically correlate with albite layers of small isometric grains in the type A matrix microstructure (sect. 4.4; Figs. 9a, b; 10; 11). In contrast, narrow quartz layers with fine-grained quartz aggregates and marked CPO are correlated with elongate coarser albite in the type B matrix microstructure (sect. 4.4; Figs. 9c, d; 12; 13).

The elongate shape of albite and zones of high porosity at boundaries at high angle to the stretching lineation in the type B matrix microstructure indicates growth by precipitation (Fig. 13e, f). The microstructure correlates with the overall strain of the mylonitic matrix (Fig. 14). "

*43. p.12, lines1-2: What is the difference between "strain-induced replacement of albite with granular flow" and "dissolution precipitation creep"? The old albite (or K-spar) has to be dissolved in some way, and the replacement corresponds to a precipitation. So, given the fact that chemical changes are involved, it still is a type of dissolution precipitation creep process.*

The differencecan be expressed as follows:

"Dissolution precipitation creep" refers to dissolution at boundaries parallel to the foliation and precipitation with nucleation at areas at high angle to the stretching lineation, which usually results in polyphase aggregates in strain shadows.

"Strain-induced replacement of albite" refers to fracturing and dislocation glide of porphyroclasts along boundaries parallel to the foliation followed by growth by strain-induced grain boundary and involving precipitation to form grains with high aspect ratio with the long axes in the foliation. The results are monophase aggregates.

Because of the characteristically different microstructures and the characteristic sequence for"Strain-induced replacement of albite" (see points 32, 35, 36) with fracturing and dislocation glide followed by growth involving precipitation (including chemical driving forces in addition to strain) we feel that this difference is important.

We strengthened this difference when revising the manuscript, as this is one of our main points, see especially the new discussion in Chapter 5.6.

***Response to referee commentsRC2 Solid Earth se-2018-81***

The authors gratefully acknowledge the critical and constructive comments by Dr.Rüdiger Kilian. In the following, we respond to each of the points raised. The *comments by the referee are given in cursive characters*, our response in blue, specific changes in the revised manuscript in red. You will find an annotated pdf-file of our revised manuscript below, in which all changes are correlated to the reviewer's comments.

*1) "Rheologically dominant processes" and strain The author set the scope of the manuscript to identify the rheologically governing processes (e.g. p1,l3; p3,l13, p11,l29ff). It is noted that monophase layers define the "mylonitic microstructure and clearly correlate with strain". However, it remains unclear 1) how strain is determined (overall in the manuscript when reference is taken to "high strain" or "low strain") and b) which is the rheologically governing process. Dissolution-precipitation creep is a deformation mechanism, granular flow can be a mechanism or a process, quartz layers deforming by dislocation creep are another ingredient to bulk rheology of a rock. Which one out of all of these mechanisms in now dominant, is from my point of view still very open - and for a given mechanism, which process dominates the rheology is also not accessed. For example, does "dissolution-precipitation creep with granular flow" mean- so any interpretation based on those relations are rather speculations or somewhat vague?*

Concerning the first comment on strain, please see answer to point 7, as the relative strain is indicated by the width of the alternating quartz-albite layers referring to the "end-member matrix microstructures.

Concerning "rheologically dominant processes", we agree that with our "post-mortem" approach it is difficult to quantitatively judge the role of several fundamental deformation mechanisms and associated processes. The goal of this study is to correlate specific microstructures to different processes and to evaluate how the microstructure evolved. From our findings, we discuss some aspects on the rheological behaviour through the deformation history, which we feel are a relevant outcome of our study.

This is now more carefully phrased throughout the manuscript (please see annotated pdf-file) and especially discussed in Chapter 5.6., now headed: "Implications for rock rheology and deformation history".

*2) Similarly, (e.g. p11, l23): "growth parallel to the stretching lineation" , the stretching lineation is finite strain, why would a grain grow towards this direction? Similarly "sites of shortening" appears multiple times in the text should refer rather to to e.g. contractional quadrant (in relation to prophyroclasts), surfaces at a high angle with respect to the inferred principal shortening direction or similar, but I'd argue a site of shortening is something like a point, and hence it does not make sense to refer to shortening of a point.*

We used "site" rather in the sense of a volume and were not referring to a point. We usually refered to a crystalline volume close to boundaries of porphyroclasts parallel to the foliation (perpendicular to *z*).

We now refer to boundaries parallel to the foliation, boundaries perpendicular to the stretching lineation and/or strain shadows throughout the revised manuscript.

*3) Dislocation glide in albite: Bending of kfs is suggested to be mainly due to microfracturing while bending of albite porphyroclasts should primarily relate to dislocation glide. While microfracturing in Kfs might have been identified in the SEM or thinsection (?both, see Fig. 5), I do not see on which data, the absence of microfracturing in favour of dislocation glide in albite is based on? How was microfracturing in ab excluded?*

We did not exclude microfracturing of albite. In chapters 4.3 and 5.2 we describe albite porphyroclasts that show characteristically a mixture of fragments (twinned, bent, see Fig. 7 and 8) and strain-free new grains along boundaries parallel to the foliation, resembling "micro-crush zones" described in Tullis and Yund (1987). Microcracking can produce dislocations but also dislocation glide can cause micro fracturing. Pile up of dislocations during dislocation glide with ineffective dislocation climb (thus ineffective recovery) can cause strain hardening finally leading to brittle fracturing. The relative role of microcracking versus dislocation glide is clearly difficult to assess from natural microstructures. Yet, qualitatively, bent and twinned albite porphyroclasts without any evidence of microcracks on the light-optical and SEM-scales together with the albite replacement in "micro-crush zones" at boundaries parallel to the foliation would indicate that dislocation glide plays a more important role for their formation in comparison to the formation of healed and sealed intragranular microcracks at high angle to the stretching lineation visible on both light-optical and SEM scales (as characteristically observed here for K-feldspar, Fig. 5).

We strengthened this point throughout the manuscript. See also comments to referee #1 (points 31 and 34) and point 38 below.

Chapter 5.2: Strain-induced replacement of albite

"...Bent twins are corresponding to an undulous extinction indicating a continuous internal misorientation, which is usually taken to result from the presence of geometrically necessary dislocations (e.g., Nicolas and Poirier, 1976; Poirier, 1985; Wheeler et al., 2009), though some microcracking might also be involved, as pointed out by Tullis and Yund (1987). For albite this continuous internal misorientation is not associated to distributed healed microfractures, as observed for K-feldspar (compare Figs. 5 and 7), which indicates the relative higher importance of dislocation glide for the deformation of albite compared to K-feldspar."

4) Absence of an orientation relation between ab and kfs (p6,l21): The authors present polefigures for three crystal directions (a partial representation of the full crystal orientation e.g. Fig 6e) to discard an orientation relationship between e.g. kfs and albite. However, pole figures are not the suitable object to explore such relationships. Most easily, orientation relations are explored in misorientation space (for example see Krakow et al, 2017). Additionally, as far as I can tell from Fig. 6e, there is quite a lot of coincidence between kfs and ab directions in the pole figures already, so how comes that such a conclusion is drawn?

We agree that full misorientation space would probably be most telling. For triclinic minerals, this is unfortunately not easy to visualize. In the revised Fig. 5 (which the referee is probably referring to), we present angle and axis distribution separately and also color-code the phase boundaries according to their misorientation angles. Independent on the way to visualize the EBSD data, a systematic crystallographic relationship between the new grains (green) and the original K-feldspar clast (yellow) is not evident.

5) Kfs replacement is independent on specific direction and hence not directly related to strain (p8,l23): How is the rotation of porphyroclasts excluded? I do not see a strong argument here, also no quantitative data to support or reject this claim.

We do not exclude rotation of porphyroclasts, which to some extent appears likely in mylonites. However, we argue that the replacement is not influenced by the orientation of the porphyroclast in the strain field, as the cuspate-phase boundaries to albite occur symmetrically at the boundary of the porphyroclast. If replacement would be significantly influenced by a specific orientation to the strain field, such a symmetric pattern would be difficult to explain by rotation of the porphyroclast with respect to the foliation. Furthermore, the replacements are cut-off by microcracks that occur exclusively at high angle to the stretching lineation (Figs. 5; 6). In addition, the elongate shape of the K-feldspar porphyroclasts with the long axis being parallel to the stretching lineation of the finite strain ellipsoid, excludes major rotation of the porphyroclast independently to its surrounding after the formation of these microstructures.

6) Interface-coupled dissolution-precipitation: Conceptually, it has been demonstrated in mostly static environments (see: references in (2) p8,l13ff refer to static features mostly without any deformation involved) and one could argue, that it might be unrelated to strain. However, the opposite argument - because it is apparently independent on (the last state of) strain, it should be icdpc is not tested (see comment on rotating prophyroclasts).

The observed cuspate phase boundary of the K-feldspar to albite indicates replacement by dissolution-precipitation processes at the specific phase boundary, i.e. interface-coupled dissolution-precipitation.

7) "end-member matrix microstructures which correlate with strain" (p1, l18; p7, l13): While the microstructural differences are clearly present, I do not understand where the relation to strain could be established. How was strain measured? How could it be said that one is more strained than the other? Also, do those occur only different samples from different locations - as far as it seems in the way presented here - or could both also be found within the same sample?

We only refer to a relative difference in strain. The correlation is, that albite layers with elongated grains and SPO occur in samples, where monophase quartz layers are narrow (a few tens of μm) and several mm long, where the quartz aggregate is fine-grained (several μm in diameter). As opposed to the fine-grained (several μm in diameter) monophase albite layers with isometric grains that occur to coarse-grained (several tens of μm in diameter) quartz layers that have width of hundreds of μm. These microstructures are sample specific. The variations in the width of alternating monophase albite and quartz layers in thin section, correlate with mesoscopic observation of hand specimen and in the field by the width and spacing of the foliation planes defined by elongate mineral assemblages and is interpreted to reflect strain. We rephrased this more specifically throughout the manuscript and in the last paragraph in chapter 4.4. The distribution of the endmember microstructures is now shown in Fig. 1 and listed in an additional supplementary table (see comments to points 9, 17).

8) A few missing explanations in methods and or /figure captions: - how was grain size established - why was frequency distribution and frequency mean chosen over area weighted mean? We also checked area weighted, which is now changed in all grain area histograms-how was twinning dealt with in ebsd data wrt grain size or other grain related measures -For grain reconstruction a thresholding value of 10° was used. For grain reconstruction, Dauphiné twin boundaries in quartz are neglected. Evaluating albite grain boundaries in full misorientation space (Krakow et al., 2017) revealed that almost all twins correspond to the albite law and some to the pericline law. Also for grain reconstruction of albite, these twins are neglected by merging along twin boundaries. To evaluate also mean grain orientations, requires to use a higher symmetry, which contains the symmetry element responsible for twinning, which is the point group 121 for albite and 622 for quartz. The mean orientation of the "higher symmetry" grain is the modal orientation of the "lower symmetry" grain. Using the higher symmetry yields the same grain reconstruction result as merging along twin boundaries.why are point plots chosen over properly contoured pole figures. In many cases point plots may not be very useful. We will use contoured pole figures in the new figures - Misorientation angle profiles: Misor. angle to origin - please specify what is meant with the various occurrences of "relative misorientation (angle or map)", " internal misorientation (angle)" –we mean the angle to a reference point or to mean orientation, respectively, as will be described in the caption - the authors note that orientation contrasts camouflages subtle compositional differences in the BSE images, - just adding that the latter then should be, what is seen in CL - so why are then EBSD polished section used for BSE analysis to begin with, if this is a known problem? We checked the grey-scale contrast whether it is derived by compositional or orientation contrasts using EDS and EBSD measurements. BSE orientation contrast is present when there is a difference in orientation, irrespective whether Syton-polished or not. - How were apatite needles identified? P signal in EDX? Yes.

We added the information in the methods-chapter in the revised manuscript.

9) Notes on Figures: Fig. 1: Great to see where samples come from, however out of all of these, only 4 appear in the text. Were the other not suitable or were the selected samples the ones that fit the observation?

All samples were carefully analysed and the systematic and characteristic observationsare described. We do not want to present the same characteristic and systematic observations from all analysed samples.

We added a table (see comments to points 7, 17), which gives some more overview on our comprehensive data. Changes in Fig. 1: We colored the sample numbers according to the type of albite-quartz matrix.

10) Fig. 2: Fractures oriented at small angle to shortening direction - where should that be ? (please indicate shear sense); abbreviation Pl not in the image - see also comments above on fracture orientation

We added in the figure caption that the fracture is indicated by the white arrow in Fig. 2 b and also changed Pl (plagioclase) to Ab (albite). We rephrased to fractures oriented at high angle to the stretching lineation.

11) Fig. 3: Unclear what this figure adds to the overall story of the manuscript. Is it needed?

The figure is intended to give the reader some context on the Alpine metamorphic mineral assemblages. We feel that this information is important even though it does not directly relate to the investigation of feldspar deformation.

12) Fig. 4: Pole plots (d,e,g) cannot should be properly contoured. If the message should be, that they are all different, not to distinguish from uniform etc… a proper contouring is needed as pole plots are hard/not to interpret for this purpose. Why are only pole figures plotted for poles to planes and not for directions? Maybe plotting IPDFs for a reasonable reference direction might be even more telling. "relative misorientation map" -> misorientation angle; also relative to what? An arbitrary reference orientation? Grain size histograms: Why are bins chosen to be so narrow that many of them have populations of just one or two grains? Also, please indicate total number. What is the reasoning for the choice of frequency distribution instead if area fraction?

Changes in Fig. 4:

We present contoured plots, calculated from the ODF, with texture index and pole figure strength as measured. We also give contour intervals now. ODFs were calculated for the mean orientation of grains. Relative misorientation in the map is towards mean orientation, we mention this now in the text. We revised the histograms and now show area fractions.

13) Fig. 5: (e) Pole figures are not very suitable to establish/discard any orientation relationship between the two phases. Maybe colorcoding the misorientation angles might be more telling - or better, colorcoding either for the full misorientation or e.g. misorientation axis might be more telling.

See point 4): We agree that full misorientation space would probably be most telling. For triclinic minerals this is unfortunately not easy to visualize, so we chose to use to present angle and axis distribution separately and also color-coded the phase boundaries according to their misorientation angles.

Changes in Fig. 5:
We colored the grain boundaries between replacing albite and the Kfs porphyroclast according to their misorientation angle and exclude the grains in the fracture. Additionally, misorientation axis and angle distributions are shown for these boundaries.

14) Fig. 5/6: Could it be that the albite growing into kfs is larger than the matrix albite?

From our observation, both populations have a similar size distribution. However, we feel that there are too few albite grains replacing K-feldspar to make a meaningful analysis.

15) Fi. 7: "bent and kinked" Where do I see the difference? Indeed, in this image the change in orientation is rather continuous (i.e., "bent"), however, rarely more abrupt changes in orientation occur, which rather resembles "kinking". Yet, because bending is much more common, we will no longer refer to "kinking". (f) What is the bright phase ? Apatite? Some other Ca-phase? It seems that it grows over the clast-new grains boundary (vertical one at the left side). The bright phase is zoisite, which grows in the rare cases of plagioclase with An-contents up to 14 %. This information will be added in the caption. The zoisite grain is actually fractured at the boundary between clast and new grain. From the positions of other zoisite grains, it is very likely that the zoisite formed before the new grains.

Changes in Fig. 7:
We now mark the zoisite grains.

16) Fig. 8: (a) relative misorientation -> angle ; also relative to what?Sorry this was mistake, it relative to the mean orientation, this information will be added. As noted in the text, I do not see the necessity that the core-rim orientation gradient in the fragmented clast should relate to crystal plasticity. (see comments to point 3 and comments to referee #1, points 31 and 34). (a)-(b) Why is the choice of grains different. Also, if in (a) only the central big grain is displayed, why does it seem that in (b) several grains occupy the same area? This is a misunderstanding, the choice of grains is not different. In (a) not only the central big grain is colorized, but every albite grain within a maximum misorientation of 30°. We did this exactly because the porphyroclast was fragmented into several grains. (Although, there is still one central grain, which we also used for the pole plot in (c). We will clarify this in the revised manuscript.Red lines being low angle boundaries: In (a), they are barely visible, in (b) it looks like they follow direction which could be consistent with the trend of albite twin boundaries - see also the misorientation profile. Also, comparing (a) and (b), again the segmentation seems to be different i.e. in (a) some of the "low angle boundaries" seem to be actually grain boundaries. So maybe something n the segmentation/ handling of twin boundaries went wrong? Please clarify. Sorry, yes, we made a mistake in labelling, we corrected it and now more clearly display the LAGBs (see comment to point 30). The segmentation is now corrected. The general information is not affected by this. (c) what is thecolorcoding of points in the pole plot? The color-coding is the same as in (a) (d) a proper contouring might be nicer. As we only want to show the orientation of the porphyroclast, we do not think contouring would add any information. (d) a proper contouring might be nicer.As we only want to show the orientation of the porphyroclast, we do not think contouring would add any information. (e) Grain size histogram -> see comment on Fig. 4 (g,h) We revised the histograms and now show area fractions. please indicate that this is most likely misorientation angle to origin.We now indicate this.

Changes in Fig. 8:
For (a) and (b) the grain segmentation with space group 121 is now consistent between both figures. The pole figures in (d) where recalculated from odf and texture index and pfj are given now. We now give an area normalized distribution in (e).

17) Fig. 9: Do both matrix types also occur in one and the same sample? Here it's FH5 and CT599 which come from different locations. Any systematics about their occurrences? The microstructure type B is more common than the microstructure type A and there is a slight different distribution from West-East.

The distribution is now shown in Fig. 1 and in an additional table, we state this information now in the revised text (see comments to points 7, 9).

18) Fig. 10: (d) please provide number of grains, what is contoured (1 point per grain or all points) While contouring is much better than the point plots in Fig.9d, it looks like a broader kernel might be more appropriate. (e) Grain size histogram -> see comment on Fig. 4

We provide the number of grains and we recalculated the contouring with a more appropriate (and broader) kernel, determined by the cross-validation approach provided by MTex.

The pole figures in (d) where recalculated from ODF and texture index and pfj are given now. We now give an area normalized distribution in (e).

19) Fig. 11: (c) So orientation contrast camouflages compositional contrast, so what should be learn from the image? That we can see something in the CL (d) what we might have seen in the BSE if the sample wouldn't have had EBSD-quality polishing?

The BSE signal is showing both, orientation and chemical contrast, independent of Syton-polishing (see also comment to point 7). Orientation contrast does not camouflage compositional contrast inside single grains in this case, but leads to an additional contrast between grains. CL images shows internal structures, which do not cause a strong enough contrast in BSE-images.

20) Fig. 12: Where do color artefacts (center lower part and lower left) in (b) come from?

(c) please use a proper kernel for contouring (e) misorientation angle distribution of "albite" Pole figures of pixels or 1 point-per grain? How many data points? It looks like both, ab and kfs is colorcoded in the ipf map: Is that useful? How should one distinguish both there? "maximum mud …": Maxima of pole figures are often relatively meaningless, especially if a relatively arbitrary kernel seems to be chosen or multiple maxima exist. The 2-norm of the pole figure (sometimes called pfJ) or any other measure that suits the symmetry and application might be better, or any of proper measures for orientation distribution functions.

We think the color artefacts come from a high density of unusual twins (not albite/pericline), which are present in these grains. We could not find these twins at any other occasion.

We will present contoured plots, calculated from the ODF, with texture index and pole figure strength as measured. We also will give contour intervals. ODFs were calculated for the mean orientation of grains. The one Kfs grain will not be colored in IPF-colors any more.

Changes in Fig. 12:
We now exclude Kfs from the ipf-coloring in (b). The pole figures in (c) and (d) where recalculated from odf and texture index and pfj are given now. We now give an area normalized distribution in (f).

21) Fig. 13: "Preferred growth parallel to stretching lineation" Why would it grow parallel to the finite stretching direction - unless the pure shear p.d. contribution is very large shouldn't it grow parallel to the extending ISA and eventually rotate? All figures, where a shear sense is available but not provided, should have nice arrows indicating the shear sense.

Please see also comments to points 28, 46.

Changes in Fig. 14: "Growth, granular flow, dislocation creep of quartz -> SPO, alternating quartz-albite layers"

22) General notes on figures: Please make sure the reproduced quality will be better than in the manuscript. I assume that the authors submitted high quality figures - and I am aware of the eagerness of file size reduction at the cost of quality at the side of the Copernicus graphics office/ layout people - so please double check later, that the quality of figures remains very good.

Thank you, we will take care of that.

23) A few more notes: p1, l12: Doesn't kinking and twinnign indicate that glide can't be too effective in accommodating deformation?

Twinning involves glide of dislocations (e.g., Groshong 1988).

24) p1. l21: layers … parallel to the foliation rather than lineation

We corrected this.

25) p3,l10ff: The last paragraph of the introduction reads like a conclusion, or at least mentions the processes which are later interpreted based on specific microstructures. Is that intentional?

We rephrased this part.

26) p5. l7: Was ebsd da cleaned of orientation noise? That's usually a good idea before doing KAM/ gKAm

Data was cleaned with a half-quadratic filter before gkam.

This is now more comprehensively described in the methods section.

27) p5, l17: sentence

We changed that sentence to "The K-feldspar is Na-poor (<10%) and rarely shows perthitic exsolution." See also our comment to the first referee (point 11.).

28) p5, l30: dilation or extension? (also in other places, please clarify why you think it is dilation and not simply not sites of e.g. lower P)

In this specific sentence, we refer to "strain shadows".

During dissolution-precipitation creep, boundaries at high angle to the extensional direction are "sites"where new material is precipitated for example in veins or strain shadows, causing dilation/extension in this specific direction and represented in the finite strain state by the stretching lineation *x* (e.g., Groshong, 1988; Passchier and Trouw, 2005,Wassmann and Stöckhert, 2013). See also our response to comment 30.

Groshong, H., 1988. Low-temperature deformation mechanisms and their interpretation. Geol. Soc. Am. Bull. 100, 1329–1360.

Passchier, C.W., Trouw, R.A.J., 2005. Micro-Tectonics. Springer, Heidelberg 159–187.

Wassmann, S., Stöckhert, B., 2013. Rheology of the plate interfaced - dissolution precipitation creep in high pressure metamorphic rocks. Tectonophysics 608, 1-29.

29) p6, l33, p7 1ff: Quantifying lattice bending using a misorientation angle wrt origin as a function of distance is not very satisfying since this may only make sense if it can be reasonably assumed that all misorientation is realised around the same axis and rotations remain so low (or at a given symmetry element) that crystal symmetry does not yet matter.

We give the misorientation angle along a distance in addition to the gKam value, as we find this information more intuitively and indeed it refers to the continuous bending of a crystal that is already visible in polarized light micrograph.

30) p7, l2: LAB parallel to shortening direction: anything quantitative on that? We could not detect continuous LABs in this direction with MTEX. Also, where is the shortening direction? First paragraph of section 4.: "The foliation and stretching lineation are characterized by large fragmented magmatic tourmaline and feldspar, here referred to as porphyroclasts and alternating quartz-, albite- and mica-rich layers (Fig. 2a-d). The plane normal to the foliation and the stretching lineation are taken as the principal axes of the finite strain ellipsoid *z* and *x*, respectively, which are indicated in micrographs."

31) p7, l22: What are (monophase) layers composed of aggregates.

Monophase means just one mineral phase, aggregate means different grains (of the same phase, but this is already included in the term "grain").

32) p7, l29: How were traces of planes related to real 3d boundary planes?

We will make clear, that these straight segments can be parallel to traces of (001) and (010) cleavage planes.

 "Straight segments can be parallel to the traces of (001) and (010) cleavage planes traces, representing energetically favoured boundaries (e.g., Tröger, 1982)." (see point 39)

33) p8, l2: i.. not show an internal orientation contrast ...

The term "internal" is added.

34) p9,l11-15 (but also elsewhere): Observations and interpretations of the authors are mixed with references to the literature in a way making it hard to figure what information is claimed by the authors and what comes from literature. These sections can benefit from a more clear separation of citation and authors interpretation.

We revised the text accordingly.

35) p9. l21: influence of water on diffusion e.g. R&D2004: this most likely relates to gbtransport phenomena, at least it was never demonstrated that it is intracrystalline diffusion, hence it's a bit of a brave jump to speculate on climb enhanced by the presence of fluid - or the absence

This is a misunderstanding, we did not mean to speculate on the enhancement of climb by the presence of fluid. We referred to findings from the literature and transferred a few of these aspects into the introduction (see point below).

36) p9,l15-26: this is a collection of citations in relation to the inability of dislocation climb and the sluggishness of diffusion in the absence of a hydrous fluid. However, this section might be better placed into the introduction.

We agree and placed a few of these aspects into the introduction.

37) p9,l30: "as opposed to solid state grains boundary migration": please explain/clarify; there needs to be transport across the boundary in each case

Our point was to stress the role of dissolution-precipitation as opposed to for example climb-involved subgrain rotation recrystallization (Drury and Urai, 1990; Schenk and Urai, 2005; Stipp and Kunze, 2008).

Subsequently, grains grow by strain-induced grain boundary migration, where crystalline volume with higher strain energy is dissolved and strain-free crystalline volumes precipitated, as suggested by Tullis and Yund (1987).

38) p10. l1ff(and earlier): While all reasonable in very general terms and something one would expect for such a rock, here a few ingredients to the interpretation are somewhat speculative: a) glide and b) strain induced gbm are not demonstrated. While both may be likely, here it remains a speculation since it is not backed by any (semi) quantitative data

Here, we disagree, the indication of dislocation glide is not speculative (see comments to referee #1 (points 31 and 34 and point 3 above). Dislocation glide is demonstrated, e.g., by twinning (which involves glide of dislocations, e.g., Groshong 1988, see point 23) and the continuous bending of the crystal lattice. Even the formation of "micro-crush zones" sensu Tullis and Yund (1987) involves dislocation glide. It would be much more speculative to argue that continuous bending of a crystal is purely brittle, especially without any evidence of fracturing on SEM and polarized-light microscopic scales.We will discuss this in some detail in Chapter 5.2. Strain-induced grain boundary migration is likely by the presence of new grains that are basically strain-free and not represent fragments of the original clasts, we will further strengthen this important point when revising the manuscript. However, also overgrowth of grains by precipitation parallel to the extensional direction during deformation will be important in addition to strain-induced grain-boundary migration.

39) p10. l8: reaction of fracture to crystal directions: a) How was this investigated? and B) is there any data on that?

For K-feldspar, we investigated this by comparing the orientation of microcracks to EBSD data. See, for example Fig. 5. Fractures in K-feldspar are clearly related to the axes of the finite strain ellipsoid and not to the crystallography of the crystal. For fragmentation of plagioclase in the "micro-crush" zones, however, cleavage fractures might indeed play a role.

new version:

Section 4.4. "Straight segments can be parallel to the traces of (001) and (010) cleavage plane, representing energetically favoured boundaries (e.g., Tröger, 1982)."

Section 5.3 "A preferred crystallographic relation of the intragranular fractures was not detected given their orientation at high angle to the stretching lineation independent on crystallographic orientation (Fig. 5), ruling out a major influence of cleavage fractures."

40) p10, l19: Dilation: Please explain, it this true dilation or low P sites or surfaces near orthogonal to extensional directions?

See comment to comment to point 28.

41) p11,l8: albite aggregates instead of albite taking up some deformation

The aggregates formed from strained albite porphyroclasts…

42) p11, l15: grain boundary sliding: while one can see a few straight boundaries in Fig.13, a) why should they indicate gbs b) how frequent are those compared to others ? Anything more convincing on gbs?

We indeed do not have further microstructural evidence on granular flow except of the fine-grained albite layers deflected around porphyroclasts with minor straight boundaries but mostly lobate boundaries (Fig. 13).

"We suggest that after grain size reduction, the fine-grained albite matrix was undergoing a mixture of dissolution-precipitation processes, microcracking and sliding of grains, commonly referred to asgranular flow(e.g., Behrmann and Mainprice, 1987; Stünitz and Fitz Gerald, 1993; Jiang et al., 2000). Sliding might have occurred along straight boundaries weakly inclined to the foliation (Fig. 13b). Microcracking is indicated by the fractures at high angles to the stretching lineation (Fig. 13a, b, e).The weak zoning of grains (Figs. 11 and 13) suggests the involvement of dissolution-precipitation. Granular flow would also cause weakening of adomainal CPO resulting fromthe replacement of albite porphyroclasts (e.g. Jiang et al., 2000; Hildyard et al., 2011)."

43) p11, l19: Hildyard needs year

 (Hildyard et al., 2011).

44) p11, l25: "Microstructure correlates with strain": again, where does strain come from? How does such a "correlation" manifest? Simply elongated vs more equiaxed grains?

Please see comment to point 7).

45) p11, l25: "The higher ..." Sentence

We corrected the sentence.

46) p11. l27: growth parallel to the stretching lineation: While this does not make a lot of sense for non-coaxial p.d. (see comment 2), why preferred growth? Preferred by what? Crystallography? Where should the "dilation" come from? Anything tested on that? What is the CPO of the most elongated grains, or which crystal direction is parallel to the maximum grain elongation direction?

The observation is: The long axes of grains is within the foliation plane. There is no preferred crystallographic orientation of grains with high aspect ratio (see also comments to points 21 and 28). We rephrased the last paragraph of chapter 5.5. accordingly.

47) Entire section 5.6 does not allow me to understand which by now is the process that dominates rheology.

We rewrote the discussion in this chapter and changed the heading to: "Implications for rock rheology and deformation history.", see point 1.

48) p12, l1: "granular flow" (here and elsewhere) please define your understanding of granular flow within the context of a mylonite. Or do you refer to grain boundary sliding in the sense of Rachinger sliding?

Please see comments to point 42).

49) p12, l7: why probably?

We omitted "probably", as this is indeed too speculative.

50) p12. l7: Please enlighten (probably not in the conclusion) why the lobate boundaries between newly grown albite and kfs should be chemical disequilibrium and not due to other driving forces, i.e. gb-width, porosity variations in kfs, defect densities etc.?

We agree that not only chemical disequilibrium but also other factors do play a role for the replacement and discuss this more comprehensively in the section 5.1. However, as albite does replace K-feldspar, the chemical driving force is an important factor.

Last paragraph chapter 5.1. "The cuspate sawtooth-shaped boundaries between new grains of albite and K-feldspar porphyroclasts are interpreted to indicate interface-coupled replacement (Fig. 6), supported by the porosity and apatite inclusions in albite replacing K-feldspar (Fig. 6b, d). The K-feldspar replacement is independent on the orientation of the boundary to the foliation and stretching lineation and is therefore interpreted to be not directly related to the strain field during deformation, not excluding some influence of higher strain along the boundary compared to within the crystal."

51) p12, l19: Why would glide drive gs-reduction in this combined mixture of mechanisms?

As discussed in section 5.2, dislocation glide in association with microfractures in the sense of low-T plasticity is causing a reduction in grain size similar to the "micro-crush" zone in Tullis and Yund, 1987 (see comments to points 3, 38).

52) p12,l20: "observed tendency of slightly enriched Na-content…" Any data on that?

We mention the range of compositions of both the porphyroclast and new grains (p5,18; p6,l29-30). They overlap with a slight tendency for new grains to be more Na-rich.

53) p12,l22: Why subordinate? The balance between chemical driving force vs. e.g. strain energy depends on a lot of variables. For some variables we might have good estimates while for others, we are simply guessing, i.e. dislocation density and elastic energy added by dislocations during deformation etc.

It is true that many variables are not known and we do not try to ignore this problem. Estimates on the influence of chemical driving forces, consider the chemical differences that we observe as too small to play a significant role (e.g., Stünitz, 1998). We discuss this in Chapter 5.1.

Here we show the changes made to the figures and present the new captions.

*Changes in Fig. 1:* We colored the sample numbers according to the type of albite-quartz matrix.

Fig. 1: Geologic map of the study area (modified after Mancktelow et al., 2001). The sample numbers are colored according to the type of albite-quartz matrix (see text and Table 1)

Fig. 2: Photograph of polished surface (a) and thin section micrograph taken with crossed polarizers (b) of sample CT599. K-feldspar (Kfs), albite (Ab) and tourmaline (Tur)

porphyroclasts are embedded in a fine-grained matrix. Elongate fractured tourmaline crystals are oriented with their long axes parallel to the stretching lineation (x). Fractures are commonly oriented at low angle to the shortening direction (z). White arrows point to strain shadows surrounding porphyroclast and prismatic strain shadows between fragments of tourmaline and feldspar. Black arrow points to mylonitic foliation flowing around strain shadow. (c, d)

Polarized light micrographs (crossed polarizers, sample FH5b) showing mylonitic foliation defined by quartz layers (Qz) flowing around garnet (Grt) and albite porphyroclasts (Ab), which are partly disintegrated into a fine-grained albite matrix.

*Fig. 3:* BSE images from sample FH27 showing the typical accessory mineral assemblage in the deformed pegmatites: (a) Ca-rich garnet (Grt2) replacing magmatic Fe-rich garnet (Grt1).

(b) Epidote and white mica aligned in the foliation with apatite porphyroclasts.

*Changes in Fig. 4:* All pole figures have been recalculated from odf after segmentation with space group 121. Texture index (TI) and pfj are given now. We now show area fractions for grain size analysis. The reference orientation is indicated.

*Fig. 4:* (a) Asymmetric strain shadow around albite porphyroclast in sample CT599 in thin section micrograph with crossed polarizers. (b) EBSD-phase map of the same area (quartz: blue, albite: green, K-feldspar: red) and (c) EBSD-relative misorientation map (0-10°) of the albite porphyroclast. Polyphase aggregates occur mostly in the upper left and lower right of the clast.

In the other quadrants monophase albite dominates. Pole figures show the orientation of albite in the strain shadow in the upper left quadrant (d), lower left quadrant (e), albite porphyroclast (f) and in the upper right quadrant (g). Grain area distribution histograms of albite in polyphase aggregates strain shadow (h) and in monophase albite aggregates (i).

*Changes in Fig. 5:* We colored the grain boundaries between replacing albite and the Kfs porphyroclast according to their misorientation angle and exclude the grains in the fracture.

Additionally, misorientation axis and angle distributions are shown for these boundaries.

[revised manuscript text omitted]

---

## Author Response (AR2)

Dear Florian,

Thank you very much for the editorial handling and your additional comments!

We agree that there is good reason to separate instantaneous from finite strain ellipsoids, if the rocks record information about it. The mylonitic pegmatites show a foliation and stretching lineation, characterized by large fragmented magmatic tourmaline and feldspar porphyroclasts and alternating quartz-, albite- and mica-rich layers. The plane normal to the sample-scale foliation and stretching lineation are interpreted to indicate the principal axes of the finite strain ellipsoid $Z$ and $X$, respectively (as stated in the revised manuscript beginning of chapter 4 p. 6, line 9, 10). The foliation can be deflected around larger porphyroclasts. In rare cases, strain shadows are asymmetric (Fig. 4) and few shear bands are present (Fig. 12 a, b). The scale that we are addressing when we discuss the development of the microstructure is the sample scale. We do not intend to discuss the relative contributions of pure shear / simple shear / co-axial / non-coaxial deformation or any kinematics on a larger scale, which is beyond the scope of this study. We changed our wording throughout the manuscript and refer to the observed sample-scale foliation and stretching lineation, as this is the objective observation.

In the submitted final manuscript we made the following changes:

At the beginning of chapter 4 we rephrased to: "The plane normal to the observed foliation and the stretching lineation on sample-scale are interpreted to represent the principal axes of the finite strain ellipsoid $Z$ and $X$, respectively, which are indicated in micrographs."

In the captions to Fig. 2b, Fig. 5a, b, Fig. 13 we now also refer to the observed stretching lineation / foliation (as opposed to referring to a "shortening direction", which we used in the sense of the plane normal to the observed foliation on sample scale, but we understand that this wording is misleading).

We omitted p9, l2-3. ("The porosities parallel to the short axes of grains, the elongate shape and the zoning indicate that grains grew at boundaries perpendicular to the stretching lineation".)

A few additions to the specific comments of Ruedigers review:

21) Fig. 13: "Preferred growth parallel to stretching lineation" Why would it grow parallel to the finite stretching direction - unless the pure shear p.d. contribution is very large shouldn't it grow parallel to the extending ISA and eventually rotate? All figures, where a shear sense is available but not provided, should have nice arrows indicating the shear sense.

It is true that a grain would not necessarily grow parallel to the finite stretching direction, but likely that it grows parallel to the extending ISA. In our samples the long axis of most matrix grains is within the foliation. Whether any rotation relative to a microstructure on sample scale or larger scale was involved is not recorded.

In Fig. 13 we did not refer to "preferred growth parallel to the stretching lineation" therefore we assumed that Ruediger was referring to Fig. 14, where we rephrased that statement accordingly. As indeed the phrase "preferred" might have been misleading (see also point 46) we rephrased this throughout the revised manuscript.

The microstructures shown in the SE-images of Fig. 13 do not indicate a shear sense.

30) p7, l2: LAB parallel to shortening direction: where is the shortening direction?

We rephrased that sentence in the revised manuscript to:

"Low-angle boundaries are typically observed oriented at high angle to the stretching lineation, indicating that they represent healed cracks associated with a slight misorientation rather than indicating subgrains (Fig. 8a)."

46) p11. l27: growth parallel to the stretching lineation: While this does not make a lot of sense for non-coaxial p.d. (see comment 2), why preferred growth? Preferred by what? Crystallography? Where should the "dilation" come from? Anything tested on that? What is the CPO of the most elongated grains, or which crystal direction is parallel to the maximum grain elongation direction?

The paragraph 5.5 has been rewritten in the revised manuscript, such that the sentence, Ruediger was referring to in the original manuscript, is not present anymore.

The phrase "preferred" might have been misleading (see also point 21) and we rephrased this throughout the revised manuscript. We did not refer to a crystallographic preferred growth but formation of growth rims causing a SPO (not CPO) with the long axes of grains within the foliation plane. The observation is: The long axes of grains is within the foliation plane. There is no preferred crystallographic orientation of grains with high aspect ratio (see also comments to points 21 and 28).

To the comments on "extension / dilation" we answered at point 28 the following: During dissolution-precipitation creep, boundaries at high angle to the extensional direction are "sites", where new material is precipitated for example in veins or strain shadows, causing dilation/extension in this specific direction. Throughout the revised manuscript, we used the term "strain shadow" to refer to these areas.

We hope that our manuscript is now acceptable for publication in Solid Earth.

Thank you and with best regards,

Felix Hentschel

[revised manuscript text omitted]